# PRE-TRAINING PURE GNNs AS GRAPH LEARNERS

## ABSTRACT

Graphs from different datasets exhibit diverse numbers of features and labels, where each feature or label is associated with different semantic meanings. Such diversity poses challenges in adapting pre-trained graph neural networks (GNNs) to different datasets with a single set of input and output (I/O) module parameters. This raises a fascinating question: Can pure GNNs be pre-trained on diverse datasets, adapting to various datasets effectively without additional effort? To explore this, we propose unified I/O modules that enable pre-training with pure GNNs. Unlike traditional methods that tightly couple parameters to specific datasets, our approach decouples parameters through a shared relation function for the input and uniformly sampled points for the output. These designs effectively resolve the challenges in quantity inconsistency and semantic discrepancies of dataset features and labels. By integrating our I/O modules with various GNN architectures, we demonstrate that pure GNNs can be effective graph learners for direct adaptation to downstream tasks. Pre-training experiments under different setups show that increasing hidden dimensions and the average number of nodes per training dataset enhances model performance. Moreover, fine-tuning the I/O modules with frozen pre-trained graph operators significantly simplifies the model hyperparameter tuning process, achieving superior or comparable performance to supervised models on downstream datasets.

## 1 INTRODUCTION

Pre-trained foundation models have shown exceptional adaptability across different datasets, such as large language models (LLMs) (OpenAI et al., 2024) for MMLU (Hendrycks et al., 2020) and HumanEval (Chen et al., 2021), large vision models (LVMs) (Kirillov et al., 2023) for CityScapes (Cordts et al., 2016) and PIDRay (Zhang et al., 2023a). A key factor underpinning this capability lies in the unified feature and label space across datasets. In the feature space, although the encoded information varies, the semantic meanings of features remain consistent. For instance, a landscape image and a colored X-ray image can both be represented as 3D tensors, with features corresponding to their RGB values. In the label space, data can be assigned to a limited number of labels.

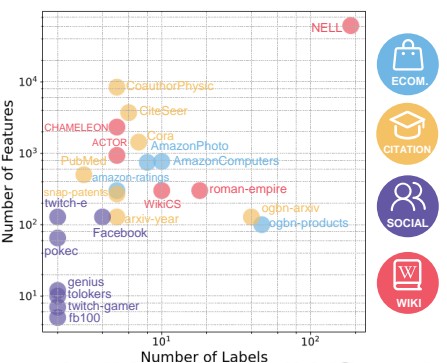

Figure 1: **Diversity of Features and Labels across Graph Datasets.** Graphs from different datasets exhibiting diverse numbers of features and labels, while each feature or label is associated with different semantics.

In contrast to data with the unified feature and label space, graph data typically exhibits diverse numbers of features and labels, with each associated with different semantics (Fig. 1). For instance, feature and label semantics in tolokers (Platonov et al., 2023) correspond to user profiles and user status, whereas those in CoauthorCS (Shchur et al., 2019) correspond to publication keywords and research interest. This inherent diversity presents substantial challenges in developing pre-trained graph models. First, the inconsistent number of features and labels hinders the unified design of the input/output modules (I/O). Second, the semantic discrepancies across graphs impede the effective adaptation of pre-trained models to diverse graph datasets. Traditional I/O fails to tackle these challenges by tightly coupling the quantity and values of the learnable parameters to specific datasets, thereby limiting their adaptability.

To address this problem, methods have proposed leveraging language models (LMs) as I/O for graph neural networks (GNNs) (Liu et al., 2023a; Li et al., 2024b). By transforming graph features and labels into natural language, GNNs integrated with LMs can effectively handle graphs with diverse features and labels. Beyond advancements in pre-training GNNs with LMs, researchers have also explored fine-tuning strategies (Sun et al., 2022; 2023; Huang et al., 2024) to accommodate different features and labels. However, these methods still rely on few-shot knowledge for effective adaptation. It remains an open problem that *can pure GNNs be pre-trained on diverse datasets and directly adapted to downstream datasets?*

In this paper, we propose unified I/O modules to achieve pre-training with pure GNNs by decoupling model parameters from specific datasets. Instead of treating the whole mapping matrix as learnable parameters, unified I/O decomposes the mapping into two successive mapping matrices: one associated with the datasets and the other associated with the hidden space. Our objective is to model the dataset-associated mapping in a transferable manner. For input features, unified I/O employs a shared parametric relation function to learn a predefined number of relations between feature dimensions. The shared function targets and analyzes the same relation patterns across different datasets, which can be employed to construct the dataset-associated mapping with the same set of parameters. For output labels, the module samples uniformly distributed points as pseudo labels, enabling prediction over diverse label spaces. When label information is available for downstream graphs, these uniform points can be aligned with the real labels without additional training.

Our unified I/O modules enable graph pre-training with pure GNNs. Empirical results with different GNN architectures demonstrate that pre-trained pure GNNs can be effective graph learners for direct adaptation to downstream tasks. Specifically, we evaluate the performance of pre-trained models with scaling parameters, scaling training data, and varying domain gaps. Results show that increasing either hidden dimensions or the average number of nodes per graph during pre-training enhances the performance. Moreover, fine-tuning the I/O modules with frozen pre-trained graph operators on downstream graphs substantially reduces the need for extensive hyperparameter tuning, achieving superior or comparable performance with the supervised models. Our contribution can be summarized as

- We propose a novel method to unify the I/O modules for pre-training with general pure GNNs, providing graph operators to simplify the extensive hyperparameter tuning process.
- We demonstrate that pre-trained pure GNNs can serve as effective graph learners on eight classic GNN architectures across diverse real-world datasets.
- We experimentally verify the adaptation performance of pre-trained GNNs with scaling parameters, scaling training data, and different domain gaps.

## 2 RELATED WORK

Due to space limitations, we provide a brief overview of related work, with a comprehensive discussion in Appendix B. Existing methods for adapting pre-trained graph models fall into two categories: I/O unification and fine-tuning. The former mainly leverage LMs to encode textual attributes for unifying diverse features and labels (Liu et al., 2023a; Wang et al., 2023; Chen et al., 2024a; Kong et al., 2024; Li et al., 2024b; Tang et al., 2024; Zhu et al., 2025). Except for LM-based methods, both parametric (Jing et al., 2023; Zhao et al., 2024b) and parameter-free (Sun et al., 2023; Tang et al., 2024; Sun et al., 2025) unification strategies for pure GNNs have been explored, but they fail to unify label spaces. GraphAny (Zhao et al., 2024a) extends this direction to label unification with linear GNNs. Yet these methods remain limited in scope and often require observed labels, leaving pre-training with general pure GNN architectures an open challenge. The latter fine-tuning methods adapt pre-trained models through graph adapters (Li et al., 2024a; Gui et al., 2024) or graph prompts (Sun et al., 2023; Yu et al., 2025). Distinct from both, our method can be applied to general GNN architectures and enables pre-training for direct adaptation to diverse datasets.

## 3 UNIFIED I/O FOR GENERAL PURE GNN ARCHITECTURES

To enable training-free adaptation of pure GNNs, we formulate the unification of the I/O modules as the modeling of the semantics associated with each feature and label (Fig. 2). Specifically, our unified

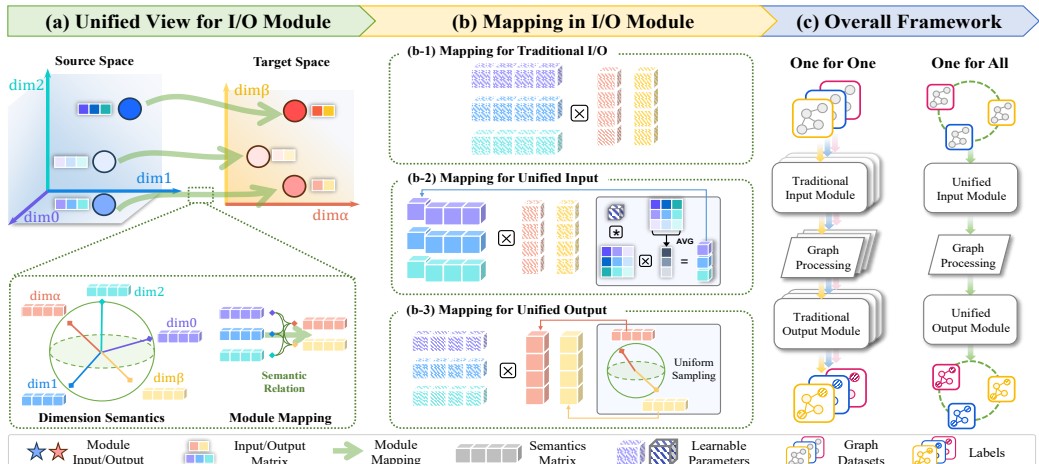

Figure 2: **Illustration of the Unified Input/Output Pipeline.** (a) A linear mapping projects source dimensions onto a set of target dimensions. Each dimension corresponds to certain semantic descriptions that can be embedded in the semantic space. Modeling semantic relations between source and target dimensions results in module mappings. (b) Instead of treating both semantic matrices as parameters, unified I/O modules model the feature semantics as parametric relations and sample label semantics uniformly in the semantic space. (c) The parameter quantity and values in GNNs with traditional I/O are coupled with specific datasets. Our unified I/O decouples the parameters from specific datasets, enabling one set of parameters for different datasets.

input module encodes the feature semantics as parametric relations of the input features, while our output module represents diverse label semantics as uniformly sampled points in the semantic space.

## 3.1 PROBLEM SETUP

**Notations.** Given an input graph $\mathcal{G} = (\mathcal{V}, \mathcal{E})$, $\mathcal{V} = \{v_1, \cdots, v_n\}$ denotes a set of $n$ nodes and $\mathcal{E} = \{e_{i,j} | v_j \in \mathcal{N}(v_i)\}$ denotes a set of $m$ edges. $\mathcal{N}(\cdot)$ denotes the set of one-hop neighbors for a given node. Each node $v \in \mathcal{V}$ corresponds to a feature vector $\mathbf{x}_v \in \mathbb{R}^{d_{in}}$ where $d_{in}$ is the number of input features. Let $\mathbf{X} = (\mathbf{x}_{v_1}, \cdots, \mathbf{x}_{v_n})^\top \in \mathbb{R}^{n \times d_{in}}$ be the node feature matrix composed of feature vectors. Let $\mathbf{A} \in \mathbb{R}^{n \times n}$ be the adjacency matrix of $\mathcal{G}$. $\mathbf{A}_{i,j} = 1$ if $e_{i,j} \in \mathcal{E}$. Let $\mathbf{C} = (\mathbf{c}_{v_1}, \cdots, \mathbf{c}_{v_n})^\top \in \mathbb{R}^{n \times c}$ be the node label matrix composed of label vectors, where $c$ is the number of labels, $\mathbf{C}_{i,j} \in \{0, 1\}$. Although we take node classification as an example in this paper, our method can be applied to general graph learning tasks (Appendix A).

**Modules in GNN.** Let $\text{GNN}(\mathcal{G}) = (\mathbf{F}_{\text{out}} \circ \mathbf{F}_{\text{g}} \circ \mathbf{F}_{\text{in}})(\mathcal{G})$ be a GNN model. The target of $\text{GNN}(\mathcal{G})$ is to optimize the sets of learnable parameters $\mathcal{W}_{\text{in}}$, $\mathcal{W}_{\text{g}}$, and $\mathcal{W}_{\text{out}}$ and form the optimal mapping for $\mathbf{F}_{\text{in}}$, $\mathbf{F}_{\text{g}}$, and $\mathbf{F}_{\text{out}}$. The input module $\mathbf{F}_{\text{in}} : \mathbb{R}^{d_{in}} \mapsto \mathbb{R}^d$ maps input features to the hidden space, yielding $\mathbf{H}^{(0)} = \mathbf{F}_{\text{in}}(\mathbf{X}; \mathcal{W}_{\text{in}})$, where $d$ denotes the hidden dimensionality and ";" separates module input from the parameters. The module $\mathbf{F}_{\text{g}} : \mathbb{R}^d \times \mathbb{R}^{n \times n} \mapsto \mathbb{R}^d$ applies general graph processing methods (Kipf & Welling, 2017; Rampášek et al., 2022), giving $\mathbf{H}^{(L)} = \mathbf{F}_{\text{g}}(\mathbf{H}^{(0)}, \mathbf{A}; \mathcal{W}_{\text{g}})$ with $L$ layers. Finally, the output module $\mathbf{F}_{\text{out}} : \mathbb{R}^d \mapsto \mathbb{R}^c$ performs prediction, giving $\hat{\mathbf{C}} = \mathbf{F}_{\text{out}}(\mathbf{H}^{(L)}; \mathcal{W}_{\text{out}})$.

Specifically, the focus of this paper is to model the mappings of the I/O modules $\mathbf{F}_{\text{in}}$ and $\mathbf{F}_{\text{out}}$, which can be uniformly formulated as $\mathbf{F}_{\text{i/o}} : \mathbb{R}^{d_{\text{src}}} \mapsto \mathbb{R}^{d_{\text{tgt}}}$, yielding $\mathbf{F}_{\text{i/o}}(\mathbf{H}; \mathcal{W}_{\text{i/o}})$. The input and output of the modules are termed as "source" and "target" to distinguish from the input and output of the whole model. $\mathbf{F}_{\text{i/o}}$ as input modules has $d_{\text{src}} = d_{\text{in}}, d_{\text{tgt}} = d, \mathbf{H} = \mathbf{X}, \mathcal{W}_{\text{i/o}} = \mathcal{W}_{\text{in}}$. $\mathbf{F}_{\text{i/o}}$ as output modules has $d_{\text{src}} = d, d_{\text{tgt}} = c, \mathbf{H} = \mathbf{H}^{(L)}, \mathcal{W}_{\text{i/o}} = \mathcal{W}_{\text{out}}$.

## 3.2 MAPPINGS OF THE I/O MODULES

To decouple the model parameters from specific datasets, we start by formulating the module mappings as dimension relations. Note that the nonlinearity in module mappings is obtained via

element-wise operations, which are independent of the I/O dimensionality. Therefore, we only focus on the linear part of the module mappings. Specifically, a linear mapping defines a projection from the source dimensions onto a new set of target dimensions (Fig. 2(a)). Learning the dimension relations enables the model to infer the optimal mapping from the source space to the target space.

**Theorem 3.1** (Mapping with Dimension Relations). *Given any linear mapping* $\mathbf{W} \in \mathbb{R}^{d_{\mathrm{src}} \times d_{\mathrm{tgt}}}$ *and* $s \in \mathbb{N}^+$, *there always exist two representation matrices* $\mathbf{S}_{\mathrm{src}} \in \mathbb{R}^{d_{\mathrm{src}} \times s}$ *and* $\mathbf{S}_{\mathrm{tgt}} \in \mathbb{R}^{d_{\mathrm{tgt}} \times s}$, *such that* $\mathbf{W} = \psi(\mathbf{S}_{\mathrm{src}}, \mathbf{S}_{\mathrm{tgt}})$, *where* $\psi(\cdot, \cdot)$ *is a bilinear composition function.*

This theorem shows that we can model the mappings as relations. A typical choice of $\psi(\cdot, \cdot)$ is the inner product form, where $\psi(\mathbf{S}_{\mathrm{src}}, \mathbf{S}_{\mathrm{tgt}}) = \mathbf{S}_{\mathrm{src}} \mathbf{S}_{\mathrm{tgt}}^\top$. Proof of Theorem 3.1 is presented in Appendix E.1. Without loss of generality, the I/O modules can be formulated as

$$\mathtt{F}_{\mathtt{i/o}}\left(\mathbf{H}; \mathcal{W}_{\mathtt{i/o}}\right) = \sigma\left[\mathbf{HW}\right] = \sigma\left[\mathbf{H}\psi(\mathbf{S}_{\mathrm{src}}, \mathbf{S}_{\mathrm{tgt}})\right] = \sigma\left[\mathbf{HS}_{\mathrm{src}}\mathbf{S}_{\mathrm{tgt}}^\top\right], \quad (1)$$

where $\sigma$ can be any nonlinear function.

**Decomposition as Semantics.** $\mathbf{S}_{\mathrm{src}}$ and $\mathbf{S}_{\mathrm{tgt}}$ are the decomposition results of the original weight matrix $\mathbf{W}$, which can be interpreted as the semantic embeddings associated with the dimensions of the source and target spaces, respectively. Each row of $\mathbf{S}_{\mathrm{src}}$ and $\mathbf{S}_{\mathrm{tgt}}$ encodes the specific semantic meaning of a dimension, typically characterizing graph nodes and their associated labels. For example, CoauthorCS (Shchur et al., 2019) provides node features representing the frequency of the paper keywords for each author's papers. These semantic descriptions can be embedded into a semantic space $\mathbb{R}^s$, giving rise to semantic embeddings such as $\mathbf{S}_{\mathrm{src}}$ and $\mathbf{S}_{\mathrm{tgt}}$.

**Problems in Traditional Solutions.** In traditional graph learning solutions (Zhou et al., 2020), $\mathtt{F}_{\mathtt{i/o}}$ is highly sensitive to the source and target spaces. It directly treats the space semantics as parameters for mapping, where $\mathbf{S}_{\mathrm{src}}, \mathbf{S}_{\mathrm{tgt}} \in \mathcal{W}_{\mathtt{i/o}}$. For instance, a single-layer perceptron can be formulated as $\sigma\left(\mathbf{HS}_{\mathrm{src}}\mathbf{S}_{\mathrm{tgt}}^\top; \mathbf{S}_{\mathrm{src}}, \mathbf{S}_{\mathrm{tgt}}\right)$. Consequently, the learned parameter set $\mathcal{W}_{\mathtt{i/o}}$ becomes intrinsically tied to the specific source and target spaces, with its values tailored to particular spaces and its quantity scales to the number of dimensions in those spaces. This inherent sensitivity significantly limits the adaptability of pure GNNs to diverse datasets.

**Our Solution.** To address this issue, we propose to decouple the parameters $\mathcal{W}_{\mathtt{i/o}}$ from the source and target dimension semantics (Fig. 2), where feature semantics are redefined as parametric relations and label semantics are sampled uniformly in the semantic space.

### 3.3 SOURCE-ADAPTIVE INPUT MODULE

A input module $\mathtt{F}_{\mathtt{in}}$ maps the input features to the hidden space. Based on Eq. 1, $\mathtt{F}_{\mathtt{in}}$ can be formulated as $\mathtt{F}_{\mathtt{in}}(\mathbf{X}) = \sigma[\mathbf{XW}_{\mathtt{in}}] = \sigma[\mathbf{XS}_{\mathrm{src}}^{(\mathrm{in})}\mathbf{S}_{\mathrm{tgt}}^{(\mathrm{in})\top}]$, where $\mathbf{S}_{\mathrm{tgt}}^{(\mathrm{in})} \in \mathbb{R}^{d \times s}$ is a learnable parameter matrix. Our focus is modeling the source space semantics $\mathbf{S}_{\mathrm{src}}^{(\mathrm{in})} \in \mathbb{R}^{d_{\mathrm{in}} \times s}$ regarding specific inputs, *i.e.*, the semantics of the input features $\mathbf{X}$. To decouple the parameters from the number of source dimensions, $\mathbf{S}_{\mathrm{src}}^{(\mathrm{in})}$ can be formulated as a parametric function of $\mathbf{X}$:

$$\mathtt{f}_{\mathtt{in}} : \mathbb{R}^{n \times d_{\mathrm{in}}} \mapsto \mathbb{R}^{d_{\mathrm{in}} \times s}, \quad \mathbf{S}_{\mathrm{src}}^{(\mathrm{in})} = \mathtt{f}_{\mathtt{in}}(\mathbf{X}; \mathcal{W}_{\mathtt{in}}), \quad (2)$$

where $\mathtt{f}_{\mathtt{in}}(\cdot)$ is subject to two conditions: (1) Permutation invariance to the order of input nodes and equivariance to that of source dimensions; (2) Size independence of the parameter set $\mathcal{W}_{\mathtt{in}}$ to the values of $n$ and $d_{\mathtt{in}}$. Input edges are disregarded in Eq. 2 as our focus lies in unifying the input module across different node features, while the unification of graph structures is left for future work.

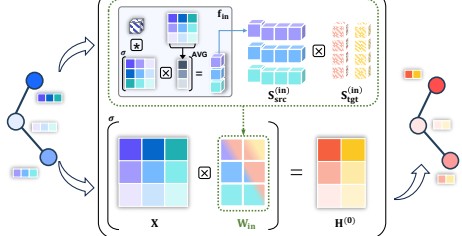

Figure 3: **Pipeline for the Unified Input Module.** A parametric relation function is employed to construct the source space semantics $\mathbf{S}_{\mathrm{src}}^{(\mathrm{in})}$, *i.e.*, the semantics of input features $\mathbf{X}$. The target space semantics $\mathbf{S}_{\mathrm{tgt}}^{(\mathrm{in})}$ is a learnable parameter matrix.

**Modeling Features as Sets.** Given the absence of graph structures and the permutation conditions for Eq. 2, the input features can be modeled as a set of channels $\{\mathbf{X}_{\cdot, j}\}$, where each channel corresponds to a set of nodes $\{\mathbf{X}_{i, j}\}$. As a result, Eq. 2 is transformed into a set-learning problem at both the channel level and the node level. Based on the universal functions for set learning (Zaheer et al.,

2017), $\mathtt{f}_{\mathtt{in}}(\cdot)$ can be formulated as follows. See Appendix E.2 for detailed derivation.

$$\mathtt{f}_{\mathtt{in}}(\mathbf{X}) = \sigma\left[\Theta\rho(\mathbf{X}^\top)\mathbf{1}\alpha^\top\right], \quad \Theta = \frac{d_{\mathtt{in}}\mathbf{X}^\top\mathbf{X}}{\mathbf{X}^\top\mathbf{X}\mathbf{1}}. \tag{3}$$

$\Theta \in \mathbb{R}^{d_{\mathtt{in}} \times d_{\mathtt{in}}}$ denotes the channel mixer that provides global information. $\rho(\cdot)$ computes the channel representations. $\mathbf{1}$ denotes the all-one vector and $\alpha \in \mathbb{R}^{s \times 1}$ is a learnable vector. The multiplication of $\alpha$ enables different activations through the nonlinearity of $\sigma(\cdot)$, giving $s$ row embeddings in $\mathbf{S}_{\mathtt{src}}^{(\mathtt{in})}$. Although Zaheer et al. also provides an implementation named Deep Sets based on the universal set function, our input module differs from this specific implementation regarding problem formulation, conditions, and operator selections. Please refer to Appendix E.2 for a detailed discussion.

**Unification via Relations.** A direct implementation for $\rho(\cdot)$ is to take the specific values in $\mathbf{X}$ as channel representations. However, identical numerical values across different source spaces may correspond to entirely different semantics, making it nontrivial to uniformly map these values into a common semantic space via $\mathtt{f}_{\mathtt{in}}$. To tackle this, we propose modeling the channel relations as a proxy for feature semantics. By applying a shared relation function on feature channels, $\rho(\cdot)$ measures the same relation patterns across diverse source spaces. Crucially, the extracted patterns, such as similarity and co-variation, carry consistent semantic meaning. This consistency makes them naturally comparable and provides a stable foundation for unified semantic mapping. Among the typical choices for relation measure, such as Euclidean distance and inner product, we implement $\rho(\cdot)$ with the scaled product for its computing efficiency and training stability, giving $\rho(\mathbf{X}^\top) = \mathbf{X}^\top\mathbf{X}/\sqrt{n}$.

**Source-adaptive Input.** Compiling Eq. 1-3 gives rise to the input module as

$$\mathbf{H}^{(0)} = \mathtt{F}_{\mathtt{in}}\left(\mathbf{X}; \mathbf{S}_{\mathtt{tgt}}^{(\mathtt{in})}, \alpha\right) = \sigma\left[\mathbf{X}\mathbf{W}_{\mathtt{in}}\right] = \sigma\left[\mathbf{X}\mathtt{f}_{\mathtt{in}}(\mathbf{X})\mathbf{S}_{\mathtt{tgt}}^{(\mathtt{in})\top}\right], \tag{4}$$

where the mapping changes with specific input features, forming source-adaptive input for different datasets. $\mathtt{F}_{\mathtt{in}}$ is permutation invariant to the order of the source dimensions (see Appendix E.3) and has the parameter quantity independent of $d_{\mathtt{in}}$.

### 3.4 TARGET-INSENSITIVE OUTPUT MODULE

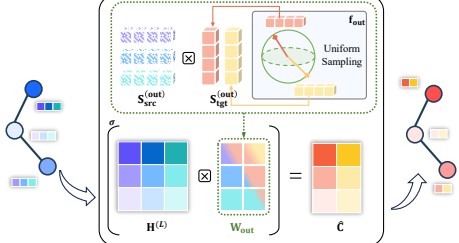

An output module $\mathtt{F}_{\mathtt{out}}$ maps the hidden representations to the label space. Based on Eq. 1, $\mathtt{F}_{\mathtt{out}}$ can be formulated as $\mathtt{F}_{\mathtt{out}}(\mathbf{H}^{(L)}) = \sigma[\mathbf{H}^{(L)}\mathbf{W}_{\mathtt{out}}] = \sigma[\mathbf{H}^{(L)}\mathbf{S}_{\mathtt{src}}^{(\mathtt{out})}\mathbf{S}_{\mathtt{tgt}}^{(\mathtt{out})\top}]$, where $\mathbf{S}_{\mathtt{src}}^{(\mathtt{out})} \in \mathbb{R}^{d \times s}$ is a learnable parameter matrix. Unifying the output module requires the modeling of the target space semantics $\mathbf{S}_{\mathtt{tgt}}^{(\mathtt{out})}$, *i.e.*, the semantics of the labels. However, the label knowledge of downstream datasets is typically unavailable. To tackle this problem, we propose a two-step approach, including prediction and assignment. In the prediction step, $\mathtt{F}_{\mathtt{out}}$ uniformly samples $c$ points in $\mathbb{R}^s$ as pseudo-label semantics. By learning the relations between the parameter matrix $\mathbf{S}_{\mathtt{src}}^{(\mathtt{out})}$ and the pseudo-label semantics, the module can make predic-

Figure 4: **Pipeline for the Unified Output Module.** Target space semantics $\mathbf{S}_{\mathtt{tgt}}^{(\mathtt{out})}$, *i.e.*, the semantics of pseudo labels, are selected uniformly in the semantic space. The source space semantics $\mathbf{S}_{\mathtt{src}}^{(\mathtt{out})}$ is a learnable parameter matrix.

tions without prior label knowledge. In the assignment step, the module assigns the pseudo labels to the actual labels of each dataset, enabling precise adaptation to diverse target spaces.

**Target-insensitive Prediction.** The pseudo labels are independent of specific datasets, so their relations should remain consistent, without assuming that some labels are inherently closer than others. To ensure this, we uniformly sample pseudo labels in the semantic space. Note that the output module implements the relation function $\psi$ as an inner product, quantifying relations by the angles between vectors. Accordingly, pseudo-label semantics are uniformly sampled on the unit sphere and mapped into Cartesian coordinates. In the $s$-dimensional spherical coordinates, a pseudo-label semantic vector can be denoted as $(1, \theta_1, \cdots, \theta_{s-1})$ with $\theta_i \in [0, \pi]$. To ensure the coverage of the unit sphere and maintain a uniform density of the semantics across all dimensions, we sample a number of $c^{1/(s-1)}$ values of equal intervals in $[0, \pi]$ for each $\theta_i$ and consider all combinations of these values. This results in a number of $c$ semantic vectors distributed uniformly

on the $s$-dimensional unit sphere. The $j$-th semantic vector can be converted into the Cartesian coordinates with $\mathbf{S}^{(\text{out})}_{\text{tgt}_{j,i}} = \sigma[\prod_{k=1}^{i-1} \sin^{\text{sgn}(i>1)}(\theta_k)\cos^{\text{sgn}(i<s)}(\theta_i)]$, $i \in \{1, \cdots, s\}$ (Blumenson, 1960). sgn denotes the sign function, $\text{sgn}(\text{True}) = 1$ and $\text{sgn}(\text{False}) = 0$. Compiling the space division process as $\mathbf{S}^{(\text{out})}_{\text{tgt}} = \mathtt{f}_{\text{out}}(c, s)$, the output module can be formulated as

$$\hat{\mathbf{C}} = \mathtt{F}_{\text{out}}\left(\mathbf{H}^{(L)}; \mathbf{S}^{(\text{out})}_{\text{src}}\right) = \sigma\left[\mathbf{H}^{(L)}\mathbf{W}_{\text{out}}\right] = \sigma\left[\mathbf{H}^{(L)}\mathbf{S}^{(\text{out})}_{\text{src}}\mathtt{f}_{\text{out}}(c, s)^{\top}\right]. \quad (5)$$

The implementation of the nonlinear function $\sigma$ depends on the specific tasks (Appendix A).

**Pseudo Label Assignment.** The prediction step enables the output module to make predictions without prior knowledge of the target labels. When the label knowledge is available, one can further assign the pseudo labels to the real labels. In this paper, we consider two assignment strategies. (1) To explore the potential of pre-trained pure GNNs on general downstream tasks, observed target labels are only included during inference to match pseudo labels with real labels. Given the observed real labels $\mathbf{C}$, the mapping relations between the pseudo labels and the real labels can be formulated as $\hat{\mathbf{C}}\mathbf{P}(c\mathbf{I} - \mathbf{1}\mathbf{1}^{\top}) = c\log(c\mathbf{C})$. The assignment matrix $\mathbf{P}$ can be solved as the least-squares solution of the linear equation without additional training. For detailed derivation, please refer to Appendix E.4. (2) To ensure fair comparison with LM-based graph models in the zero-shot setting, we replace observed labels with label knowledge by employing the embedded label semantics $\mathbf{S}^{\text{LM}}_{\text{tgt}}$ from language models (Li et al., 2024b) and construct the assignment matrix with $\mathtt{f}_{\text{out}}(c, s)\mathbf{S}^{\text{LM}^{\top}}_{\text{tgt}}$.

### 3.5 Pipeline and Pre-training Strategy

GNNs with unified I/O take a mini-batch from a single dataset as input and are optimized on different datasets sequentially. During pre-training, GNNs only perform the first prediction step in the output module. Consequently, traditional optimization objectives that require strict alignment between the ordering of outputs and labels become inapplicable, such as cross-entropy loss and mean squared error. To address this problem, we draw inspiration from contrastive loss (Qiu et al., 2020) and propose to optimize the predicted distributions within the same class and across different classes. Given the node sets of each class $\{\mathcal{V}_1, \cdots, \mathcal{V}_c\}$, the loss function is formulated as

$$\mathcal{L} = \frac{1}{n}\left(\underbrace{\sum_{k,i,j,v_j \in \mathcal{V}_i}\left|\bar{\mathbf{C}}_{i,k} - \hat{\mathbf{C}}_{j,k}\right|}_{\mathcal{L}_{\text{inner}}} + \underbrace{2 - \frac{1}{c-1}\sum_{k,i,j,v_j \notin \mathcal{V}_i}\left|\bar{\mathbf{C}}_{i,k} - \hat{\mathbf{C}}_{j,k}\right|}_{\mathcal{L}_{\text{intra}}}\right), \quad \bar{\mathbf{C}}_{i,\cdot} = \frac{1}{|\mathcal{V}_i|}\sum_{j,v_j \in \mathcal{V}_i}\hat{\mathbf{C}}_{j,\cdot}, \quad (6)$$

where $\bar{\mathbf{C}} \in \mathbb{R}^{c \times c}$ denotes the average prediction of each class. The inner-class loss, $\mathcal{L}_{\text{inn}}$, minimizes the prediction variance within the same class by ensuring that they are close to their class average. In contrast, the intra-class loss, $\mathcal{L}_{\text{int}}$, encourages differences in predictions across classes. To ensure positivity, $\mathcal{L}_{\text{int}}$ employs a constant bias of 2.

## 4 Experiment

### 4.1 Unified I/O enables Pre-training with Pure GNNs

We now evaluate pre-trained GNNs with unified I/O modules, focusing on (1) whether pure GNNs can be pre-trained on diverse datasets and directly adapted to downstream datasets; (2) how pure GNNs generalize under different conditions, *i.e.*, the amount of training data and parameters, and the gap between the training and inference domains. Various real-world datasets of different scales are adopted from four domains (Tab. S11), including electronic commerce (e-com.), citation, social, and Wikipedia (wiki) graphs. Eight GNN methods are employed as the backbone, including GCN (Kipf & Welling, 2017), GraphSAGE (Hamilton et al., 2017), GAT (Veličković et al., 2018a), GIN (Xu et al., 2019), MixHop (Abu-El-Haija et al., 2019), DeepGCN (Li et al., 2019), GraphGPS (Rampášek et al., 2022), and $\mathbf{N}^2$ (Sun et al., 2024). Detailed experimental setup can be found in Appendix C.

**Non-textual Datasets.** To evaluate the pre-trained graph models on traditional datasets with non-textual attributes, we conduct a comparison with models that can make inference on downstream

Table 1: **Evaluation Results on Non-textual Datasets (Measured by accuracy except ROC AUC for tolokers: %).** Bold values denote the best results per test dataset. SUP denotes the best-performing results among supervised baselines. SSL denotes the best-performing results among self-supervised baselines. LP, Lap, and Rand denote Label Propagation, Laplacian decomposition, and random projection, respectively.

| | AMAZON COMPUTERS | CORA | COAUTHOR PHYSICS | ARXIV -YEAR | TWITCH -GAMER | TOLOKERS | CHA- MELEON | ACTOR | AVG. RANK |
|---|---|---|---|---|---|---|---|---|---|
| ORIGINAL SPLIT | | | | | | | | | |
| SUP | $\mathbf{91.09}_{\pm 0.13}$ | $81.80_{\pm 0.28}$ | $\mathbf{95.52}_{\pm 0.20}$ | $\mathbf{48.03}_{\pm 0.41}$ | $\mathbf{61.09}_{\pm 0.32}$ | $\mathbf{80.72}_{\pm 0.26}$ | $61.74_{\pm 0.30}$ | $31.34_{\pm 0.20}$ | $\mathbf{1.87}$ |
| SSL | $89.28_{\pm 0.34}$ | $81.50_{\pm 0.66}$ | $92.32_{\pm 0.08}$ | $41.63_{\pm 0.27}$ | $59.19_{\pm 0.05}$ | $75.92_{\pm 0.18}$ | $61.01_{\pm 0.72}$ | $27.61_{\pm 0.27}$ | 5.62 |
| FUG | $88.22_{\pm 0.09}$ | $30.70_{\pm 0.96}$ | $91.09_{\pm 0.72}$ | $42.54_{\pm 0.30}$ | $58.16_{\pm 0.27}$ | $75.95_{\pm 0.34}$ | $22.59_{\pm 0.06}$ | $25.53_{\pm 0.11}$ | 7.15 |
| GRAPHANY | $82.94_{\pm 0.82}$ | $79.41_{\pm 0.35}$ | $92.43_{\pm 0.21}$ | $38.36_{\pm 0.53}$ | $59.96_{\pm 0.02}$ | $78.16_{\pm 0.18}$ | $61.84_{\pm 0.81}$ | $28.75_{\pm 0.69}$ | 5.25 |
| LABEL PROPAGATION | $87.27_{\pm -}$ | $81.53_{\pm -}$ | $95.67_{\pm -}$ | $17.02_{\pm -}$ | $58.30_{\pm -}$ | $71.9_{\pm -}$ | $18.86_{\pm -}$ | $18.82_{\pm -}$ | 6.87 |
| SVD | $83.89_{\pm 0.76}$ | $79.92_{\pm 0.56}$ | $92.87_{\pm 0.55}$ | $42.54_{\pm 0.22}$ | $59.73_{\pm 0.06}$ | $76.12_{\pm 0.26}$ | $62.04_{\pm 0.85}$ | $30.05_{\pm 0.13}$ | 4.45 |
| LAP | $84.98_{\pm 0.78}$ | $78.33_{\pm 0.84}$ | $91.11_{\pm 0.78}$ | $42.18_{\pm 0.14}$ | $59.05_{\pm 0.12}$ | $76.40_{\pm 0.20}$ | $60.66_{\pm 0.55}$ | $24.91_{\pm 0.02}$ | 6.62 |
| RAND | $88.85_{\pm 1.65}$ | $81.64_{\pm 1.48}$ | $91.75_{\pm 2.79}$ | $41.37_{\pm 1.26}$ | $59.03_{\pm 1.85}$ | $76.02_{\pm 1.34}$ | $61.22_{\pm 1.86}$ | $34.39_{\pm 1.08}$ | 5.12 |
| UNIFIED I/O | $89.85_{\pm 0.18}$ | $\mathbf{82.32}_{\pm 0.97}$ | $92.85_{\pm 0.48}$ | $42.58_{\pm 0.17}$ | $59.99_{\pm 0.05}$ | $76.44_{\pm 0.44}$ | $\mathbf{62.13}_{\pm 0.43}$ | $\mathbf{35.35}_{\pm 0.26}$ | 2.00 |
| 1-SHOT FOR TRAINING-FREE INFERENCE | | | | | | | | | |
| SUP | $36.80_{\pm 0.53}$ | $32.00_{\pm 0.98}$ | $53.44_{\pm 0.99}$ | $26.87_{\pm 0.75}$ | $54.83_{\pm 0.68}$ | $66.17_{\pm 0.73}$ | $30.18_{\pm 0.74}$ | $23.44_{\pm 0.59}$ | 6.50 |
| SSL | $55.67_{\pm 0.36}$ | $42.80_{\pm 0.25}$ | $77.86_{\pm 0.12}$ | $29.76_{\pm 0.12}$ | $57.11_{\pm 0.05}$ | $67.82_{\pm 1.02}$ | $25.07_{\pm 2.08}$ | $20.72_{\pm 0.89}$ | 4.62 |
| FUG | $27.26_{\pm 0.29}$ | $41.83_{\pm 1.03}$ | $67.70_{\pm 1.35}$ | $27.58_{\pm 0.48}$ | $49.93_{\pm 0.48}$ | $56.86_{\pm 0.93}$ | $23.39_{\pm 0.36}$ | $22.83_{\pm 0.17}$ | 7.25 |
| GRAPHANY | $\mathbf{62.87}_{\pm 0.31}$ | $\mathbf{53.63}_{\pm 0.49}$ | $80.81_{\pm 0.78}$ | $25.03_{\pm 0.58}$ | $49.65_{\pm 0.41}$ | $52.59_{\pm 0.38}$ | $28.51_{\pm 0.43}$ | $19.80_{\pm 0.54}$ | 5.50 |
| LABEL PROPAGATION | $46.26_{\pm 2.95}$ | $18.81_{\pm 1.55}$ | $22.98_{\pm 2.28}$ | $18.54_{\pm 0.01}$ | $53.00_{\pm 1.00}$ | $66.20_{\pm 2.92}$ | $18.86_{\pm 0.62}$ | $11.45_{\pm 0.69}$ | 8.12 |
| SVD | $55.42_{\pm 0.31}$ | $42.22_{\pm 0.83}$ | $76.31_{\pm 0.12}$ | $33.44_{\pm 0.25}$ | $56.94_{\pm 0.10}$ | $67.62_{\pm 0.49}$ | $31.14_{\pm 0.18}$ | $23.61_{\pm 0.06}$ | 4.15 |
| LAP | $55.53_{\pm 1.21}$ | $43.73_{\pm 0.43}$ | $78.41_{\pm 1.26}$ | $33.08_{\pm 0.35}$ | $57.00_{\pm 0.09}$ | $67.62_{\pm 0.58}$ | $29.82_{\pm 0.16}$ | $21.89_{\pm 0.03}$ | 4.15 |
| RAND | $58.80_{\pm 1.85}$ | $42.34_{\pm 1.42}$ | $76.05_{\pm 2.27}$ | $32.73_{\pm 1.03}$ | $57.72_{\pm 1.75}$ | $67.98_{\pm 2.07}$ | $30.75_{\pm 0.91}$ | $24.71_{\pm 0.74}$ | 3.37 |
| UNIFIED I/O | $59.89_{\pm 0.80}$ | $43.94_{\pm 0.50}$ | $\mathbf{85.13}_{\pm 0.68}$ | $\mathbf{33.47}_{\pm 0.14}$ | $\mathbf{57.90}_{\pm 0.15}$ | $\mathbf{68.51}_{\pm 0.13}$ | $\mathbf{32.00}_{\pm 0.11}$ | $\mathbf{25.69}_{\pm 0.12}$ | $\mathbf{1.25}$ |
| 3-SHOT FOR TRAINING-FREE INFERENCE | | | | | | | | | |
| SUP | $65.64_{\pm 0.30}$ | $37.10_{\pm 0.55}$ | $76.44_{\pm 0.48}$ | $27.80_{\pm 0.51}$ | $54.02_{\pm 0.62}$ | $68.89_{\pm 0.81}$ | $33.57_{\pm 0.41}$ | $20.88_{\pm 0.43}$ | 5.62 |
| SSL | $64.51_{\pm 2.68}$ | $48.73_{\pm 1.86}$ | $84.22_{\pm 0.12}$ | $26.53_{\pm 1.11}$ | $56.78_{\pm 0.16}$ | $59.24_{\pm 2.24}$ | $31.99_{\pm 0.37}$ | $20.42_{\pm 0.15}$ | 5.25 |
| FUG | $50.59_{\pm 1.43}$ | $47.77_{\pm 1.21}$ | $66.52_{\pm 2.47}$ | $24.02_{\pm 0.34}$ | $49.83_{\pm 0.18}$ | $57.93_{\pm 1.17}$ | $25.47_{\pm 0.56}$ | $20.35_{\pm 0.33}$ | 7.75 |
| GRAPHANY | $\mathbf{70.04}_{\pm 0.70}$ | $\mathbf{66.32}_{\pm 0.61}$ | $\mathbf{91.33}_{\pm 0.79}$ | $24.74_{\pm 0.80}$ | $54.71_{\pm 0.81}$ | $54.12_{\pm 0.63}$ | $33.69_{\pm 0.73}$ | $18.55_{\pm 0.72}$ | 4.37 |
| LABEL PROPAGATION | $60.04_{\pm 2.64}$ | $31.64_{\pm 2.22}$ | $32.82_{\pm 2.47}$ | $18.49_{\pm 0.06}$ | $52.95_{\pm 2.12}$ | $66.36_{\pm 3.47}$ | $16.45_{\pm 2.18}$ | $11.91_{\pm 0.73}$ | 8.12 |
| SVD | $59.86_{\pm 0.57}$ | $45.23_{\pm 0.90}$ | $76.88_{\pm 0.35}$ | $33.44_{\pm 0.30}$ | $56.93_{\pm 0.13}$ | $68.47_{\pm 0.34}$ | $33.64_{\pm 0.18}$ | $23.01_{\pm 0.04}$ | 4.81 |
| LAP | $56.58_{\pm 0.31}$ | $47.31_{\pm 0.45}$ | $76.88_{\pm 0.62}$ | $34.64_{\pm 0.38}$ | $56.75_{\pm 0.11}$ | $69.34_{\pm 0.35}$ | $32.21_{\pm 0.16}$ | $21.54_{\pm 0.02}$ | 4.93 |
| RAND | $66.11_{\pm 1.00}$ | $48.98_{\pm 1.57}$ | $79.31_{\pm 1.34}$ | $35.05_{\pm 1.07}$ | $56.96_{\pm 1.78}$ | $69.09_{\pm 1.09}$ | $33.68_{\pm 1.00}$ | $24.99_{\pm 0.74}$ | 2.75 |
| UNIFIED I/O | $68.33_{\pm 0.28}$ | $49.21_{\pm 0.91}$ | $84.68_{\pm 0.67}$ | $\mathbf{35.32}_{\pm 0.29}$ | $\mathbf{57.64}_{\pm 0.06}$ | $\mathbf{71.77}_{\pm 0.25}$ | $\mathbf{33.76}_{\pm 0.11}$ | $25.20_{\pm 0.11}$ | $\mathbf{1.37}$ |

datasets without additional training efforts, including Label Propagation (Kothari & Jain, 2002), GraphAny (Zhao et al., 2024a), and parameter-free feature alignment methods SVD-based (Sun et al., 2023), Laplacian-based (Sun et al., 2025), and random-based (Tang et al., 2024) input combined with our output module. We also include parameterized feature alignment method FUG (Zhao et al., 2024b), supervised baselines (GCN (Kipf & Welling, 2017), GraphSAGE (Hamilton et al., 2017), GAT (Veličković et al., 2018a)) and self-supervised baselines, including contrastive-based methods (DGI (Veličković et al., 2018b), GraphCL (You et al., 2020), GraphACL (Xiao et al., 2023), GRACE (Zhu et al., 2020), SimGRACE (Xia et al., 2022)) and reconstruction-based methods (MaskGAE (Li et al., 2023) and GraphMAE2 (Hou et al., 2023)). For FUG and self-supervised baselines, we follow GraphAny to solve the mapping matrix from the learned hidden representations to the labels in a training-free manner. The models are pre-trained on datasets from four distinct domains: amazon-ratings, ogbn-arxiv, Facebook, and roman-empire. The best results among the eight GNN backbones are reported for both our unified I/O and parameter-free feature alignment methods.

Results are summarized in Tab. 1. Compared to supervised baselines, pre-training with unified I/O achieves comparable or even superior performance on the original split. Under 1-shot and 3-shot settings, unified I/O consistently outperforms the supervised baselines. These results demonstrate that model pre-training is necessary for graph learning, particularly in data-scarce settings. Compared to other baselines, pre-trained GNNs with unified I/O obtain clear advantages on heterophilic datasets, whereas GraphAny performs better on homophilic datasets in certain cases. This difference reflects the higher transferability of node-feature knowledge, compared to structural knowledge learned by GNNs with unified I/O (see Appendix D.4 for details). Nevertheless, pure GNNs with unified I/O achieve the best average rank across different datasets. This indicates the superior ability of our unified I/O to support effective pre-training and downstream adaptation.

**Textual Datasets.** In comparison to the LM-based models, we adopt textual datasets (Chen et al., 2024c) for pre-training and inference. Models are pre-trained on PubMed, bookhistory, amazon-ratings, and arxiv with textual features. To ensure fair comparison, pure GNNs employ the second pseudo label assignment strategy in Sec 3.4. Tab. 2 presents comparison results with methods employing LM as I/O (OFA (Liu et al., 2023a), ZeroG (Li et al., 2024b), LLaGA (Chen et al., 2024a), GraphCLIP (Zhu et al., 2025), and RiemannGFM (Sun et al., 2025)), and parameter-free feature

Table 2: **Evaluation Results on Textual Datasets (Measured by accuracy: %).** Bold values denote the best results per test dataset. LP, Lap, and Rand denote Label Propagation, Laplacian decomposition, and random projection, respectively.

| | WIKICS | BOOKCHILD | COMPUTERS | PHOTO | SPORTSFIT | PRODUCTS | DBLP | TOLOKERS |
|---|---|---|---|---|---|---|---|---|
| LLAGA | $2.65_{\pm0.07}$ | $21.05_{\pm0.16}$ | $23.00_{\pm0.04}$ | $5.10_{\pm0.04}$ | $5.45_{\pm0.24}$ | $10.40_{\pm0.02}$ | $11.55_{\pm0.07}$ | $71.80_{\pm0.28}$ |
| ZEROG | $\mathbf{37.13}_{\pm0.41}$ | $12.62_{\pm2.25}$ | $6.72_{\pm0.52}$ | $3.84_{\pm0.17}$ | $30.04_{\pm0.20}$ | $\mathbf{24.79}_{\pm1.68}$ | $\mathbf{54.86}_{\pm0.52}$ | $78.13_{\pm0.01}$ |
| GRAPHCLIP | $3.54_{\pm0.51}$ | $16.16_{\pm0.01}$ | $25.43_{\pm0.16}$ | $4.19_{\pm0.01}$ | $7.61_{\pm0.10}$ | $9.31_{\pm0.08}$ | $34.94_{\pm0.37}$ | $78.01_{\pm0.07}$ |
| OFA | $33.89_{\pm0.09}$ | $1.98_{\pm0.01}$ | $5.98_{\pm0.10}$ | $14.72_{\pm0.01}$ | $12.65_{\pm0.01}$ | $2.64_{\pm0.02}$ | $51.01_{\pm0.02}$ | $78.16_{\pm0.01}$ |
| RIEMANNGFM | $4.26_{\pm0.11}$ | $1.74_{\pm0.06}$ | $5.77_{\pm0.01}$ | $6.93_{\pm0.49}$ | $3.83_{\pm0.02}$ | $2.45_{\pm0.05}$ | $38.68_{\pm0.37}$ | $77.65_{\pm0.01}$ |
| SVD | $30.24_{\pm0.37}$ | $16.16_{\pm0.34}$ | $27.65_{\pm0.43}$ | $49.15_{\pm0.98}$ | $41.93_{\pm1.04}$ | $13.04_{\pm0.20}$ | $46.91_{\pm0.60}$ | $78.50_{\pm0.52}$ |
| LAP | $23.43_{\pm0.24}$ | $30.47_{\pm0.82}$ | $27.05_{\pm0.38}$ | $49.09_{\pm0.64}$ | $42.05_{\pm1.43}$ | $13.05_{\pm0.28}$ | $43.63_{\pm0.39}$ | $78.47_{\pm0.58}$ |
| RAND | $23.41_{\pm0.36}$ | $30.32_{\pm1.01}$ | $29.82_{\pm0.38}$ | $49.04_{\pm1.22}$ | $41.83_{\pm1.13}$ | $13.42_{\pm0.78}$ | $47.06_{\pm0.77}$ | $78.16_{\pm0.83}$ |
| UNIFIED I/O | $32.63_{\pm0.21}$ | $\mathbf{31.60}_{\pm0.69}$ | $\mathbf{30.82}_{\pm0.26}$ | $\mathbf{49.20}_{\pm0.31}$ | $\mathbf{42.87}_{\pm1.03}$ | $15.93_{\pm0.32}$ | $52.04_{\pm0.86}$ | $\mathbf{78.66}_{\pm0.49}$ |

alignment methods. Pure GNNs with our unified I/O achieve superior performance to baselines except for ZeroG on WikiCS, products, and DBLP. Notably, baseline models with LM introduce knowledge gained from enormous training data for graph learning. In contrast, pure GNNs with only the knowledge of PubMed, bookhistory, amazon-ratings, and arxiv during pre-training achieve better performance. Compared to ZeroG, which is restricted to graphs with rich textual attributes, pure GNNs can be applied to either textual or non-textual datasets. This demonstrates the potential of pre-training with pure GNNs in tackling various graph tasks.

## 4.2 PRE-TRAINING CONDITION STUDY

**Pre-training with Scaling Parameters.** Pure GNNs are pre-trained on amazon-ratings, ogbn-arxiv, Facebook, and roman-empire, with scaling parameters. To mitigate the impact of the domain gap, one dataset is selected from each domain, encompassing both homophilic and heterophilic graphs. Performance results are averaged across test graphs (Tab. S11). Fig. 5(a) presents the impact of hidden dimensionality $d$, showing that larger values of $d$ consistently improve performance across all GNN architectures. In contrast, the impact of the number of layers $L$ varies by architecture. As shown in Fig.5(b), deeper network configurations consistently enhance performance for GCN, GAT, GraphSAGE, and MixHop, whereas GIN, DeepGCN, and GraphGPS experience performance degradation with more layers. For $N^2$, performance initially improves with more layers but eventually declines when $L$ exceeds 8. In Appendix D.2, we examine larger parameter configurations for GCN and GIN. The results show that increasing the number of layers leads to over-parameterization, whereas enlarging the hidden dimension does not. All these results suggest that scaling up hidden dimensionality is the prior strategy for enhancing pre-trained GNN models, while the configuration of the number of layers depends on the specific GNN architectures.

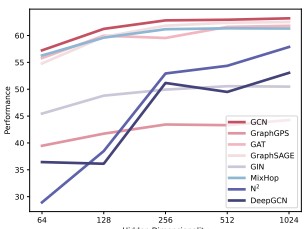

(a) Hidden Dimensions

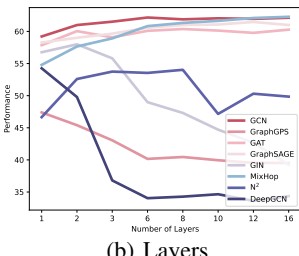

(b) Layers

Figure 5: **Pre-training with Scaling Parameters.**

**Pre-training with Scaling Data.** To assess the effect of data scaling in GNN pre-training, we consider two strategies: (1) increasing the number of training datasets (one, three, and four combined), and (2) enlarging individual datasets (1k, 10k, and 100k nodes per dataset). Detailed combinations of the training and test datasets are provided in Appendix Tab. S11, S12. Due to the varying difficulty of specific downstream tasks, model performance cannot be directly compared across different test datasets. To address this problem, training datasets are split into three groups with an average number of nodes around 1k, 10k, and 100k. Model performance is then normalized with the group average performance. For more analysis on the data scaling strategies, please refer to Appendix D.3. The averaged results over 1,848 data points across different backbones and test datasets are presented in Fig. 6. We can see that increasing either the minimum dataset sizes or increasing training datasets with the same node scale improves the model performance. This suggests that scaling up training data enhances the adaptation ability of the pre-trained models to test datasets.

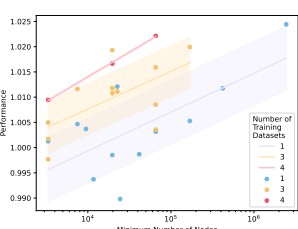

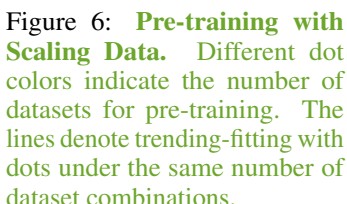

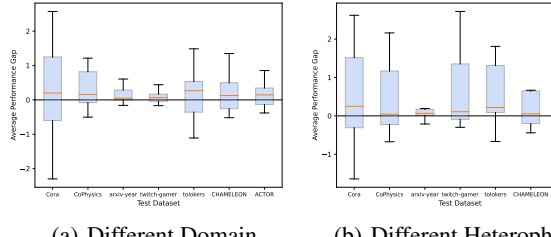

(a) Different Domain      (b) Different Heterophily

Figure 6: **Pre-training with Scaling Data.** Different dot colors indicate the number of datasets for pre-training. The lines denote trending-fitting with dots under the same number of dataset combinations.

Figure 7: **Adaptation Results with Different Domain Gap.** The cross-domain performance gap is calculated as the performance difference between GNNs pre-trained on the same-domain group as the test datasets and those on the different-domain group. The heterophily performance gap is computed by subtracting the same homophily–heterophily training type as the test datasets with that of the opposite type.

**Inference with Different Domain Gap.** We further evaluate pre-trained GNNs on adaptation tasks: (1) pre-training in one domain and adapting to the others, (2) pre-training across three domains and adapting to the remaining one, and (3) transferring across datasets with varying homophily–heterophily. Dataset combinations are listed in Tab. S11, S12.

The adaptation results for tasks (1) and (2) are presented in Fig. 7(a). For each test dataset, the training datasets are categorized into two groups: those collected from the same domain as the test dataset and those from different domains. The performance gap is computed by subtracting the performance of GNNs pre-trained on the different-domain group from that of the same-domain group ($\texttt{Metric}_{\texttt{same}} - \texttt{Metric}_{\texttt{diff}}$). To avoid the influence of the dataset scales, the model performance is first grouped based on the average number of nodes per training dataset (1k, 10k, and 100k), and then averaged and compared within each group. As shown in Fig. 7(a), pre-trained GNNs generally exhibit positive performance gaps between the same-domain and different-domain groups across diverse test datasets. This observation indicates that aligning the training and test domains tends to improve model performance during inference.

The adaptation results for task (3) with different homophily-heterophily are presented in Fig. 7(b). When comparing on the same test dataset, the training datasets are categorized into two groups: those with the same homophily-heterophily as the test dataset and those with the opposite. The performance gap is the performance difference between GNNs pre-trained on the same-homophily-heterophily group and that of the opposite-homophily-heterophily group ($\texttt{Metric}_{\texttt{same}} - \texttt{Metric}_{\texttt{oppo}}$). Since training datasets of certain scales are missing for heterophily/homophily graphs in certain domains, our experimental analysis for task (3) directly averages results by mixing all training scales together. Fig. 7(b) shows that models pre-trained on datasets with the same homophily-heterophily as the test datasets tend to achieve better performance during inference. This suggests that one may construct a training dataset based on the homophily-heterophily of downstream tasks to gain better results. We also compared adaptation results with the same pre-training datasets in Appendix D.4. Results in Fig. 11(a) show that models pre-trained on either homophilic or heterophilic graphs gain better results on heterophilic graphs than homophilic graphs. This can be attributed to the better transferability of the node feature knowledge learned by our unified I/O than the structural knowledge (Fig. 11(d)), where node feature knowledge better benefits the adaptation to heterophilic graphs (Fig. 11(b)) and structural knowledge benefits homophilic tasks (Fig. 11(c)). Please refer to Appendix D.4 for more details.

### 4.3 PRE-TRAINED PURE GNNs PROVIDE COMPETITIVE GRAPH OPERATORS

Our unified I/O modules enable seamless adaptation of pure GNN architectures across diverse datasets. To further evaluate the effectiveness of the pre-trained GNN operators, we fine-tune the models pre-trained on amazon-ratings, ogbn-arxiv, Facebook, and roman-empire. The internal GNN module $\texttt{F}_{\texttt{g}}$ within the pre-trained models is frozen during the fine-tuning. The unification-oriented I/O function

Table 3: **Evaluation Results of Fine-tuning the I/O Modules in the Pre-trained GNNs (Measured by accuracy except ROC AUC for tolokers: %).** Bold values denote the best results per test dataset.

| | CORA | PUBMED | AMAZON COMPUTERS | WIKICS | AMAZON -RATINGS | MINESWEEPER | TOLOKERS |
|---|---|---|---|---|---|---|---|
| | SELF-SUPERVISED LEARNING AND SUPERVISED OUTPUT LEARNING | | | | | | |
| DGI | $84.00_{\pm0.28}$ | $83.73_{\pm0.41}$ | $82.11_{\pm0.16}$ | $75.05_{\pm0.39}$ | $40.80_{\pm0.74}$ | $88.45_{\pm0.38}$ | $77.73_{\pm0.14}$ |
| GRACE | $84.30_{\pm0.57}$ | $85.81_{\pm0.18}$ | $89.67_{\pm0.36}$ | $75.80_{\pm0.56}$ | $42.19_{\pm0.12}$ | $86.15_{\pm0.45}$ | $75.06_{\pm0.14}$ |
| GRAPHACL | $75.00_{\pm0.75}$ | $82.93_{\pm0.17}$ | $80.58_{\pm0.15}$ | $68.00_{\pm0.75}$ | $40.65_{\pm0.14}$ | $87.23_{\pm0.11}$ | $77.68_{\pm0.25}$ |
| GRAPHCL | $63.33_{\pm1.76}$ | $63.53_{\pm1.08}$ | $84.57_{\pm0.34}$ | $76.32_{\pm0.18}$ | $42.35_{\pm0.37}$ | $79.85_{\pm0.12}$ | $80.03_{\pm0.12}$ |
| MASKGAE | $75.13_{\pm1.78}$ | $75.27_{\pm1.03}$ | $92.15_{\pm0.05}$ | $78.25_{\pm0.20}$ | $43.54_{\pm0.30}$ | $84.33_{\pm0.16}$ | $81.13_{\pm0.34}$ |
| SIMGRACE | $67.07_{\pm0.82}$ | $77.63_{\pm0.82}$ | $87.44_{\pm0.18}$ | $78.75_{\pm0.28}$ | $43.46_{\pm0.23}$ | $84.23_{\pm0.16}$ | $80.29_{\pm0.30}$ |
| GRAPHMAE2 | $79.50_{\pm0.51}$ | $67.07_{\pm0.91}$ | $91.03_{\pm0.19}$ | $76.24_{\pm0.11}$ | $40.95_{\pm0.71}$ | $80.16_{\pm0.10}$ | $80.17_{\pm0.07}$ |
| | SUPERVISED LEARNING | | | | | | |
| GRAPHSAGE | $78.83_{\pm0.50}$ | $88.11_{\pm0.05}$ | $91.09_{\pm0.02}$ | $78.13_{\pm0.15}$ | $45.71_{\pm0.38}$ | $90.55_{\pm0.10}$ | $83.06_{\pm0.59}$ |
| GAT | $77.51_{\pm2.35}$ | $85.30_{\pm0.15}$ | $89.78_{\pm0.02}$ | $76.35_{\pm0.80}$ | $44.54_{\pm0.52}$ | $82.07_{\pm1.17}$ | $77.37_{\pm0.28}$ |
| GIN | $77.36_{\pm0.15}$ | $85.13_{\pm0.55}$ | $90.51_{\pm0.80}$ | $74.02_{\pm0.62}$ | $46.33_{\pm0.11}$ | $74.93_{\pm0.58}$ | $60.93_{\pm2.25}$ |
| GCN | $80.35_{\pm0.25}$ | $85.44_{\pm0.50}$ | $90.66_{\pm0.13}$ | $78.55_{\pm0.01}$ | $46.71_{\pm0.25}$ | $76.43_{\pm1.05}$ | $77.79_{\pm0.12}$ |
| GRAPHGPS | $58.61_{\pm0.05}$ | $85.21_{\pm0.30}$ | $88.87_{\pm0.20}$ | $75.18_{\pm0.04}$ | $47.85_{\pm0.29}$ | $89.64_{\pm0.24}$ | $79.82_{\pm0.06}$ |
| | PRE-TRAINING AND I/O FINE-TUNING | | | | | | |
| UNIFIED I/O | $\mathbf{84.63}_{\pm0.12}$ | $\mathbf{88.91}_{\pm0.45}$ | $\mathbf{92.33}_{\pm0.25}$ | $\mathbf{78.98}_{\pm0.12}$ | $\mathbf{51.59}_{\pm0.05}$ | $\mathbf{91.39}_{\pm0.15}$ | $\mathbf{83.29}_{\pm0.13}$ |

$f_{in}(\cdot)$ in Eq. 4 and $f_{out}(\cdot)$ in Eq. 5 are replaced with learnable parameters. The best fine-tuning results among different backbones on downstream graphs are summarized in Tab. 3. For full results, please refer to Appendix D.5. The pre-trained operators achieve superior performance compared to supervised methods and self-supervised methods (DGI (Veličković et al., 2018b), GRACE (Zhu et al., 2020), GraphACL (Xiao et al., 2023), GraphCL (You et al., 2020), SimGRACE (Xia et al., 2022), MaskGAE (Li et al., 2023), GraphMAE2 (Hou et al., 2023)). Notably, the pre-trained operators require minimal hyperparameter tuning, with only dropout adjusted during fine-tuning. This significantly simplifies the hyperparameter tuning process, enabling efficient adaptation of pre-trained GNNs to various graphs with promising performance.

### 4.4 COST ANALYSIS ON THE UNIFIED I/O

Both the space complexity and time complexity of our unified I/O are $O(n)$, with $d_{in}, d, s, c \ll n$. Empirical time consumption and information loss results are provided in Appendix D.6 and D.7. Results show that unified I/O maintains a reasonable time cost under various scales of graphs and numbers of input features, and does not cause severe information loss. This demonstrates the effectiveness of our unified I/O in learning input and output mappings.

## 5 CONCLUSION

In this paper, we achieved unified input and output for graphs, enabling pre-training with pure GNNs across diverse datasets. To decouple learnable parameters from the number and semantics of input features and output labels, our unified I/O modules employ a shared relation function for the feature semantics and uniformly sampled points for the label semantics. By integrating our unified I/O modules with various GNN architectures, we demonstrated that pure GNNs can serve as effective graph learners for direct adaptation to downstream tasks and provide competitive pre-trained graph operators. For the usage of LLM and the limitation discussion, please refer to Appendix F and G.

### REPRODUCIBILITY STATEMENT

We have made efforts to ensure the reproducibility of our work. Specifically, we provide a detailed description of the experimental setups in Appendix C, including evaluation settings, dataset information, architecture configurations, and hyperparameter setups. All datasets employed are publicly available. In addition, the code implementation of our proposed methods is provided in the supplementary material. The complete source code will be released publicly upon acceptance of the paper.

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

# A  GENERAL TARGET-INSENSITIVE OUTPUT MODULE

Sec. 3.4 takes node classification as an example to formulate our unified output module. In practice, our method can be applied to general classification and regression tasks. Specifically, the general output module in Eq. 5 can be formulated as

$$\hat{\mathbf{C}} = \mathtt{F}_{\mathtt{out}}\left(\mathbf{H}^{(L)}\right) = \sigma\left(\mathbf{g}(\mathbf{H}^{(L)})\mathbf{S}_{\mathtt{src}}^{(\mathtt{out})}\mathtt{f}_{\mathtt{out}}(c,s)^{\top}\right), \tag{S7}$$

where $\mathbf{g}$ denotes the read-out function for different levels of tasks. For classification tasks, the nonlinear function $\sigma$ can be implemented as `softmax`, `sigmoid`, or `tanh`, and the loss function in Eq. 6 remains a similar form. For regression tasks, $\sigma$ can be omitted and the loss function only contains the inner-class loss with node sets of each class constituted with a single node.

# B  RELATED WORK

## B.1  I/O UNIFICATION FOR TRAINING-FREE ADAPTATION

**Unified with Language Models.** One of the key challenges in developing pre-trained graph foundation models lies in the diverse features and labels. Inspired by the remarkable success of the pre-trained LLMs (OpenAI et al., 2024), researchers have proposed to adopt LMs as I/O modules for graphs (Liu et al., 2023a; Kong et al., 2024; Tang et al., 2024; Zhu et al., 2025). Chen et al. (2024b) investigates the potential of LMs as enhancers for input and predictors for output, separately. ZeroG (Li et al., 2024b) further integrates the two approaches by employing an LM to encode textual descriptions of nodes and classes into unified embeddings. While ZeroG achieves unified input and output for graph pre-training, its applicability is limited to text-attributed graphs (TAGs, *i.e.*, graphs with rich textual features). OFA (Liu et al., 2023a) generalizes TAGs by introducing templates to convert numerical node and edge features into textual descriptions, thus extending its applicability to general graphs. In addition to integrating LMs with GNNs, researchers have also explored pure LLMs in addressing graph-related tasks (Wang et al., 2023; Chen et al., 2024a).

**Unified with Specific Design.** Distinct from the aforementioned approaches, our work aims to pre-train purely GNN-based models and explore their potential for training-free adaptation. Related efforts in this area include the use of singular value decomposition (SVD) (Sun et al., 2023), Laplacian decomposition (Sun et al., 2025), random projection (Tang et al., 2024), adversarial reprogramming attacks (Jing et al., 2023), and parametric principal component analysis (PCA) (Zhao et al., 2024b)to align different numbers of features. However, these methods couple the parameter values of the input module with specific inputs or fail to unify label spaces for GNNs. A recent approach, GraphAny, attempts to address this challenge by solving the pseudo-inverse of the transformation weight matrix in a linear GNN (Zhao et al., 2024a). Despite the unified input and output for different datasets, GraphAny requires observed labels from test datasets to compute the weight matrix, and is constrained to node-level tasks and linear GNN architectures (Wu et al., 2019). As a result, pre-training graph models with general pure GNN architectures remains an open problem.

## B.2  MODEL FINE-TUNING FOR FEW-SHOT ADAPTATION

In addition to I/O unification, model fine-tuning has been extensively explored to adapt pre-trained graph models to diverse features and labels (Yu et al., 2024). Approaches like GCC (Qiu et al., 2020) employ full fine-tuning on pre-trained models, which is resource-intensive and prone to overfitting, particularly when downstream datasets involve limited labeled data. To address these limitations, researchers have proposed parameter-efficient graph fine-tuning methods, such as graph adapters (Li et al., 2024a; Gui et al., 2024) and graph prompts (Sun et al., 2023). Graph adapters incorporate additional tunable modules for GNNs, effectively bridging the gap between pre-training and inference domains. Alternatively, graph prompt learning introduces input-specific prompts to modify node features (Sun et al., 2022; Fang et al., 2023; Liu et al., 2023b) or graph structures (Sun et al., 2023; Huang et al., 2023; Zhang et al., 2023b; Tan et al., 2023; Yu et al., 2025), enabling better adaptation to various datasets. Different from model fine-tuning, this paper focuses on directly unifying diverse graph features and labels at the pre-training stage. By exploring this foundational problem, our findings show that the pre-trained GNN models can provide competitive graph operators for further fine-tuning and simplify the hyperparameter tuning process (Sec 4.3).

Table S4: **Original Dataset Split for Training-free Inference.**

|                  | TRAIN  | TEST   | TOTAL   |
|------------------|--------|--------|---------|
| AMAZONCOMPUTERS  | 8,252  | 2,750  | 13,381  |
| CORA             | 1,208  | 1,000  | 2,708   |
| COAUTHORPHYSICS  | 20,697 | 6,898  | 34,493  |
| ARXIV-YEAR       | 84,671 | 42,337 | 169,343 |
| TWITCH-GAMER     | 84,057 | 42,029 | 168,114 |
| TOLOKERS         | 5,879  | 2,940  | 11,758  |
| CHAMELEON        | 1,092  | 456    | 2,277   |
| ACTOR            | 3,698  | 1,520  | 7,600   |

Table S5: **Dataset Split for Model Fine-tuning.**

|                  | TRAIN  | VALID | TEST  | TOTAL  |
|------------------|--------|-------|-------|--------|
| CORA             | 1,208  | 500   | 1,000 | 2,708  |
| PUBMED           | 18,217 | 500   | 1,000 | 1,9717 |
| AMAZONCOMPUTERS  | 8,252  | 2,379 | 2,750 | 1,3381 |
| WIKICS           | 580    | 5,274 | 5,847 | 1,1701 |
| AMAZON-RATINGS   | 12,246 | 6,123 | 6,123 | 2,4492 |
| MINESWEEPER      | 5,000  | 2,500 | 2,500 | 1,0000 |
| TOLOKERS         | 5,879  | 2,939 | 2,940 | 1,1758 |

## C EXPERIMENTAL SETUP

### C.1 SETTINGS

Zero-shot learning with label priors requires test label knowledge. However, label semantics vary across graph datasets, making full coverage of different semantics during pre-training impractical. To align with real-world scenarios, we exclude label knowledge from pre-training. Models do not have access to the label description knowledge during training and target-insensitive prediction. For further evaluation, the prediction results are permuted by aligning pseudo labels with actual labels. Specifically, to ensure fair comparison with LM-based graph models in the zero-shot setting, we employ the embedded label semantics $\mathbf{S}_{\text{tgt}}^{\text{LM}}$ from language models (Li et al., 2024b) and construct the assignment matrix with $\mathbf{f}_{\text{out}}(c, s)\mathbf{S}_{\text{tgt}}^{\text{LM}\top}$. This scenario does not require any labeled samples from test datasets. To explore the potential of pre-trained pure GNNs on general downstream tasks, observed target labels are only included during inference to match pseudo labels with real labels. Given the observed real labels $\mathbf{C}$, the mapping relations between the pseudo labels and the real labels can be formulated as $\hat{\mathbf{C}}\mathbf{P}(c\mathbf{I} - \mathbf{1}\mathbf{1}^\top) = c\log(c\mathbf{C})$. The assignment matrix $\mathbf{P}$ can be solved as the least-squares solution of the linear equation without additional training effort.

### C.2 DATASETS

Various real-world datasets are adopted for model pre-training and evaluation. These datasets can be categorized into four domains, including electronic-commerce graphs (e-com.), citation graphs, social graphs, and Wikipedia graphs (wiki). Three levels of average node number per dataset including 1k, 10k, and 100k are incorporated for each domain. The statistics of these datasets are summarized in Tab. S11 and Tab. S12. For the test datasets, we follow the standard split as the supervised learning setting in the original paper.

The e-com. domain consists of one test dataset AmazonComputers (Shchur et al., 2019), and three train datasets AmazonPhoto (Shchur et al., 2019), amazon-ratings (Platonov et al., 2023), and ogbn-products (Hu et al., 2020). All these datasets are collected from Amazon. The citation domain consists of three test datasets Cora (Sen et al., 2008), CoauthorPhysics (Shchur et al., 2019), and arxiv-year (Lim et al., 2021), and four train datasets CiteSeer (Sen et al., 2008), PubMed (Sen et al., 2008), ogbn-arxiv (Hu et al., 2020), and snap-patents (Lim et al., 2021). These datasets are collected from academic graphs, encoding coauthorship and citation relations. The social domain consists of two test datasets twitch-gamer (Lim et al., 2021), tolokers (Platonov et al., 2023), and five train datasets twitch-e (Lim et al., 2021), fb100 (Lim et al., 2021), genius (Lim et al., 2021),

Table S6: **Textual Dataset Statistics.**

|  | DATASET | USAGE | #NODES | #EDGES | #LABELS |
|---|---|---|---|---|---|
| | ARXIV | TRAIN | 169,343 | 2,315,598 | 40 |
| CITATION | PUBMED | TRAIN | 19,717 | 88,648 | 3 |
| | DBLP | TEST | 14,376 | 431,326 | 4 |
| | AMAZON-RATINGS | TRAIN | 24,492 | 186,100 | 5 |
| | BOOKHISTORY | TRAIN | 41,551 | 503,180 | 12 |
| | BOOKCHILD | TEST | 76,875 | 2,325,044 | 24 |
| E-COM. | COMPUTERS | TEST | 87,229 | 1,256,548 | 10 |
| | PHOTO | TEST | 48,362 | 873,782 | 12 |
| | SPORTSFIT | TEST | 173,055 | 3,020,134 | 13 |
| | PRODUCTS | TEST | 316,513 | 19,337,722 | 39 |
| WIKI. | WIKICS | TEST | 11,701 | 431,726 | 10 |
| SOCIAL | TOLOKERS | TEST | 11,758 | 1,038,000 | 2 |

Facebook (Rozemberczki et al., 2021), and pokec (Lim et al., 2021). These datasets are collected from online social media, encoding social relationships between different users. Specifically, tolokers encapsulates crowdsourcing participation data sourced from the Toloka platform. Edges in tolokers link toloker pairs that have completed the same tasks. This graph indicates certain interests of the participants. Therefore, we classify it in the social domain. The wiki. domain consists of two test datasets CHAMELEON (Pei et al., 2019), ACTOR (Pei et al., 2019), and three train datasets WikiCS (Mernyei & Cangea, 2022), romain-empire (Platonov et al., 2023), and NELL (Carlson et al., 2010). These datasets are collected from Wikipedia. The splits employed in Tab. 1 and Tab. 3 are summarized in Tab. S4 and Tab. S5, respectively.

We also employ textual datasets (Chen et al., 2024c) to compare with LM-based models. The datasets can also be categorized into citation, e-com., social, and Wiki graphs. The citation graphs include arxiv, PubMed, and DBLP. The e-com. graphs include amazon-ratings, bookhistory, bookchild, computers, photo, sportsfit, and products. The Wiki. graph refers to WikiCS, and the social graph refers to tolokers. The statistics of these datasets are summarized in Tab. S6.

## C.3 IMPLEMENTATION

Eight GNN methods are employed as the backbone for model pre-training, including GCN (Kipf & Welling, 2017), GAT (Veličković et al., 2018a), GraphSAGE (Hamilton et al., 2017), GIN (Xu et al., 2019), MixHop (Abu-El-Haija et al., 2019), GraphGPS (Rampášek et al., 2022), DeepGCN (Li et al., 2019), and $N^2$ (Sun et al., 2024). The evaluations are conducted on a single NVIDIA GeForce RTX 4090 or a single NVIDIA A100. Models are pre-trained on the TRAIN datasets and evaluated on the TEST datasets in Tab. S11.

Except for $N^2$, backbones are implemented with the framework of PyTorch Geometric. The pre-training process is conducted for 5000 epochs with a learning rate fixed at $1e-5$. The supervised result reproducing is conducted for 500 epochs and will be early stopped if there is no further reduction in the validation loss during 200 epochs. The total epoch for model fine-tuning is 1000, with early-stopping for 200 epochs. We adopt Adam (Kingma & Ba, 2015) as optimizer and set weight decay as $1 \times 10^{-6}$. The supervised results are reproduced under the same architecture, with grid search performed on the number of layers in $\{2, 3, 5, 10\}$, dropout in $\{0., 0.1, 0.2, 0.3, 0.5\}$, and the number of hidden dimensions in $\{64, 128, 256\}$. The self-supervised models are first pre-trained in the self-supervised setting and then frozen with a trainable linear output for supervised learning. The hyperparameter configuration follows the original implementation. For model fine-tuning, we perform grid search on dropout in $\{0., 0.1, 0.2, 0.3, 0.5\}$ based on the validation results. For the information loss study in Appendix D.7, grid search is performed on the number of layers in $\{2, 3, 5, 10\}$, and the number of hidden dimensions in $\{64, 128\}$. Except for performance comparison with baselines on textual/non-textual datasets and pre-training with scaling parameters, we fix the number of dimensions at 256 for all backbones. The configuration for the number of layers during pre-training for data scaling and domain gap is presented in Tab. S7.

Table S7: **Layer Configuration.**

|  | #LAYERS |
|---|---|
| GCN | 6 |
| GAT | 6 |
| GRAPHSAGE | 6 |
| GIN | 2 |
| DEEPGCN | 1 |
| $N^2$ | 6 |
| GRAPHGPS | 1 |
| MIXHOP | 8 |

Table S8: **Evaluation Results with Graph-level Datasets (Measured by accuracy: %).** Bold values denote the best results per test dataset. LP, Lap, and Rand denote Label Propagation, Laplacian decomposition, and random projection, respectively.

| | ENZYMES | REDDIT-BINARY | PROTEINS | AMAZONCOMPUTERS | CORA | ARXIV-YEAR | TWITCH-GAMER |
|---|---|---|---|---|---|---|---|
| | | | ORIGINAL SPLIT | | | | |
| GCN (SUP) | $37.41_{\pm0.32}$ | $\mathbf{78.40}_{\pm0.41}$ | $71.70_{\pm0.39}$ | $91.09_{\pm0.13}$ | $81.80_{\pm0.28}$ | $48.03_{\pm0.41}$ | $59.44_{\pm0.18}$ |
| GRAPHCL | $19.77_{\pm0.41}$ | $71.26_{\pm0.32}$ | $67.23_{\pm1.01}$ | $88.67_{\pm0.48}$ | $61.93_{\pm1.47}$ | OOM | OOM |
| LAP | $52.38_{\pm0.37}$ | $72.04_{\pm0.71}$ | $70.07_{\pm0.52}$ | $78.69_{\pm0.76}$ | $76.80_{\pm0.46}$ | $40.86_{\pm0.64}$ | $56.25_{\pm0.50}$ |
| RAND | $18.33_{\pm0.37}$ | $52.20_{\pm0.43}$ | $59.67_{\pm0.73}$ | $34.73_{\pm0.68}$ | $15.50_{\pm0.71}$ | $37.00_{\pm0.76}$ | $52.29_{\pm0.44}$ |
| SVD | $53.65_{\pm0.54}$ | $71.08_{\pm0.54}$ | $70.89_{\pm0.69}$ | $75.67_{\pm0.49}$ | $73.50_{\pm0.66}$ | $40.57_{\pm0.55}$ | $56.82_{\pm0.52}$ |
| FUG | - | - | - | $88.22_{\pm0.09}$ | $30.70_{\pm0.96}$ | $42.54_{\pm0.30}$ | $58.16_{\pm0.27}$ |
| GRAPHANY | - | - | - | $82.94_{\pm0.82}$ | $79.41_{\pm0.35}$ | $38.36_{\pm0.53}$ | $59.96_{\pm0.02}$ |
| UNIFIED I/O (NODE) | $52.83_{\pm1.06}$ | $69.65_{\pm0.36}$ | $70.73_{\pm0.26}$ | $89.85_{\pm0.18}$ | $82.32_{\pm0.97}$ | $42.58_{\pm0.17}$ | $59.99_{\pm0.05}$ |
| UNIFIED I/O (GRAPH) | $\mathbf{54.88}_{\pm0.40}$ | $73.09_{\pm0.43}$ | $\mathbf{72.45}_{\pm0.44}$ | $80.29_{\pm0.55}$ | $80.30_{\pm0.70}$ | $41.35_{\pm0.43}$ | $57.73_{\pm0.66}$ |
| | | | 1-SHOT FOR TRAINING-FREE INFERENCE | | | | |
| GRAPHCL | $16.71_{\pm0.69}$ | $52.85_{\pm1.53}$ | $51.89_{\pm3.68}$ | $41.39_{\pm2.64}$ | $27.83_{\pm1.20}$ | OOM | OOM |
| LAP | $17.33_{\pm2.46}$ | $56.88_{\pm1.66}$ | $55.62_{\pm1.91}$ | $45.71_{\pm2.18}$ | $22.80_{\pm2.00}$ | $28.57_{\pm1.90}$ | $53.45_{\pm1.88}$ |
| RAND | $17.68_{\pm2.06}$ | $55.36_{\pm1.86}$ | $49.56_{\pm1.56}$ | $15.67_{\pm1.78}$ | $16.20_{\pm2.26}$ | $26.61_{\pm1.93}$ | $52.56_{\pm2.16}$ |
| SVD | $17.87_{\pm1.37}$ | $56.80_{\pm1.35}$ | $55.47_{\pm1.60}$ | $43.46_{\pm1.27}$ | $18.90_{\pm2.44}$ | $24.78_{\pm1.37}$ | $53.45_{\pm1.17}$ |
| FUG | - | - | - | $27.26_{\pm0.36}$ | $41.83_{\pm0.26}$ | $27.58_{\pm0.11}$ | $49.93_{\pm0.01}$ |
| GRAPHANY | - | - | - | $62.87_{\pm0.29}$ | $53.63_{\pm1.03}$ | $25.03_{\pm0.48}$ | $49.65_{\pm0.48}$ |
| UNIFIED I/O (NODE) | $17.82_{\pm1.27}$ | $48.21_{\pm1.34}$ | $52.27_{\pm1.75}$ | $59.89_{\pm0.80}$ | $43.94_{\pm0.50}$ | $33.47_{\pm0.14}$ | $57.90_{\pm0.15}$ |
| UNIFIED I/O (GRAPH) | $\mathbf{18.08}_{\pm1.82}$ | $\mathbf{58.89}_{\pm2.40}$ | $\mathbf{56.07}_{\pm0.81}$ | $60.33_{\pm2.16}$ | $25.90_{\pm0.92}$ | $29.96_{\pm0.95}$ | $54.60_{\pm1.47}$ |
| | | | 3-SHOT FOR TRAINING-FREE INFERENCE | | | | |
| GRAPHCL | $18.16_{\pm0.88}$ | $57.13_{\pm1.46}$ | $53.73_{\pm0.82}$ | $55.41_{\pm1.41}$ | $34.97_{\pm0.64}$ | OOM | OOM |
| LAP | $20.42_{\pm1.29}$ | $58.23_{\pm1.23}$ | $58.60_{\pm1.26}$ | $45.35_{\pm1.10}$ | $34.80_{\pm1.39}$ | $24.62_{\pm1.35}$ | $54.51_{\pm1.26}$ |
| RAND | $18.62_{\pm1.28}$ | $56.30_{\pm1.16}$ | $51.38_{\pm1.16}$ | $19.02_{\pm1.31}$ | $30.70_{\pm1.37}$ | $25.64_{\pm1.09}$ | $53.04_{\pm1.43}$ |
| SVD | $20.67_{\pm1.10}$ | $58.28_{\pm1.16}$ | $55.95_{\pm1.35}$ | $50.76_{\pm1.36}$ | $35.60_{\pm1.26}$ | $21.62_{\pm1.23}$ | $53.26_{\pm1.13}$ |
| FUG | - | - | - | $50.59_{\pm0.29}$ | $47.77_{\pm0.29}$ | $24.02_{\pm0.19}$ | $49.83_{\pm0.09}$ |
| GRAPHANY | - | - | - | $70.04_{\pm1.43}$ | $66.32_{\pm1.21}$ | $24.74_{\pm0.34}$ | $54.71_{\pm0.18}$ |
| UNIFIED I/O (NODE) | $19.00_{\pm1.93}$ | $49.07_{\pm1.68}$ | $54.46_{\pm1.36}$ | $68.33_{\pm0.28}$ | $49.21_{\pm0.91}$ | $35.32_{\pm0.29}$ | $57.64_{\pm0.06}$ |
| UNIFIED I/O (GRAPH) | $\mathbf{22.05}_{\pm2.00}$ | $\mathbf{59.32}_{\pm0.93}$ | $\mathbf{60.46}_{\pm1.57}$ | $65.78_{\pm2.27}$ | $38.90_{\pm1.17}$ | $23.97_{\pm1.48}$ | $55.56_{\pm1.43}$ |

# D ADDITIONAL RESULTS

## D.1 GRAPH-LEVEL TASKS

As noted in Appendix A, except for the node-level evaluation, unified I/O can be employed for more tasks. To demonstrate this, GNNs are pre-trained on graph-level datasets (COLLAB, IMDB-BINARY, MUTAG, and D&D) (Morris et al., 2020), and evaluated on graph-level tasks (PROTEINS, REDDIT-BINARY, ENZYMES) (Morris et al., 2020), and node-level tasks (AmazonComputers (Shchur et al., 2019), Cora (Watts & Strogatz, 1998), arxiv-year, twitch-gamer (Lim et al., 2021)). Baselines include supervised method GCN (Kipf & Welling, 2017), self-supervised learning method GraphCL (You et al., 2020), parameter-free feature alignment methods SVD (Sun et al., 2023), Laplacian projection (Sun et al., 2025), and random projection (Tang et al., 2024)) combined with our unified output module. We also adopt FUG (Zhao et al., 2024b) and GraphAny (Zhao et al., 2024a) as baselines for node-level tasks. Neither methods support graph tasks and are thus pre-trained on (amazon-ratings, ogbn-arxiv, Facebook, and roman-empire).

Results in Tab. S8 show that pre-training with unified I/O surpasses GraphCL and parameter-free feature alignment methods on both graph-level and node-level tasks. On graph-level tasks, unified I/O also delivers performance comparable to, or better than, supervised GCN. However, transferring between graph-level and node-level tasks introduces a noticeable performance gap in both directions. In particular, for node-level downstream tasks, models pre-trained on graph-level datasets perform competitively on the original split but lag behind under the 1-shot and 3-shot settings when compared with models pre-trained directly on node-level datasets.

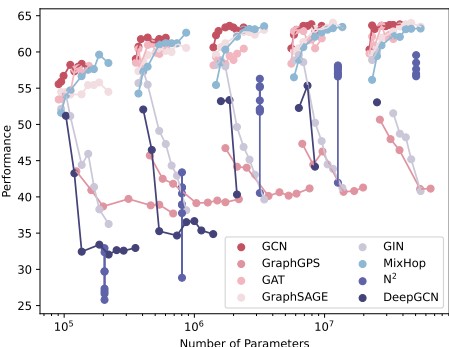

Figure S8: **Pre-training with Scaling Parameters.** The connected dots denote scaling the number of parameters by stacking multiple layers. The adjacent line segments denote scaling by expanding the hidden dimension.

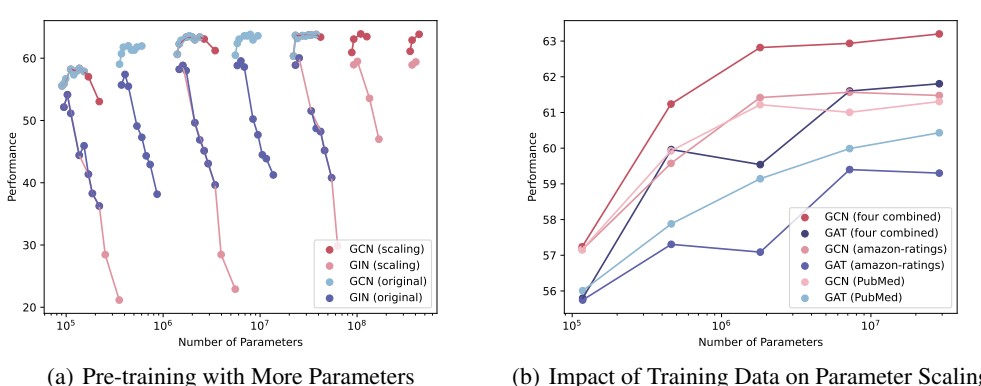

(a) Pre-training with More Parameters      (b) Impact of Training Data on Parameter Scaling

Figure S9: **Over-parameterization in Parameter Scaling.** (a) The connected dots denote scaling the number of parameters by stacking multiple layers. The adjacent line segments denote scaling by expanding the hidden dimension. (b) "Four combined" denotes employing amazon-ratings, ogbn-arxiv, Facebook, and roman-empire for pre-training.

## D.2 SCALING PARAMETERS

Following the common practice in LLM (OpenAI et al., 2024), model performance is also compared with different numbers of parameters. The experimental settings are the same as Fig. 5, with models pre-trained on (amazon-ratings, ogbn-arxiv, Facebook, roman-empire) and performance averaged across different test graphs. The connected dots denote scaling the number of parameters by stacking multiple layers. Note that $N^2$ is a recurrent model, where the number of parameters does not change with different layer depths. Results in Fig. S8 exhibit the clear influence driven by layer depth and hidden dimension. Notably, increasing parameter count by adding more layers does not consistently improve performance, while scaling hidden dimensionality is a more stable and beneficial strategy to improve pre-trained GNN models. These observations indicate that model capacity cannot be assessed solely through parameter volume. Instead, layer depth must be chosen appropriately for each architecture, as simply increasing parameters by stacking layers does not always yield better performance.

We further explore the boundary of parameter scaling with more parameters, *i.e.*, hidden dimensions of {2048, 4096} and depths of {20, 32} layers. Fig. S9(a) shows that further scaling the number of layers (connected dots) causes over-parameterization. This can be alleviated by expanding training datasets, where results in Fig. S9(b) indicate that more training datasets can better support parameter scaling.

In contrast, increasing the number of hidden dimensions to 4096 does not show over-parameterization. This observation further supports the conclusion that scaling hidden dimensionality is a more stable and beneficial strategy for improving pre-trained GNN models.

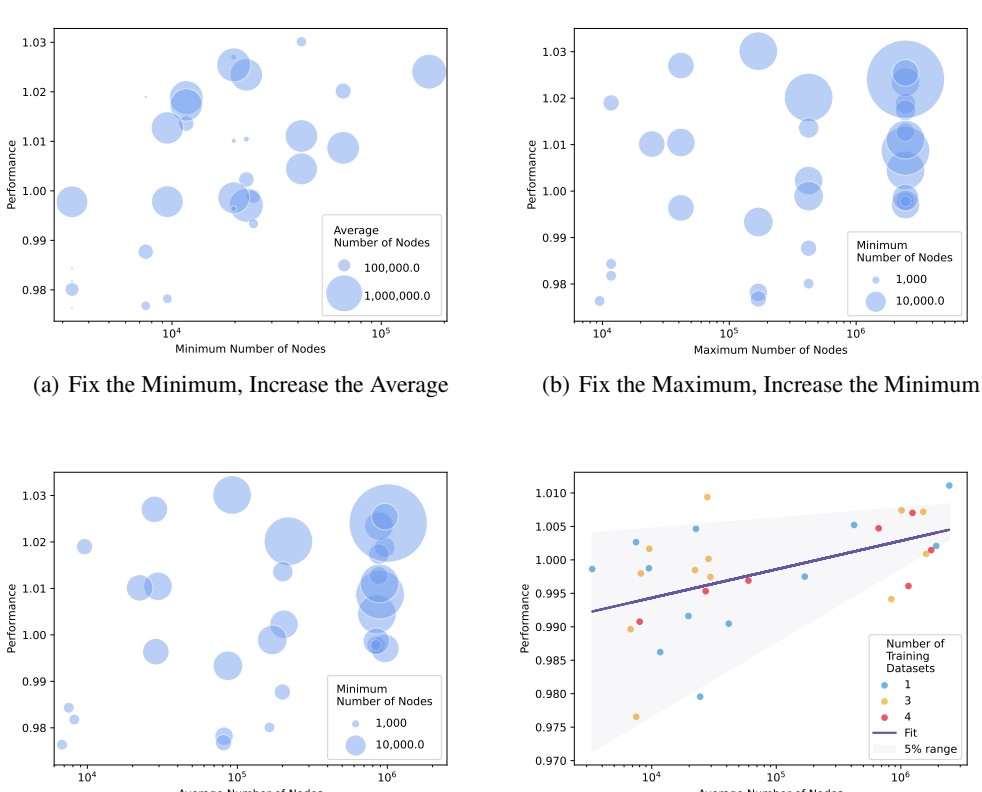

(a) Fix the Minimum, Increase the Average  (b) Fix the Maximum, Increase the Minimum

(c) Fix the Average, Increase the Minimum  (d) Scaling with the Average Number of Nodes

Figure S10: **Study on the Data Scaling Strategy.** (c) Different dot colors indicate the number of datasets employed for pre-training. "5%" denotes envelope-fitting with the top and bottom 5% of the data in each bin.

## D.3 DATA SCALING STRATEGY

To explore an effective data scaling strategy, we compare model pre-training with different training data. Specifically, different combinations of three datasets are employed as training data. Model performance is normalized by dividing the median result of the same backbone pre-trained with various datasets on the corresponding test dataset. Fig. S10(a) shows the results of fixing the minimum dataset size and varying the average number of nodes. When the minimum training dataset size remains, increasing the average number of nodes does not consistently improve performance. This indicates that simply adding larger datasets while keeping the smallest size unchanged is not an effective way to expand the training set.

Moreover, Fig. S10(b) and Fig. S10(c) show the results of fixing the maximum dataset size or the average number of nodes and increasing the minimum. In both cases, raising the minimum dataset sizes yields improved performance. Together, these findings suggest that mixing datasets with widely varying node scales is inefficient; datasets should instead be chosen to maintain a similar node scale. Therefore, we compare dataset combinations that are aligned in node scale for the data scaling study in Fig. 6. Additionally, we directly expand the average number of nodes in dataset combinations without considering the smallest dataset size. Model performance is normalized by dividing the median result of the same backbone pre-trained with various datasets on the corresponding test dataset. The averaged results over 1,848 data points across different backbones and test datasets are

presented in Fig. S10(d). Results show that increasing the average number of nodes benefits model performance. In contrast, incorporating more training datasets with various node scales does not necessarily result in superior performance, which verifies the inefficiency of this strategy.

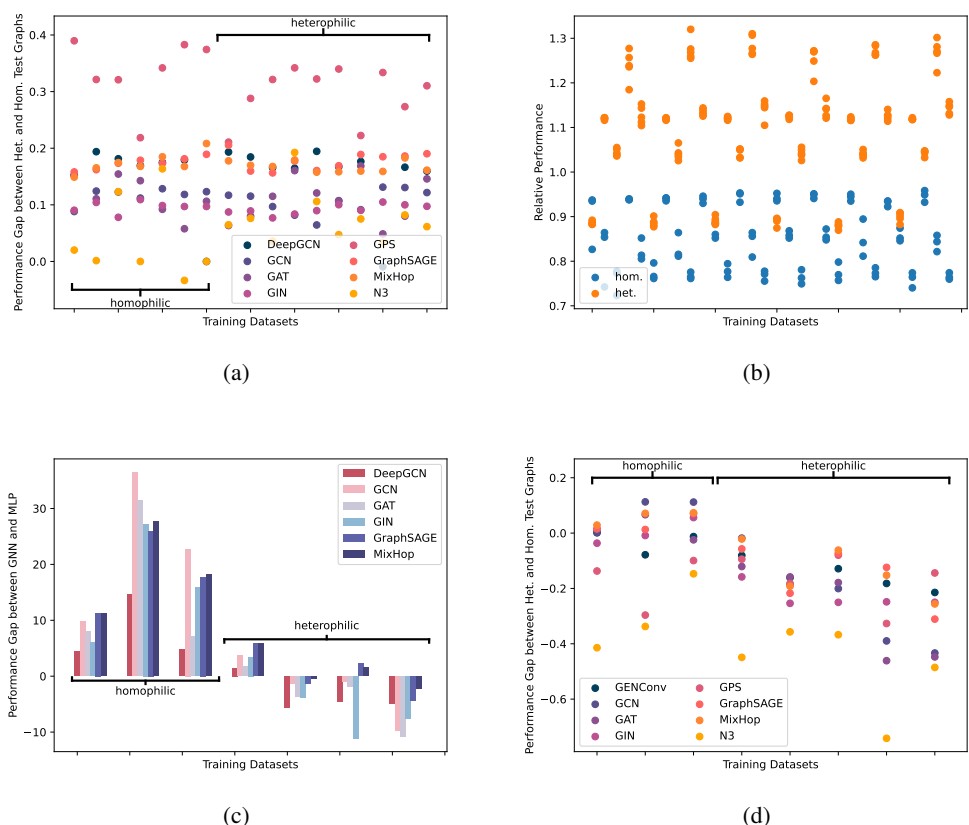

(a)                                                          (b)

(c)                                                          (d)

Figure S11: **Adaptation Performance Comparison on Heterophilic and Homophilic graphs.** **(a)** presents the difference between the performance of the same pre-trained model evaluated on heterophilic graphs and homophilic graphs. **(b)** presents the normalized performance of MLP by dividing the state-of-the-art supervised results of classic GNN models on the corresponding test datasets. **(c)** presents the difference between the absolute performance of pre-trained GNNs and MLP. **(d)** presents the difference between the performance of pre-trained GNNs and MLP normalized by their supervised counterpart.

## D.4   INFERENCE WITH DIFFERENT HOMOPHILY-HETEROPHILY

**Comparison with the same pre-training dataset.** Sec. 4.2 examines model adaptation using different pre-training datasets on the same test dataset, showing that training on datasets with the same homophily–heterophily characteristics as the test dataset generally yields better results. In this section, we shift the focus to comparing model performance across different test datasets under the same pre-training setting. However, direct comparison is challenging because the inherent difficulty of each test dataset varies, resulting in different absolute performance values. To address this issue, we normalize the results by dividing them by the corresponding reproduced supervised results. A higher normalized value indicates that the pre-trained model adapts more effectively to the corresponding downstream task. Fig. S11(a) presents the difference between the averaged performance of the same pre-trained model on heterophilic test datasets and homophilic test datasets ($\texttt{Metric}_{het} - \texttt{Metric}_{hom}$). We can see that models pre-trained on either homophilic or heterophilic graphs gain a positive performance gap, indicating better results on heterophilic graphs than homophilic graphs. To further study this phenomenon, we contrast pre-trained MLPs with pre-trained GNNs to evaluate the requirement of

homophilic and heterophilic graphs during model adaptation, and thus derive the root cause of the phenomenon for Fig. S11(a).

**Requirement of heterophilic graphs.** We compare the performance of pre-trained MLP with supervised GNNs. Supervised GNNs are directly optimized to meet task-specific requirements on downstream datasets. Comparing them with pre-trained models assesses whether the latter also fulfill these requirements. Fig. S11(b) presents the normalized performance of MLP by dividing the state-of-the-art supervised results of classic GNN models on the corresponding test datasets. MLP achieves more comparable performance on heterophilic test graphs compared to supervised results. This indicates that node features better benefit the adaptation to heterophilic graphs. In contrast, the requirement of the adaptation to homophilic graphs cannot be well satisfied by MLP and leads to inferior normalized performance.

**Requirement of homophilic graphs.** We compare the performance of pre-trained MLPs and GNNs. MLPs that only employ node features serve as a baseline for adapting to homophilic graphs. Comparing more complex models with MLP highlights the further requirement of homophilic graphs. Fig. S11(c) presents the performance gap between pre-trained GNNs and MLPs. We can see that GNNs gain better performance on homophilic graphs and similar performance on heterophilic graphs compared to MLP. This indicates that capturing graph structures contributes to the adaptation to homophilic graphs.

**Transferability of the learned knowledge.** Given the requirement of capturing node features to adapt to heterophilic graphs and capturing structures for homophilic graphs, we further analyze the difficulty of transferring these learned patterns. Specifically, supervised models transfer knowledge within the same dataset, while pre-trained models transfer across datasets. Therefore, comparing the performance of pre-trained models with their supervised counterparts shows the transferability of knowledge learned during pre-training, where GNNs correspond to the structural knowledge and MLP corresponds to the node feature knowledge. Fig. S11(d) presents the normalized performance gap between pre-trained GNNs and MLP, where pre-training results are divided by the supervised results of the same model on the same datasets. Results show that the performance gap between normalized pre-trained GNNs and MLP is generally negative, where the pre-trained MLP is more comparable to its supervised counterpart. This suggests that the node feature knowledge is consistently transferable within and across datasets. Conversely, structural knowledge transfers well within the same dataset but fails to generalize across different datasets. As a result, pre-trained GNNs with only transferable node feature knowledge cannot satisfy the requirement of the homophilic graphs and thus achieve better normalized performance on heterophilic graphs.

Based on the above conclusions, the phenomenon in Fig. S11(a) can be attributed to the inherent differences between homophilic and heterophilic graphs. Adapting to heterophilic graphs mostly requires the capturing of node features, while adapting to homophilic graphs requires models to adhere closely to the input graph structures. However, structural knowledge fails to transfer across different datasets compared to the better transferability of the node feature knowledge, resulting in consistently better performance when adapting to heterophilic graphs.

## D.5 MODEL FINE-TUNING

Our unified I/O modules enable seamless adaptation of pure GNN architectures across diverse datasets. To further evaluate the effectiveness of the pre-trained GNN operators, we fine-tune the models pre-trained on amazon-ratings, ogbn-arxiv, Facebook, and roman-empire. The internal GNN module $F_g$ within the pre-trained models is frozen during the fine-tuning. The unification-oriented I/O function $f_{in}(\cdot)$ in Eq. 4 and $f_{out}(\cdot)$ in Eq. 5 are replaced with learnable parameters. We implemented GCN (Kipf & Welling, 2017), GAT (Veličković et al., 2018a), GIN (Xu et al., 2019), GraphSAGE (Hamilton et al., 2017), and GraphGPS (Rampášek et al., 2022) in the supervised-learning setting and DGI (Veličković et al., 2018b), GRACE (Zhu et al., 2020), GraphACL (Xiao et al., 2023), GraphCL (You et al., 2020), SimGRACE (Xia et al., 2022), MaskGAE (Li et al., 2023), GraphMAE2 (Hou et al., 2023) in the self-supervised setting. The fine-tuning results with different backbones on downstream graphs are summarized in Tab. S9. The pre-trained operators achieve superior performance to supervised methods and self-supervised methods. Notably, the pre-trained operators require minimal hyperparameter tuning, with only dropout adjusted during fine-tuning. This

Table S9: **Evaluation Results of Fine-tuning the I/O Modules in the Pre-trained GNNs (Measured by accuracy except ROC AUC for tolokers: %).** Bold values denote the best results per test dataset.

| | CORA | PUBMED | AMAZON COMPUTERS | WIKICS | AMAZON-RATINGS | MINESWEEPER | TOLOKERS |
|---|---|---|---|---|---|---|---|
| | | | SELF-SUPERVISED | | | | |
| DGI | $84.00_{\pm0.28}$ | $83.73_{\pm0.41}$ | $82.11_{\pm0.16}$ | $75.05_{\pm0.39}$ | $40.80_{\pm0.74}$ | $88.45_{\pm0.38}$ | $77.73_{\pm0.14}$ |
| GRACE | $84.30_{\pm0.57}$ | $85.81_{\pm0.18}$ | $89.67_{\pm0.36}$ | $75.80_{\pm0.56}$ | $42.19_{\pm0.12}$ | $86.15_{\pm0.45}$ | $75.06_{\pm0.14}$ |
| GRAPHACL | $75.00_{\pm0.75}$ | $82.93_{\pm0.17}$ | $80.58_{\pm0.15}$ | $68.00_{\pm0.75}$ | $40.65_{\pm0.14}$ | $87.23_{\pm0.11}$ | $77.68_{\pm0.25}$ |
| GRAPHCL | $63.33_{\pm1.76}$ | $63.53_{\pm1.08}$ | $84.57_{\pm0.34}$ | $76.32_{\pm0.18}$ | $42.35_{\pm0.37}$ | $79.85_{\pm0.12}$ | $80.03_{\pm0.12}$ |
| MASKGAE | $75.13_{\pm1.78}$ | $75.27_{\pm1.03}$ | $92.15_{\pm0.05}$ | $78.25_{\pm0.20}$ | $43.54_{\pm0.30}$ | $84.33_{\pm0.16}$ | $81.13_{\pm0.34}$ |
| SIMGRACE | $67.07_{\pm0.82}$ | $77.63_{\pm0.82}$ | $87.44_{\pm0.18}$ | $78.75_{\pm0.28}$ | $43.46_{\pm0.23}$ | $84.23_{\pm0.16}$ | $80.29_{\pm0.30}$ |
| GRAPHMAE2 | $79.50_{\pm0.51}$ | $67.07_{\pm0.91}$ | $91.03_{\pm0.19}$ | $76.24_{\pm0.11}$ | $40.95_{\pm0.71}$ | $80.16_{\pm0.10}$ | $80.17_{\pm0.07}$ |
| | | | SUPERVISED | | | | |
| GRAPHSAGE | $78.83_{\pm0.50}$ | $88.11_{\pm0.05}$ | $91.09_{\pm0.02}$ | $78.13_{\pm0.15}$ | $45.71_{\pm0.38}$ | $90.55_{\pm0.10}$ | $83.06_{\pm0.59}$ |
| GAT | $77.51_{\pm2.35}$ | $85.30_{\pm0.15}$ | $89.78_{\pm0.02}$ | $76.35_{\pm0.30}$ | $44.54_{\pm0.52}$ | $82.07_{\pm1.17}$ | $77.37_{\pm0.28}$ |
| GIN | $77.36_{\pm0.15}$ | $85.13_{\pm0.55}$ | $90.51_{\pm0.80}$ | $74.02_{\pm0.62}$ | $46.33_{\pm0.11}$ | $74.93_{\pm0.58}$ | $60.93_{\pm2.25}$ |
| GCN | $80.35_{\pm0.25}$ | $85.44_{\pm0.50}$ | $90.66_{\pm0.13}$ | $78.55_{\pm0.01}$ | $46.71_{\pm0.25}$ | $76.43_{\pm1.05}$ | $77.79_{\pm0.12}$ |
| GRAPHGPS | $58.61_{\pm0.05}$ | $85.21_{\pm0.30}$ | $88.87_{\pm0.20}$ | $75.18_{\pm0.04}$ | $47.85_{\pm0.29}$ | $89.64_{\pm0.24}$ | $79.82_{\pm0.06}$ |
| | | | PRE-TRAINED AND FINE-TUNED | | | | |
| GCN | $84.32_{\pm0.09}$ | $85.28_{\pm0.21}$ | $91.02_{\pm0.01}$ | $78.42_{\pm0.07}$ | $46.17_{\pm0.09}$ | $69.10_{\pm0.11}$ | $69.50_{\pm0.69}$ |
| GAT | $78.01_{\pm0.87}$ | $84.47_{\pm0.33}$ | $89.74_{\pm0.42}$ | $77.96_{\pm0.13}$ | $47.13_{\pm0.45}$ | $71.03_{\pm0.72}$ | $75.67_{\pm0.73}$ |
| GIN | $79.31_{\pm0.82}$ | $85.61_{\pm0.38}$ | $87.89_{\pm0.43}$ | $73.49_{\pm0.14}$ | $49.93_{\pm0.22}$ | $77.62_{\pm0.28}$ | $66.83_{\pm0.56}$ |
| GRAPHGPS | $50.62_{\pm0.42}$ | $86.79_{\pm0.71}$ | $85.93_{\pm0.23}$ | $73.41_{\pm0.74}$ | $43.74_{\pm0.26}$ | $88.68_{\pm0.29}$ | $80.41_{\pm0.53}$ |
| GRAPHSAGE | $83.92_{\pm0.43}$ | $86.37_{\pm0.23}$ | $91.24_{\pm0.17}$ | $\mathbf{78.98}_{\pm0.12}$ | $48.85_{\pm0.82}$ | $\mathbf{91.39}_{\pm0.15}$ | $\mathbf{83.29}_{\pm0.13}$ |
| MIXHOP | $\mathbf{84.63}_{\pm0.12}$ | $\mathbf{88.91}_{\pm0.45}$ | $90.33_{\pm0.30}$ | $78.93_{\pm0.17}$ | $\mathbf{51.59}_{\pm0.05}$ | $90.77_{\pm0.18}$ | $83.03_{\pm0.13}$ |
| $N^2$ | $81.50_{\pm0.33}$ | $88.32_{\pm0.32}$ | $\mathbf{92.33}_{\pm0.25}$ | $76.60_{\pm0.25}$ | $49.85_{\pm0.31}$ | $90.31_{\pm0.31}$ | $81.51_{\pm0.36}$ |
| DEEPGCN | $74.40_{\pm0.33}$ | $88.50_{\pm0.32}$ | $91.02_{\pm0.25}$ | $74.67_{\pm0.25}$ | $50.76_{\pm0.46}$ | $88.00_{\pm0.28}$ | $79.86_{\pm0.36}$ |

significantly simplifies the hyperparameter tuning process, enabling efficient adaptation of pre-trained GNNs to various graphs with promising performance.

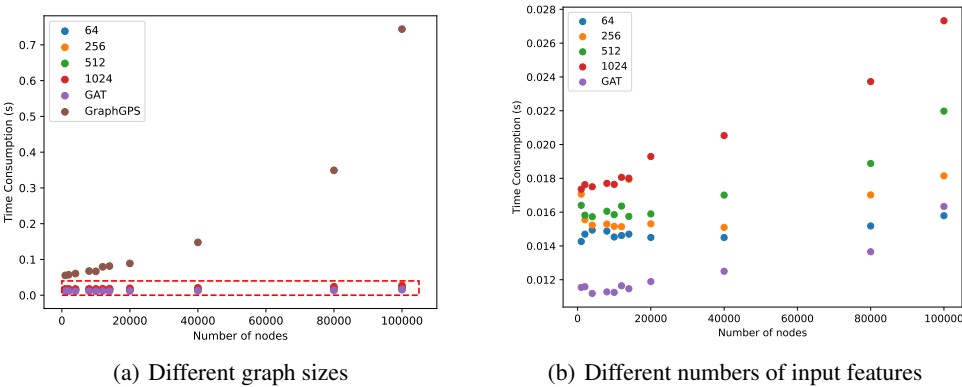

(a) Different graph sizes    (b) Different numbers of input features

Figure S12: **Time comparison.** (b) zooms in on the red block in (a).

## D.6    COMPLEXITY ANALYSIS

The space complexity of the unified I/O is $O(n)$, with $d_{\texttt{in}}, d, s, c \ll n$. To evaluate the time cost of different modules, We construct synthetic graphs. Nodes in the synthetic graphs have an average degree of 20. The largest number of edges is 100,000. The number of classes is 10, which is close to the common configuration of the real-world datasets in Fig. 1. The number of GNN layers is fixed to 3. The results are presented in Fig. S12. As the number of input features increases, the time cost of the unified I/O module increases but is constantly less than the cost of the GraphGPS module. The I/O time cost also scales linearly with the graph size. This demonstrates the efficiency of the proposed unified I/O modules.

Table S10: **Information Loss Study.** "Ori" and "Uni" denote the supervised learning results with traditional I/O modules and our unified I/O modules, respectively.

| | AMAZON-RATINGS | ARXIV-YEAR | COAUTHORCS | COAUTHORPHYSICS |
|---|---|---|---|---|
| GCN (ORI.) | 48.70 | 46.02 | 92.92 | 96.18 |
| GCN (UNI.) | 47.12 | 45.19 | 92.77 | 96.38 |
| GAT (ORI.) | 52.70 | 46.05 | 93.61 | 96.17 |
| GAT (UNI.) | 46.19 | 46.26 | 88.38 | 94.16 |
| GRAPHSAGE (ORI.) | 53.63 | 43.76 | 93.91 | 96.49 |
| GRAPHSAGE (UNI.) | 45.45 | 44.76 | 95.17 | 97.06 |

## D.7 EFFECTIVENESS OF THE UNIFIED I/O

**Information Loss.** Unified I/O decouples the learnable parameters from the numbers and semantics of dimensions for the feature and label space. To evaluate whether this leads to information loss, we conduct supervised training with our unified I/O on GCN, GAT, and GraphSAGE. As presented in Tab. S10, our unified I/O modules do not cause severe information loss. They even enable the backbone GNN methods to achieve better performance than that of their vanilla architectures for GAT and GraphSAGE on arxiv-year, GraphSAGE on CoauthorCS, GCN and GraphSAGE on CoauthorPhysics. This demonstrates the effectiveness of our unified I/O in learning input and output mappings.

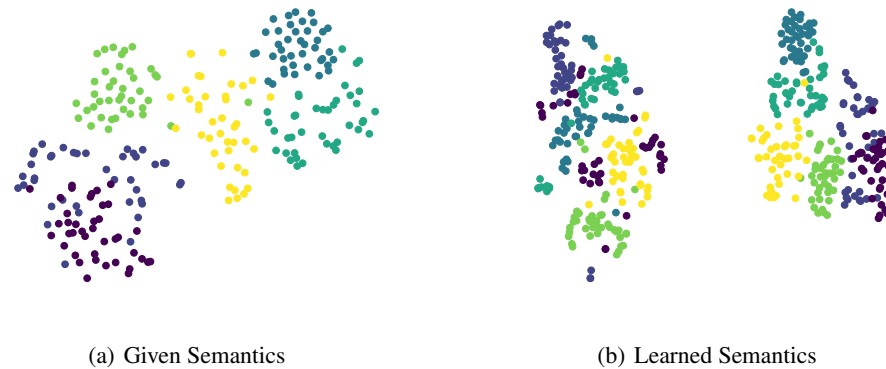

(a) Given Semantics                    (b) Learned Semantics

Figure S13: **t-SNE Results for Semantics Alignment Comparison.** Each dot represents a different feature channel, with dot colors representing different datasets.

**Feature Semantics.** To explore whether our parametric function $f_{in}(\cdot)$ empowers feature semantics unification across different inputs, we use t-SNE to visualize feature semantics from (Cora, CiteSeer, Photo, Computers, ogbn-products, and WikiCS) (Chen et al., 2024c). 40 feature channels are sampled for each dataset. Each dot represents a different feature channel, with dot colors representing different datasets. Fig. S13(a) shows the t-SNE result of the original feature semantics, where features from different datasets form clearly separated clusters rather than merging into a shared global structure. This pronounced dataset separation indicates that the representations are not well aligned within a common semantic space. Conversely, in Fig. S13(b), feature semantics from different datasets learned by unified I/O form two major clusters, indicating that the learned representations are unified into two common semantic subspaces. Within each cluster, channels from different datasets do not fully mix but show partial segregation, suggesting that the semantic representations still preserve dataset-specific characteristics. All these results demonstrate the effectiveness of our proposed method in modeling unified feature semantics.

# E  PROOF AND DERIVATION

## E.1  MAPPING WITH DIMENSION RELATIONS

**Theorem 3.1** (Mapping with Dimension Relations). *Given any linear mapping $\mathbf{W} \in \mathbb{R}^{d_{\text{src}} \times d_{\text{tgt}}}$ and $s \in \mathbb{N}^+$, there always exist two representation matrices $\mathbf{S}_{\text{src}} \in \mathbb{R}^{d_{\text{src}} \times s}$ and $\mathbf{S}_{\text{tgt}} \in \mathbb{R}^{d_{\text{tgt}} \times s}$, such that $\mathbf{W} = \psi(\mathbf{S}_{\text{src}}, \mathbf{S}_{\text{tgt}})$, where $\psi(\cdot, \cdot)$ is a bilinear composition function.*

**Definition E.1** (Bilinear Composition Function). *Let $\mathbf{X} \in \mathbb{R}^{m \times d_1}$ and $\mathbf{Y} \in \mathbb{R}^{n \times d_2}$ be two input matrices, and let $\mathbf{U} \in \mathbb{R}^{d_1 \times d_2}$ be a learnable parameter matrix. A bilinear composition function $\psi : \mathbb{R}^{m \times d_1} \times \mathbb{R}^{n \times d_2} \to \mathbb{R}^{m \times n}$ is defined as*

$$\psi(\mathbf{X}, \mathbf{Y}) = \mathbf{XUY}^\top,$$

*which computes the bilinear form between each pair of row vectors from $\mathbf{X}$ and $\mathbf{Y}$. This function is linear with respect to either argument when the other is fixed, but not jointly linear.*

We now provide the proof for Theorem 3.1:

*Proof.* Let $r = \text{rank}(\mathbf{W}) \leq \min(d_{\text{src}}, d_{\text{tgt}})$. By the full-rank factorization theorem (Meyer, 2023), there exist matrices $\mathbf{A} \in \mathbb{R}^{d_{\text{src}} \times r}$ and $\mathbf{B} \in \mathbb{R}^{d_{\text{tgt}} \times r}$ such that

$$\mathbf{W} = \mathbf{AB}^\top.$$

Let $s = r$, $\mathbf{S}_{\text{src}} := \mathbf{A}$, $\mathbf{S}_{\text{tgt}} := \mathbf{B}$, and $\mathbf{U}$ be the identity matrix $\mathbf{I}_r$. Then

$$\mathbf{S}_{\text{src}} \mathbf{U} \mathbf{S}_{\text{tgt}}^\top = \mathbf{AI}_r \mathbf{B}^\top = \mathbf{AB}^\top = \mathbf{W}.$$

Thus, such $\mathbf{S}_{\text{src}}, \mathbf{S}_{\text{tgt}}, \mathbf{U}$ always exist.

More generally, for any $s \geq r$, we can embed $\mathbf{A}$ and $\mathbf{B}$ into higher-dimensional matrices by padding zeros and set $\mathbf{U}$ as a diagonal matrix with the first $r$ entries as 1 and others as 0. Hence, the bilinear form $\psi(\mathbf{S}_{\text{src}}, \mathbf{S}_{\text{tgt}}) = \mathbf{S}_{\text{src}} \mathbf{U} \mathbf{S}_{\text{tgt}}^\top$ is expressive enough to represent any linear mapping $\mathbf{W}$. □

## E.2  SET LEARNING FOR THE UNIFIED INPUT MODULE

To decouple the parameters from the number of source dimensions, feature semantics $\mathbf{S}_{\text{src}}^{(\text{in})}$ is formulated as a parametric function $\mathbf{f}_{\text{in}}(\mathbf{X}; \mathcal{W}_{\text{in}})$. The function $\mathbf{f}_{\text{in}}(\cdot)$ is subject to two conditions: (1) Permutation invariance to the order of input nodes and equivariance to that of source dimensions; (2) Size independence of the parameter set $\mathcal{W}_{\text{in}}$ to the values of $n$ and $d_{\text{in}}$. Given the absence of topological structures and the permutation condition (Cond 1) for $\mathbf{f}_{\text{in}}(\cdot)$, the input features can be modeled as a set of channels $\{\mathbf{X}_{\cdot,j}\}$, where each channel corresponds to a set of nodes $\{\mathbf{X}_{i,j}\}$. As a result, $\mathbf{f}_{\text{in}}(\cdot)$ is transformed into a set-learning problem at both the channel level and the node level, $\mathbf{f}_{\text{in}} = \mathbf{f}_{\text{in}}^{\text{cha}} \circ \mathbf{f}_{\text{in}}^{\text{nod}}$.

**Channel-level Set Learning.** Based on the universal functions on set (Zaheer et al., 2017), $\mathbf{f}_{\text{in}}(\cdot)$ is a permutation-equivariant set function at the channel level and can be decomposed as

$$\mathbf{f}_{\text{in}}(\mathbf{X}) = \mathbf{f}_{\text{in}}^{\text{cha}}\left(\mathbf{f}_{\text{in}}^{\text{nod}}(\mathbf{X})\right) = \sigma\left[\Theta \mathbf{f}_{\text{in}}^{\text{nod}}(\mathbf{X})\right], \tag{S8}$$

where $\sigma$ can be any nonlinear function, $\Theta \in \mathbb{R}^{d_{\text{in}} \times d_{\text{in}}}$ denotes the channel mixer. To enable the scalability of the input module for input features with a large number of channels (*e.g.*, NELL $d_{\text{in}} = 61,278$, CoauthorPhysics $d_{\text{in}} = 8,415$, and CoauthorCS $d_{\text{in}} = 6,805$), we follow the linear attention (Katharopoulos et al., 2020) to construct $\Theta$ as

$$\Theta = \frac{d_{\text{in}} \mathbf{X}^\top \mathbf{X}}{\mathbf{X}^\top \mathbf{X} \mathbf{1}}, \tag{S9}$$

where $\mathbf{1}$ denotes the all-one vector.

**Node-level Set Learning.** Due to the size independence and the permutation invariance conditions for $\mathbf{f}_{\text{in}}(\cdot)$, $\mathbf{f}_{\text{in}}^{\text{nod}}(\cdot)$ in Eq. S8 can be formulated as a permutation-invariant set function at the node level. Given $\mathbf{f}_{\text{in}} : \mathbb{R}^{n \times d_{\text{in}}} \mapsto \mathbb{R}^{d_{\text{in}} \times s}$, our ultimate target is to model a number of $s$ representations

for each channel. Therefore, we apply $s$ set functions to $\{\mathbf{X}_{i,j}\}$. Each function can be decomposed following the universal set function (Zaheer et al., 2017) as

$$\mathbf{f}_{\texttt{in}}^{\texttt{nod}}(\mathbf{X}) = \left[\mathbf{f}_{\texttt{in}}^{\texttt{nod},1}(\mathbf{X}) || \cdots || \mathbf{f}_{\texttt{in}}^{\texttt{nod},s}(\mathbf{X})\right], \quad \mathbf{f}_{\texttt{in}}^{\texttt{nod},k}(\mathbf{X}) = \phi^k\left(\sum_{i \in [1,n]} \rho^k(\mathbf{X}^\top)_{\cdot,i}\right), \quad \text{(S10)}$$

where the parameters in both $\phi^k$ and $\rho^k$ are shared for each element in $\mathbf{X}$ to ensure the size independence condition (Cond 2). We share the function $\rho^k$ for all $k \in [1,s]$ as $\rho$ and implement $\phi^k$ as a parameter-weighting function to keep simplicity. As a result, Eq. S8-Eq. S10 can be complied as

$$\mathbf{f}_{\texttt{in}}(\mathbf{X}) = \sigma\left[\frac{d_{\texttt{in}}\mathbf{X}^\top\mathbf{X}}{\mathbf{X}^\top\mathbf{X}\mathbf{1}}\rho(\mathbf{X}^\top)\mathbf{1}\alpha^\top\right], \quad \text{(S11)}$$

where $\alpha = \{\alpha_k\} \in \mathbb{R}^{s \times 1}$ denotes the parameter vector with $k \in [1,s]$ to implement $\phi^k$, $\mathbf{1}$ denotes the all-one vector to implement the summation in Eq. S10.

Although Zaheer et al. also provides an implementation named Deep Sets based on the universal set function, our input module differs from this specific implementation in several key aspects. Specifically, graph learning requires permutation invariance over nodes, which constitute the set representations in our case. This demands the input module to decouple parameters from the representation dimensionality. In contrast, Deep Sets are not faced with such a condition. We further decompose the set-learning task into a bi-level formulation, while Deep Sets addresses the original single-level formulation. Moreover, our input module employs a linear-attention-like set mixer in Eq. S9, while Deep Sets applies sum or max pooling to mix sets.

### E.3 PERMUTATION INVARIANCE OF THE UNIFIED INPUT MODULE

**Theorem E.2** (Permutation Invariance of the Unified Input Module). *Let $\mathbf{P} \in \mathbb{R}^{d_{\texttt{in}} \times d_{\texttt{in}}}$ be any permutation matrix. Then the source-adaptive input module $\mathbf{F}_{\texttt{in}}$ is permutation invariant, such that $\mathbf{F}_{\texttt{in}}(\mathbf{XP}) = \mathbf{F}_{\texttt{in}}(\mathbf{X})$.*

*Proof.* Let $\mathbf{P} \in \mathbb{R}^{d_{\texttt{in}} \times d_{\texttt{in}}}$ be a permutation matrix. Consider the input module $\mathbf{F}_{\texttt{in}}(\mathbf{X}) = \sigma\left[\mathbf{X}\mathbf{f}_{\texttt{in}}(\mathbf{X})\mathbf{S}_{\texttt{tgt}}^{(\texttt{in})\top}\right]$, where $\sigma(\cdot)$ is applied element-wise. Applying the permutation giving rise to $\mathbf{F}_{\texttt{in}}(\mathbf{XP}) = \sigma\left[\mathbf{XP}\mathbf{f}_{\texttt{in}}(\mathbf{XP})\mathbf{S}_{\texttt{tgt}}^{(\texttt{in})\top}\right]$. Specifically, let $\mathbf{r} = \mathbf{X}^\top\mathbf{X}\mathbf{1}/\sqrt{n}$, $\bar{\mathbf{x}} = \mathbf{X}\mathbf{1}_{d_{\texttt{in}}}/d_{\texttt{in}}$, $\mathbf{f}_{\texttt{in}}(\mathbf{XP})$ can be formulated as

$$\mathbf{f}_{\texttt{in}}(\mathbf{XP}) = \sigma\left(\frac{(\mathbf{XP})^\top\mathbf{XP}\mathbf{P}^\top\mathbf{r}\alpha^\top}{(\mathbf{XP})^\top\bar{\mathbf{x}}}\right)$$

$$= \sigma\left(\frac{(\mathbf{XP})^\top\mathbf{X}\mathbf{r}\alpha^\top}{(\mathbf{XP})^\top\bar{\mathbf{x}}}\right).$$

Here, both the division and $\sigma(\cdot)$ are applied element-wise and invariant to consistent column permutation. Therefore, applying the permutation before or after the element-wise operations yields the same result, giving

$$\mathbf{f}_{\texttt{in}}(\mathbf{XP}) = \sigma\left(\mathbf{P}^\top\frac{\mathbf{X}^\top\mathbf{X}\mathbf{r}\alpha^\top}{\mathbf{X}^\top\bar{\mathbf{x}}}\right)$$

$$= \mathbf{P}^\top\sigma\left(\frac{\mathbf{X}^\top\mathbf{X}\mathbf{r}\alpha^\top}{\mathbf{X}^\top\bar{\mathbf{x}}}\right)$$

$$= \mathbf{P}^\top\mathbf{f}_{\texttt{in}}(\mathbf{X}).$$

Substituting $\mathbf{f}_{\texttt{in}}(\mathbf{XP})$ into $\mathbf{F}_{\texttt{in}}(\mathbf{XP})$, we have

$$\mathbf{F}_{\texttt{in}}(\mathbf{XP}) = \sigma\left[\mathbf{XP}\mathbf{P}^\top\mathbf{f}_{\texttt{in}}(\mathbf{X})\mathbf{S}_{\texttt{tgt}}^{(\texttt{in})\top}\right]$$

$$= \mathbf{F}_{\texttt{in}}(\mathbf{X}).$$

which completes the proof. $\square$

### E.4 Pseudo Label Assignment

Given the observed labels $\mathbf{C}$, the mapping relations between the pseudo labels and the observed labels can be formulated as

$$\text{softmax}(\hat{\mathbf{C}}\mathbf{P}) = \frac{\exp(\hat{\mathbf{C}}\mathbf{P})}{\exp(\hat{\mathbf{C}}\mathbf{P})\mathbf{1}\mathbf{1}^\top} = \mathbf{C}, \tag{S12}$$

where $\mathbf{P} \in \mathbb{R}^{c \times c}$ denotes the assignment matrix, $\mathbf{1}$ denotes the all-one vector. For a set of values $\{x_i\}, i \in [1, c]$, the first-order Taylor expansion of $\exp(x_i)$ around $\bar{x} = \sum_i x_i/c$ is

$$\exp(x_i) \approx \exp(\bar{x}) \cdot (1 + x_i - \bar{x}). \tag{S13}$$

As a result, the summation of $\exp(x_i)$ can be approximated as

$$\sum_i \exp(x_i) \approx \exp(\bar{x}) \sum_i (1 + x_i - \bar{x}) = n\exp(\bar{x}). \tag{S14}$$

Substituting Eq. S14 in Eq. S12 yields

$$\frac{\exp(\hat{\mathbf{C}}\mathbf{P})}{\exp(\frac{1}{c}\hat{\mathbf{C}}\mathbf{P}\mathbf{1}\mathbf{1}^\top)} = c\mathbf{C}$$

$$\exp(\hat{\mathbf{C}}\mathbf{P} - \frac{1}{c}\hat{\mathbf{C}}\mathbf{P}\mathbf{1}\mathbf{1}^\top) = c\mathbf{C} \tag{S15}$$

$$\exp[\hat{\mathbf{C}}\mathbf{P}(\mathbf{I} - \frac{1}{c}\mathbf{1}\mathbf{1}^\top)] = c\mathbf{C}$$

$$\hat{\mathbf{C}}\mathbf{P}(c\mathbf{I} - \mathbf{1}\mathbf{1}^\top) = c\log(c\mathbf{C}).$$

## F LLM Usage

In preparing this manuscript, we employed a large language model (LLM) exclusively for surface-level language refinement, such as grammar correction and improving clarity of expression. The LLM did not contribute to method ideation and experimental study.

## G Limitation

In this paper, we explore the training-free adapting capability of the pre-trained models with pure GNNs. However, although the proposed method can be employed for any graph learning task (Appendix A), the empirical evaluation in this paper is limited to node classification. Further study on graph-level and edge-level tasks is left for future work.

Table S11: **Dataset Statistics.** "✓" marks the training datasets as different scales, where "S", "M", "L" denote small-scale, middle-scale, and large-scale datasets with numbers of nodes around 1k, 10k, and 100k, respectively.

| Category | Dataset | Usage | Type | #Nodes | #Features | #Labels | E-COM. ONLY S | E-COM. ONLY M | E-COM. ONLY L | CITATION ONLY S | CITATION ONLY M | CITATION ONLY L | SOCIAL ONLY S | SOCIAL ONLY M | SOCIAL ONLY L | WIKI. ONLY S | WIKI. ONLY M | WIKI. ONLY L |
|---|---|---|---|---|---|---|---|---|---|---|---|---|---|---|---|---|---|---|
| E-COM. | AmazonComputers | TEST | HOM. | 13,381 | 767 | 10 | ✓ | | | | | | | | | | | |
| E-COM. | AmazonPhoto | TRAIN | HOM. | 7,487 | 745 | 8 | | ✓ | | | | | | | | | | |
| E-COM. | Amazon-Ratings | TRAIN | HET. | 24,492 | 300 | 5 | | ✓ | | | | | | | | | | |
| E-COM. | OGBN-Products | TRAIN | HOM. | 2,449,029 | 100 | 47 | | | ✓ | | | | | | | | | |
| CITATION | Cora | TEST | HOM. | 2,708 | 1,433 | 7 | | | | ✓ | | | | | | | | |
| CITATION | CoauthorPhysics | TEST | HOM. | 34,493 | 8,415 | 5 | | | | | ✓ | | | | | | | |
| CITATION | Arxiv-Year | TEST | HET. | 169,343 | 128 | 5 | | | | | | ✓ | | | | | | |
| CITATION | CiteSeer | TRAIN | HOM. | 3,327 | 3,703 | 6 | | | | ✓ | | | | | | | | |
| CITATION | PubMed | TRAIN | HOM. | 19,717 | 500 | 3 | | | | | ✓ | | | | | | | |
| CITATION | OGBN-Arxiv | TRAIN | HOM. | 169,343 | 128 | 49 | | | | | | ✓ | | | | | | |
| CITATION | Snap-Patents | TRAIN | HET. | 2,923,922 | 269 | 5 | | | | | | ✓ | | | | | | |
| SOCIAL | Twitch-Gamer | TEST | HET. | 168,114 | 7 | 2 | | | | | | | | | ✓ | | | |
| SOCIAL | Tolokers | TEST | HET. | 11,758 | 10 | 2 | | | | | | | | ✓ | | | | |
| SOCIAL | Twitch-E | TRAIN | HET. | 9,498 | 128 | 2 | | | | | | | ✓ | | | | | |
| SOCIAL | FB100 | TRAIN | HET. | 41,554 | 5 | 2 | | | | | | | | ✓ | | | | |
| SOCIAL | Genius | TRAIN | HET. | 421,961 | 12 | 2 | | | | | | | | | ✓ | | | |
| SOCIAL | Facebook | TRAIN | HOM. | 22,470 | 128 | 4 | | | | | | | | ✓ | | | | |
| SOCIAL | Pokec | TRAIN | HET. | 1,632,803 | 65 | 2 | | | | | | | | | ✓ | | | |
| WIKI. | Chameleon | TEST | HET. | 2,277 | 2,325 | 5 | | | | | | | | | | ✓ | | |
| WIKI. | Actor | TEST | HET. | 7,600 | 932 | 5 | | | | | | | | | | ✓ | | |
| WIKI. | WikiCS | TRAIN | HOM. | 11,701 | 300 | 10 | | | | | | | | | | | ✓ | |
| WIKI. | Roman-Empire | TRAIN | HET. | 22,662 | 300 | 18 | | | | | | | | | | | ✓ | |
| WIKI. | NELL | TRAIN | HET. | 65,755 | 61,278 | 186 | | | | | | | | | | | | ✓ |

Table S12: **Dataset Combinations for Pre-training.** "✓" marks the selected datasets for different setups. "S", "M", "L" denote small-scale, middle-scale, and large-scale datasets with numbers of nodes around 1k, 10k, and 100k, respectively. "No XX" denotes pre-training on datasets from three domains and adapting to the remaining one for the domain gap experiment.

| | DATASET | USAGE | TYPE | #NODES | #FEATURES | #LABELS | NO E-COM. | | | NO CITATION | | | NO SOCIAL | | | NO WIKI. | | | ALL | | | HET. | HOM. | MIX. |
|---|---|---|---|---|---|---|---|---|---|---|---|---|---|---|---|---|---|---|---|---|---|---|---|---|
| | | | | | | | S | M | L | S | M | L | S | M | L | S | M | L | S | M | L | | | |
| E-COM. | AMAZONCOMPUTERS | TEST | HOM. | 13,381 | 767 | 10 | | | | | | | | | | | | | | | | | | ✓ |
| | AMAZONPHOTO | TRAIN | HOM. | 7,487 | 745 | 8 | | | | ✓ | | | ✓ | | | ✓ | | | ✓ | | | | ✓ | |
| | AMAZON-RATINGS | TRAIN | HET. | 24,492 | 300 | 5 | | | | | ✓ | | | ✓ | | | ✓ | | | ✓ | | ✓ | | ✓ |
| | OGBN-PRODUCTS | TRAIN | HOM. | 2,449,029 | 100 | 47 | | | | | | ✓ | | | ✓ | | | ✓ | | | ✓ | | ✓ | ✓ |
| CITATION | CORA | TEST | HOM. | 2,708 | 1,433 | 7 | | | | | | | | | | | | | | | | | | |
| | COAUTHORPHYSICS | TEST | HOM. | 34,493 | 8,415 | 5 | | | | | | | | | | | | | | | | | | |
| | ARXIV-YEAR | TEST | HET. | 169,343 | 128 | 5 | | | | | | | | | | | | | | | | ✓ | | ✓ |
| | CITESEER | TRAIN | HOM. | 3,327 | 3,703 | 6 | ✓ | | | | | | ✓ | | | ✓ | | | ✓ | | | | | |
| | PUBMED | TRAIN | HOM. | 19,717 | 500 | 3 | | ✓ | | | | | | ✓ | | | ✓ | | | | | | ✓ | |
| | OGBN-ARXIV | TRAIN | HOM. | 169,343 | 128 | 40 | | | ✓ | | | | | | ✓ | | | ✓ | | | ✓ | | ✓ | ✓ |
| | SNAP-PATENTS | TRAIN | HET. | 2,923,922 | 269 | 5 | | | | | | | | | | | | | | | | ✓ | | ✓ |
| SOCIAL | TWITCH-GAMER | TEST | HET. | 168,114 | 7 | 2 | | | | | | | | | | | | | | | | | | |
| | TOLOKERS | TEST | HET. | 11,758 | 10 | 2 | | | | | | | | | | | ✓ | | | ✓ | | | | |
| | TWITCH-E | TRAIN | HET. | 9,498 | 128 | 2 | ✓ | | | ✓ | | | | | | ✓ | | | ✓ | | | ✓ | | |
| | FB100 | TRAIN | HET. | 41,554 | 5 | 2 | | | ✓ | | | ✓ | | | | | | ✓ | | | ✓ | | | |
| | GENIUS | TRAIN | HET. | 421,961 | 12 | 2 | | | | | | | | | | | ✓ | | | ✓ | | | | ✓ |
| | FACEBOOK | TRAIN | HOM. | 22,470 | 128 | 4 | | | | | | | | | | | | | | | | | ✓ | ✓ |
| | POKEC | TRAIN | HET. | 1,632,803 | 65 | 2 | | | | | | | | | | | | | | | | ✓ | | |
| WIKI. | CHAMELEON | TEST | HET. | 2,277 | 2,325 | 5 | | | | | | | | | | | | | | | | | | |
| | ACTOR | TEST | HET. | 7,600 | 932 | 5 | | | | | | | | | | ✓ | | | ✓ | | | | | |
| | WIKICS | TRAIN | HOM. | 11,701 | 300 | 10 | ✓ | | | ✓ | | | ✓ | | | | | | | ✓ | | | ✓ | ✓ |
| | ROMAN-EMPIRE | TRAIN | HET. | 22,662 | 300 | 18 | | ✓ | | | ✓ | | | ✓ | | | | | | ✓ | | ✓ | | ✓ |
| | NELL | TRAIN | HET. | 65,755 | 61,278 | 186 | | | ✓ | | | ✓ | | | ✓ | | | | | | ✓ | ✓ | | |

