# OpenReview forum: "Pre-training Pure GNNs as Graph Learners"
_ICLR.cc/2026/Conference — Submitted to ICLR 2026_

### Official Review · Reviewer_nQLH · 2025-10-21

**Soundness:** 3
**Presentation:** 2
**Contribution:** 3
**Rating:** 4
**Confidence:** 4

**Summary:**

This paper proposes input (on node features) and output (for node predictions) modules for GNNs to allow pre-training them and performing downstream predictions on different tasks. Empirically, this paper demonstrates that downstream performances improves with increased training data.

**Strengths:**

- **(S1) Contribution (Originality)** Cross dataset training for GNNs is an important unsolved problem. This paper proposes a relatively clean solution to the key problems of cross dataset training (mapping input and outputs).
- **(S2) Empircial Evaluation (Quality)** The empirical evaluation is well done. The authors test on many datasets from different domains and evaluate many different backbones (=GNNs). They investigate the performance of their methods in different settings (1-shot, 3-shot, fine-tuning) and compare against a diverse array of models.
- **(S3) Performance (Significance)** Finally, performance of the model in few-shot and fine-tuning scenarios is good and often outperforms all other models.

**Weaknesses:**

- **(W1) Clarity of definitions:** Section 3 is dense and difficult to understand. In particular, the definition of the I/O modules is difficult to follow due to the density of notation. Here are some points that need clarification:
	-   $\mathbf{S}\_{\text{src}}$ and $\mathbf{S}\_{\text{tgt}}$  are unintuitive, are these simply learned weight matrices?
	- The definition of $f_\texttt{in}$ states that it maps from $\mathbb{R}^{n \times d_{\text{in}}}$ to $\mathbb{R}^{d_{\text{in} \times s}}$. However, this means it does not produce node-embeddings?
	- What is the difference between your input module and DeepSet?

- **(W2) Grounding of empirical results:** Figure 5 is difficult to interpret, it is supposed to show a performance gap between models on different domains / different heterophily but it is unclear to me what the figure shows (and how big the performance gap actually is).


- **(W3) Scaling:**
	- **(W3.1)** Figure 3 shows the relation between GNN model size (#layers or hidden dimension) against test set performance. We can see that for the best performing models such as GCN or mixhop the model size has very limited impact on test performance.
	- **(W3.2)** Figure 4 shows the relation between the average number of nodes and test performance. While there is a positive correlation it seems to be quite small (it would be good if the authors could quanitfy this). Furthermore, training on more datasets seems to have little impact on test performance.
	- Combined (W3.1) and (W3.2) seem to indicate a fundamental limitation of this approach. While pre-training does clearly give us good performance in (few-shot) settings, this approach seems to be unable to scale with model size and more diverse data.

	- **(W4) Homophilic data:** The model struggles with homophilic data (Figure S6). While I think that this is not a big problem since the model works well on heterophilic data, the authors should more directly state that in the main paper. (This is not clear enough about the reslts in the appendix: _"Results show that models generally achieve better results on heterophilic test graphs than homophilic ones ..."_)



- **Minor Weaknesses**
	- While Figure looks visually appealing, it is not helpful in understanding the architecture.

**Questions:**

- See weaknesses.
- What does "original split" in Table 1 mean?


**Overall,** while I think that this paper makes some good contributions there two primary things that need to be addressed. (W1) The writing makes it difficult to understand the architectural advances. (W3) The scaling results seem to indicate a fundamental limitation of this approach. Overall, I am voting 4 - marginally below acceptance threshold.

---

> ### Author Response · Authors · 2025-11-20
> **Response to Reviewer nQLH, Part 1/2**
>
> Thank you very much for your thoughtful suggestions that improved our manuscript. We hope our response can adequately address your concerns. The discussion and new results have been added to the revision.
>
> **W1**: Clarity of definitions.
>
> **Response**: Thank you for your suggestions. We have revised Section 3 following your guidance.
> - $S_\mathtt{src}$ and $S_\mathtt{tgt}$ are the decomposition results of the original weight matrix.
>     - Difference with traditional methods: Take the input module as an example. In a standard input module, both would indeed be learnable weight matrices. In our unified input module, however, $S_\mathtt{src}$ `is generated by a parametric function rather than directly learned`.
>     - Why we call them semantic matrices: Each row of $S_\mathtt{src}$/$S_\mathtt{tgt}$ corresponds to a channel of the input feature or hidden representation. Thus, these rows can be interpreted as channel-level semantic descriptors.
>     - Why $S_\mathtt{src}$ cannot be a learnable weight matrix: In traditional I/O mappings, both matrices are fully learnable, meaning the parameter count of $S_\mathtt{src}$ scales with the input dimensionality and the learned values are tailored to the specific training dataset. This design lacks transferability. To `ensure adaptability to general datasets`, we propose to model $S_\mathtt{src}$ as the output of a parametric function for the input module.
>     - [Revision]: To directly specify the relation between the linear component in the I/O mappings and $S_\mathtt{src}$/$S_\mathtt{tgt}$, we have added the original weight matrix $W$ to Eq.1,  $W_\mathtt{in}$ to Eq.4, and $W_\mathtt{out}$ to Eq.5.
> - Usage of the function $\mathtt{f_{in}}$: The target of $\mathtt{f_{in}}$ is `learning the input feature semantics` $S_\mathtt{src}^\mathtt{(in)}\in\mathbb{R}^{d_\mathtt{in}\times s}$ `to construct the input mapping` as $S_\mathtt{src}^\mathtt{(in)}S_\mathtt{tgt}^{\mathtt{(in)}\top}$, `not the node embeddings`. The input mapping is employed to project the input features into hidden representations, which can be regarded as node embeddings.
>     - [Revision]: To directly specify the usage of $\mathtt{f_{in}}$, we have added "Based on Eq.1, the linear input mapping can be formulated as $W_\mathtt{in}=S_\mathtt{src}^\mathtt{(in)}S_\mathtt{src}^{\mathtt{(in)}\top}$." at the beginning of Section 3.3.
> - Differences between our unified input module and Deep Sets:
>     - Different problems: Deep Sets solves single-level set learning problems, while unifying inputs `involves a bi-level set learning problem`: learning channel-set semantics and learning node-set representation for each channel.
>     - Different conditions: Deep Sets does not decouple parameter counts from input dimensionality. In the channel-level set learning, however, nodes constitute input dimensions. As the number of nodes varies across datasets, `the input module must keep its parameters independent of the node count. Our design explicitly satisfies this requirement`.
>     - Different set mixer: Deep Sets applies sum or max pooling to mix the set, while ours `follows the linear attention to construct the set mixer`, preserving richer relational information among channels.
>     - [Revision]: To specify the differences between our input module and Deep Sets, we have added the above discussion in Appendix E.2.
>
> **W2**: What does Fig.5 (revision Fig.7) visualize?
>
> **Response**:
> - Figure interpretation: Fig.7 shows the gap of model performance pre-trained on datasets with the same domain/heterophily as the test set versus those with different domains/heterophily (Gap=Same-Diff). The x-axis label should be "test dataset". We have fixed it. Boxplot is employed to show these gaps because it reveals the full distribution shape compared to a scatter plot or reporting average results, making it easier to assess whether the gap is generally positive. The components in the boxplot are:
>     - Center line (Median): The middle performance gap of all runs, reflecting the typical performance gap.
>     - Box (25th–75th percentile): The interquartile range (IQR), showing where the central 50\% of performance gaps lie.
>     - Whiskers (minimum and maximum): They extend to the minimum and maximum performance gap within 1.5*IQR, depicting the overall spread of non-outlier results.
> - Conclusion: Our focus is to evaluate whether the gap is positive in general. Across all datasets, `the boxes lie predominantly above zero`, indicating that pre-training on datasets with `matching domains/heterophily as the test set consistently yields better performance`.
> - How big is the performance gap? From the range of the boxes and whiskers, most gaps fall within approximately 0\%–1.5\%.

---

> ### Author Response · Authors · 2025-11-20
> **Response to Reviewer nQLH, Part 2/2**
>
> **W3-1**: The impact of model size on the best-performing models, such as GCN or MixHop.
>
> **Response**:
> - `The average diminishes the impact`.
>     - The results in Fig.3 (revision Fig.5) are computed by averaging over different hidden dimensions when examining the effect of layer depth, and by averaging over different layer depths when examining the effect of hidden dimension. The average computation diminishes the overall impact.
>     - To avoid this, we directly compute the increase under different architectural configurations. Results in Tab.R1 show that
>         - Stacking layers improves the model performance by at least 3 points (3\%)
>         - Expanding hidden dimensions improves performance by 5 points (5\%).
> - `Scaling with pre-training achieves more significant gains than supervised learning`: We also explored the impact of model size on supervised models. Results in Tab.R1 show that scaling the size of pre-training models gains more performance improvement than supervised models.
>
> |**Tab.R1** Average model performance gains (\%) |\#Layers |\#Hidden dimensions |
> |:-- |--: |--: |
> |GCN (pre-train) |3.11 |6.02 |
> |GAT (pre-train) |3.78 |6.38 |
> |GraphSAGE (pre-train) |3.42 |**7.80** |
> |MixHop (pre-train) |**7.72** |5.24 |
> |GCN (supervised) |2.48 |4.44 |
> |GAT (supervised) |3.09 |4.50 |
>
> **W3-2**: The impact of the number of nodes/datasets on test performance.
>
> **Response**:
>
> Number of datasets:
> - `Comparison criterion requires a fix`. Based on your question, we analyzed the comparison criterion in Fig.4 (revision Fig.S10(d)) and found that, except for the average number of nodes, the **dataset with the minimum dataset size** also potentially affects the performance:
>     - Does the minimum dataset size affect model performance? To explore this, we compared model pre-training with different combinations of three datasets.
>         - Fixing the minimum and increasing the average: [Fig.a in this link](https://anonymous.4open.science/r/Unified-IO-F951/scaling_datasets.md) shows that `simply adding larger datasets while keeping the smallest size unchanged is not an effective way` to expand the training set.
>         - Fixing the maximum and increasing the minimum: [Fig.b in this link](https://anonymous.4open.science/r/Unified-IO-F951/scaling_datasets.md) shows that when the dataset with the maximum dataset sizes remains, `raising the minimum dataset sizes yields improved performance`.
>         - Together, these findings suggest that mixing datasets with widely varying node scales is inefficient; `datasets should instead be chosen to maintain a similar node scale`.
>     - `Increasing training datasets with similar node scale benefits performance`:
>         - Setting: Based on the above observation, we propose to compare different dataset combinations with similar node scale. Training datasets are split into three groups with an average number of nodes around 1k, 10k, and 100k. Model performance is then normalized with the group average performance.
>         - Results: [Fig.c in this link](https://anonymous.4open.science/r/Unified-IO-F951/scaling_datasets.md) shows that either increasing the minimum data sizes or increasing training datasets with the same node scale can benefit model performance. For example, if the average performance is around 70\%, an improvement of 0.03 corresponds to an absolute increase of roughly 2 points (2\%).
>     - [Revision]: The new results have been included in the revision Fig.6 and D.3.
>
> Number of nodes: Based on the results in [Fig.c in this link](https://anonymous.4open.science/r/Unified-IO-F951/scaling_datasets.md), increasing the minimum number of nodes from 1k to 1M yields an improvement of 0.04, corresponding to an absolute increase of 3 points (3\%).
>
> **W4**: Directly state the conclusion for Fig.S6 (revision S11) in the main paper.
>
> **Response**: Thank you for your suggestion. We have added a more detailed description in the revision L474-478 as "Results in Fig.S11(a) show that models pre-trained on either homophilic or heterophilic graphs gain better results on heterophilic graphs than homophilic graphs. This can be attributed to the better transferability of the node feature knowledge learned by our Unified I/O than the structural knowledge (Fig.S11(d)), where node feature knowledge better benefits the adaptation to heterophilic graphs (Fig.S11(b)) and structural knowledge benefits homophilic tasks (Fig.S11(c))."
>
> **Minor W**: Revise the figure to help understand the architecture.
>
> **Response**: Thank you for your suggestion. We have added new figures in the revision Fig.3 and 4. Please check [the pipeline for input and output modules in this link](https://anonymous.4open.science/r/Unified-IO-F951/module_pipeline.md).
>
> **Q**: What does "original split" in Tab.1 mean?
>
> **Response**: The "original split" denotes the split setups in the original papers, which are under the supervised setting. We have added the detailed configuration in the revision Tab.S4 and S5.

---

> > ### Comment · Reviewer_nQLH · 2025-11-24
> >
> > **(W1)** I thank the authors for their explanations and now understand the proposed I/O modules. Unfortunately, the explanations in the paper are not as clear as what you have written here. The definition of the I module (and the explanation of how it is used) remain dense and difficult to understand. I believe that this section (and maybe also Section 3.2) needs a significant rewrite before it can be published.
> >
> > **(W2)** Thank you. It is now clear to me that these figures demonstrate this performance gap. Unfortunately, with `0 - 1.5%` the average performance gap is rather small.
> >
> > **(W3-1)** Your explanation makes sense to me. While your new measurement seems interesting, I am a bit confused what Tab.R1 shows. For every model there are multiple #layers (or #hidden-dim) tried, how did you average this into a single number? An alternative hypothesis: Could it just be that the number of parameters is more important than depth / width alone (at least for the strong models)?
> >
> > Due to the continuing issues with clarity of the submission (W1), I choose to keep my score.

---

> ### Author Response · Authors · 2025-11-24
> **Follow-up Response to Reviewer nQLH**
>
> Thank you very much for your feedback. We hope the following response can address your remaining concerns. If the reviewer's concerns are related to other points, we are standing by to address them.
>
> > The anonymous GitHub website is currently affected by server instability. If you encounter difficulty accessing it, we kindly refer you to the corresponding materials in our revised manuscript. We apologize for the inconvenience.
>
> **W1** Writing of section 3.3 (and section 3.2)
>
> **Response**: Thank you for your comment. We understand your concern that Section 3.3 (and part of Section 3.2) appeared dense in the initial submission. At the same time, we appreciate that you found our explanations in the rebuttal clear and helpful.
>
> Importantly, every point presented in our rebuttal directly corresponds to content already present in the manuscript. Therefore, instead of a significant rewrite, we have explicitly signposted the same technical content in the revision to make the logic easier to follow. Specifically:
> - Section 3.2:
>     - $S_\mathtt{src}$ and $S_\mathtt{tgt}$ as the decomposition of the original weight matrix: Revision v2 L176-177.
>         - Differences from traditional methods: L190-192 subtitled as "Our Solution" in revision v2.
>         - Why they are called semantic matrices: L176-182 subtitled as "Decomposition as Semantics" in revision v2.
>         - Why $S_\mathtt{src}$ cannot serve as a learnable weight matrix: L183-189 subtitled as "Problems in Traditional Solutions" in revision v2.
> - Section 3.3:
>     - Purpose of the function $\mathtt{f_{in}}$: L198-206 have directly clarified that its goal is to learn $S_\mathtt{src}$ for constructing $W_\mathtt{in}$.
>     - Differences between our unified input module and Deep Sets: Highlighted in revision v2 L223-225 with a reference to Appendix E.2.
>     - Besides the above two points you raised in the review, we have added the definition of the input module and the explanation of how it is used in revision v2 L196-198. The final version of the input module is provided in Eq.4.
>
> These edits align the manuscript’s presentation with the clear explanations given in the rebuttal, ensuring readability without requiring substantial rewriting.
>
> **W2**: Average performance gap in Fig.5 (Fig.7 in v1) is rather small.
>
> **Response**: The purpose of Fig.7 is to examine whether the consistency of domain/heterophily between the training and test datasets affects the performance of pre-trained models. The results in Fig. 7 exhibit an overall positive performance gap, demonstrating the `consistent presence of such an influence` across diverse datasets. While the absolute magnitude of the gap is indeed small, this does not negate the effect; it simply indicates that the domain/heterophily mismatch acts as `a moderate but persistent factor`.
>
> **W3-1 (1)**: For every model, there are multiple #layers (or #hidden-dim) tried, how did you average this into a single number?
>
> **Response**: For the influence of layers, the performance gain is computed by
> - 1. Fixing the hidden dimensions and increasing the number of layers to compute the largest gain.
> - 2. Averaging the gains across different hidden dimensions.
>
> The influence of hidden dimensions is similar by first fixing the layers and then averaging across layers.
>
> **W3-1 (2)**: Could it just be that the number of parameters is more important than depth/width alone (at least for the strong models)?
>
> **Response**: Thank you for your suggestion! We have already compared the performance of pre-trained models with different numbers of parameters in Appendix D.2 and Fig.S8. You can also see the results in [Fig.a in this link](https://anonymous.4open.science/r/Unified-IO-F951/scaling_parameters.md). Results show that `model capacity cannot be assessed solely through parameter volume`. Instead, layer depth must be chosen appropriately for each architecture, as simply increasing parameters by stacking layers does not always yield better performance.

---

### Official Review · Reviewer_fnYM · 2025-10-23

**Soundness:** 2
**Presentation:** 3
**Contribution:** 2
**Rating:** 2
**Confidence:** 3

**Summary:**

This paper explores whether pure graph neural networks can be pre-trained across diverse graph datasets without relying on language models. The authors propose unified input and output modules that decouple model parameters from dataset-specific feature and label spaces—using a shared relational function for input features and uniformly sampled pseudo-labels for outputs. This allows GNNs to generalize across datasets with different semantics. Experiments show that the proposed method enables effective cross-dataset pre-training, achieving competitive or superior results to supervised and LM-based baselines while simplifying fine-tuning and hyperparameter tuning.

**Strengths:**

1.	The paper is well-organized and easy to follow. It provides clear motivation, formal problem formulation, and theoretical derivations that make the proposed Unified I/O framework convincing and conceptually coherent.
2.	The paper tackles a highly relevant and underexplored problem—how to unify diverse graph datasets with different input and output spaces. This direction has strong practical significance for building more generalizable graph foundation models.

**Weaknesses:**

1.	The proposed method is only demonstrated on node classification tasks, while other settings such as node regression and graph-level tasks (where graph pre-training is often most impactful) are not explored. This narrow scope limits the general applicability and practical influence of the framework.
2.	Although the experiments are extensive, the performance of Unified I/O is not consistently competitive. In many settings, it falls notably behind the best existing methods. Moreover, several chosen baselines are relatively weak — for example, on the Cora original split, Unified I/O achieves 82.32%, yet many self-supervised approaches surpass this level by a clear margin. This undermines the strength of the empirical claims.
3.	The results in Figure 4 are disappointing — increasing the number of training datasets yields almost no improvement, which questions the necessity of such pre-training compared with simply performing self-supervised learning on a single dataset. In Figure 3, the claim that performance keeps improving with larger hidden dimensions or deeper layers is counter-intuitive; the curves flatten toward the end, suggesting the authors may not have reached the turning point where over-parameterization degrades results. If not, additional evidence is needed. Moreover, in scaling analysis, it would be more appropriate to examine total model parameter count, as commonly done in LLM research, rather than only hidden dimension or layer depth.
4.	The output module and final loss design of Unified I/O resemble a clustering process, raising concerns about convergence stability. On complex datasets, if the initial parameters are far from optimal, it is unclear whether the model can still converge to a good solution. This also raises potential cold-start issues during pre-training, as the model may lack meaningful gradient signals in the early stage.

**Questions:**

Please refer to the Weaknesses section above.

---

> ### Author Response · Authors · 2025-11-20
> **Response to Reviewer fnYM, Part 1/4**
>
> We greatly appreciate your valuable suggestions that help us revise our manuscript, and hope our response can adequately address your concerns. The discussion and new results have been added to the revision.
>
> **W1**: Evaluation on more tasks.
>
> **Response**: Thank you for your suggestion, which helps us to evaluate our Unified I/O more comprehensively.
> - Settings: GNNs are pre-trained on graph-level datasets (COLLAB, IMDB-BINARY, MUTAG, and D&D) [R1], and evaluated on graph-level tasks (PROTEINS, REDDIT-BINARY, ENZYMES) [R1] and node-level tasks (AmazonComputers, Cora, arxiv-year, twitch-gamer). The experimental setup is kept the same as in Tab.1. Baselines include supervised method GCN, self-supervised learning method GraphCL [R2], parameter-free feature alignment methods (SVD, Laplacian projection, and random projection) combined with our unified output module. We also adopt FUG [R3] and GraphAny as baselines for node-level tasks. Neither methods support graph tasks and are thus pre-trained on (amazon-ratings, ogbn-arxiv, Facebook, and roman-empire).
> - Results in Tab.R1 show that
>     - Compared with GraphCL, SVD, Lap, and rand, pre-training with Unified I/O `achieves better performance on both graph and node tasks`.
>     - Compared with GCN, pre-training with Unified I/O `achieves comparable or even better performance on graph tasks`.
>     - `Transferring between graph and node tasks introduces a noticeable performance gap in both directions`. In particular, for node tasks, models pre-trained on graph-level datasets perform competitively on the original split but lag behind under the 1/3-shot settings when compared with models pre-trained directly on node-level datasets.
> - [Revision]: The new results have been included in the revision Appendix D.1.
>
> |**Tab.R1** Pre-training with graph tasks (\%) |ENZYMES |REDDIT-BINARY |PROTEINS |AmazonComputers |CORA |arxiv-year |twitch-gamer |
> |:-- |:--: |:--: |:--: |:--: |:--: |:--: |:--: |
> |||||original split ||||
> |GCN (sup) |37.41$\pm$0.32 |**78.40$\pm$0.41** |71.70$\pm$0.39 |91.09$\pm$0.13 |81.80$\pm$0.28 |48.03$\pm$0.41 |59.44$\pm$0.18 |
> |GraphCL |19.77$\pm$0.41 |71.26$\pm$0.32 |67.23$\pm$1.01 |88.67$\pm$0.48 |61.93$\pm$1.47 |OOM |OOM |
> |Lap |52.38$\pm$0.37 |72.04$\pm$0.71 |70.07$\pm$0.52 |78.69$\pm$0.76 |76.80$\pm$0.46 |40.86$\pm$0.64 |56.25$\pm$0.50 |
> |Rand |18.33$\pm$0.37 |52.20$\pm$0.43 |59.67$\pm$0.73 |34.73$\pm$0.68 |15.50$\pm$0.71 |37.00$\pm$0.76 |52.29$\pm$0.44 |
> |SVD |53.65$\pm$0.54 |71.08$\pm$0.54 |70.89$\pm$0.69 |75.67$\pm$0.49 |73.50$\pm$0.66 |40.57$\pm$0.55 |56.82$\pm$0.52 |
> |FUG |- |- |- |88.22$\pm$0.09 |30.70$\pm$0.96 |42.54$\pm$0.30 |58.16$\pm$0.27 |
> |GraphAny |- |- |- |82.94$\pm$0.82 |79.41$\pm$0.35 |38.36$\pm$0.53 |59.96$\pm$0.02 |
> |Unified I/O (node) |52.83$\pm$1.06 |69.65$\pm$0.36 |70.73$\pm$0.26 |89.85$\pm$0.18 |82.32$\pm$0.97 |42.58$\pm$0.17 |59.99$\pm$0.05 |
> |Unified I/O (graph) |**54.88$\pm$0.40** |73.09$\pm$0.43 |**72.45$\pm$0.44** |80.29$\pm$0.55 |80.30$\pm$0.70 |41.35$\pm$0.43 |57.73$\pm$0.66 |
> |||||1-shot ||||
> |GraphCL |16.71$\pm$0.69 |52.85$\pm$1.53 |51.89$\pm$3.68 |41.39$\pm$2.64 |27.83$\pm$1.20 |OOM |OOM |
> |Lap |17.33$\pm$2.46 |56.88$\pm$1.66 |55.62$\pm$1.91 |45.71$\pm$2.18 |22.80$\pm$2.00 |28.57$\pm$1.90 |53.45$\pm$1.88 |
> |Rand |17.68$\pm$2.06 |55.36$\pm$1.86 |49.56$\pm$1.56 |15.67$\pm$1.78 |16.20$\pm$2.26 |26.61$\pm$1.93 |52.56$\pm$2.16 |
> |SVD |17.87$\pm$1.37 |56.80$\pm$1.35 |55.47$\pm$1.60 |43.46$\pm$1.27 |18.90$\pm$2.44 |24.78$\pm$1.37 |53.45$\pm$1.17 |
> |FUG |- |- |- |27.26$\pm$0.36 |41.83$\pm$0.26 |27.58$\pm$0.11 |49.93$\pm$0.01 |
> |GraphAny |- |- |- |62.87$\pm$0.29 |53.63$\pm$1.03 |25.03$\pm$0.48 |49.65$\pm$0.48 |
> |Unified I/O (node) |17.82$\pm$1.27 |48.21$\pm$1.34 |52.27$\pm$1.75 |59.89$\pm$0.80 |43.94$\pm$0.50 |33.47$\pm$0.14 |57.90$\pm$0.15 |
> |Unified I/O (graph) |**18.08$\pm$1.82** |**58.89$\pm$2.40** |**56.07$\pm$0.81** |60.33$\pm$2.16 |25.90$\pm$0.92 |29.96$\pm$0.95 |54.60$\pm$1.47 |
> |||||3-shot ||||
> |GraphCL |18.16$\pm$0.88 |57.13$\pm$1.46 |53.73$\pm$0.82 |55.41$\pm$1.41 |34.97$\pm$0.64 |OOM |OOM |
> |Lap |20.42$\pm$1.29 |58.23$\pm$1.23 |58.60$\pm$1.26 |45.35$\pm$1.10 |34.80$\pm$1.39 |24.62$\pm$1.35 |54.51$\pm$1.26 |
> |Rand |18.62$\pm$1.28 |56.30$\pm$1.16 |51.38$\pm$1.16 |19.02$\pm$1.31 |30.70$\pm$1.37 |25.64$\pm$1.09 |53.04$\pm$1.43 |
> |SVD |20.67$\pm$1.10 |58.28$\pm$1.16 |55.95$\pm$1.35 |50.76$\pm$1.36 |35.60$\pm$1.26 |21.62$\pm$1.23 |53.26$\pm$1.13 |
> |FUG |- |- |- |50.59$\pm$0.29 |47.77$\pm$0.29 |24.02$\pm$0.19 |49.83$\pm$0.09 |
> |GraphAny |- |- |- |70.04$\pm$1.43 |66.32$\pm$1.21 |24.74$\pm$0.34 |54.71$\pm$0.18 |
> |Unified I/O (node) |19.00$\pm$1.93 |49.07$\pm$1.68 |54.46$\pm$1.36 |68.33$\pm$0.28 |49.21$\pm$0.91 |35.32$\pm$0.29 |57.64$\pm$0.06 |
> |Unified I/O (graph) |**22.05$\pm$2.00** |**59.32$\pm$0.93** |**60.46$\pm$1.57** |65.78$\pm$2.27 |38.90$\pm$1.17 |23.97$\pm$1.48 |55.56$\pm$1.43 |

---

> ### Author Response · Authors · 2025-11-20
> **Response to Reviewer fnYM, Part 2/4**
>
> **W2**: Performance gains and more self-supervised baselines.
>
> **Response**:
> - Performance comparison with existing baselines:
>     - The average rank shows that `Unified I/O achieves the best overall performance`, although it falls behind the best baselines (GraphAny and ZeroG) under certain settings
>     - Compared to GraphAny, which outperforms our method on certain homophilic settings
>         - Unified I/O `achieves better performance on heterophilic tasks`.
>         - Unified I/O can `provide graph representations and make zero-shot predictions without observed labels`. In contrast, GraphAny depends on observed labels to make predictions and cannot provide node representations, as it directly computes the mapping from the inputs to the labels.
>     - Compared to ZeroG, which outperforms our method on certain zero-shot tasks
>         - Unified I/O `can be applied to any type of dataset.` In contrast, ZeroG is restricted to graphs with rich textual attributes.
>         - Unified I/O `only involves the knowledge of 4 datasets during pre-training`, including PubMed, bookhistory, amazon-ratings, and arxiv. In contrast, ZeroG with LM introduces knowledge gained from enormous training data for graph learning.
> - Comparison with self-supervised methods:
>     - Training-free setting
>         - Setting: To ensure fair comparison, we follow GraphAny to solve the mapping matrix from the learned self-supervised representations to the labels in a training-free manner. Both contrastive-based methods (DGI, GraphCL [R2], GraphACL, GRACE, and SimGRACE [R4]) and reconstruction-based methods (MaskGAE [R5] and GraphMAE2 [R6]) are implemented. We reported the best-performing results as SSL.
>         - Results: Tab.R2 shows that models pre-trained with our Unified I/O can `consistently outperform the best-performing self-supervised baselines` under the training-free setting.
>     - Fine-tuning setting: Unified I/O has been compared with DGL, GRACE, and GraphACL in Tab.3 in the initial submission. We further incorporate more baselines in Tab.R3. Results show that fine-tuning the I/O modules pre-trained with Unified I/O on downstream tasks can `achieve better performance` compared to self-supervised baselines.
>     - [Revision]: The new results have been included in the revision Tab.1 and 3.
>
> |**Tab.R2** Training-free inference setting with SSL (\%) |AmazonComputers |CORA |CoauthorPhysics |arxiv-year |twitch-gamer |tolokers |CHAMELEON |ACTOR |
> |:-- |:--: |:--: |:--: |:--: |:--: |:--: |:--: |:--: |
> |||||original split |||||
> |SSL |89.28$\pm$0.34 |81.50$\pm$0.66 |92.32$\pm$0.08 |41.63$\pm$0.27 |59.19$\pm$0.05 |75.92$\pm$0.18 |61.01$\pm$0.72 |27.61$\pm$0.27 |
> |Unified I/O |**89.85$\pm$0.18** |**82.32$\pm$0.97** |**92.85$\pm$0.48** |**42.58$\pm$0.17** |**59.99$\pm$0.05** |**76.44$\pm$0.44** |**62.13$\pm$0.43** |**35.35$\pm$0.26** |
> |||||1-shot |||||
> |SSL |55.67$\pm$0.36 |42.80$\pm$0.35 |77.86$\pm$0.12 |29.76$\pm$0.12 |57.11$\pm$0.05 |67.82$\pm$1.02 |25.07$\pm$2.08 |20.72$\pm$0.89 |
> |Unified I/O |**59.89$\pm$0.80** |**43.94$\pm$0.50** |**85.13$\pm$0.68** |**33.47$\pm$0.14** |**57.90$\pm$0.15** |**68.51$\pm$0.13** |**32.00$\pm$0.11** |**25.69$\pm$0.12** |
> |||||3-shot |||||
> |SSL |64.51$\pm$2.68 |48.73$\pm$1.86 |84.22$\pm$0.12 |26.53$\pm$1.11 |56.78$\pm$0.16 |59.24$\pm$2.24 |31.99$\pm$0.37 |20.42$\pm$0.15 |
> |Unified I/O |**68.33$\pm$0.28** |**49.21$\pm$0.91** |**84.68$\pm$0.67** |**35.32$\pm$0.29** |**57.64$\pm$0.06** |**71.77$\pm$0.25** |**33.76$\pm$0.11** |**25.20$\pm$0.11** |
>
> |**Tab.R3** Fine-tuning setting with SSL (\%) |Cora |PubMed |AmazonComputers |WikiCS |amazon-ratings |minesweeper |tolokers |
> |:-- |:--: |:--: |:--: |:--: |:--: |:--: |:--: |
> |DGI |84.00$\pm$0.28 |83.73$\pm$0.41 |82.11$\pm$0.16 |75.05$\pm$0.39 |40.80$\pm$0.74 |88.45$\pm$0.38 |77.73$\pm$0.14 |
> |GRACE |84.30$\pm$0.57 |85.81$\pm$0.18 |89.67$\pm$0.36 |75.8$\pm$0.56 |42.19$\pm$0.12 |86.15$\pm$0.45 |75.06$\pm$0.14 |
> |GraphACL |75.00$\pm$0.75 |82.93$\pm$0.17 |80.58$\pm$0.15 |68.00$\pm$0.75 |40.65$\pm$0.14 |87.23$\pm$0.11 |77.68$\pm$0.25 |
> |GraphCL |63.33$\pm$1.76 |63.53$\pm$1.08 |84.57$\pm$0.34 |76.32$\pm$0.18 |42.35$\pm$0.37 |79.85$\pm$0.12 |80.03$\pm$0.12 |
> |MaskGAE |75.13$\pm$1.78 |75.27$\pm$1.03 |92.15$\pm$0.05 |78.25$\pm$0.20 |43.54$\pm$0.30 |84.33$\pm$0.16 |81.13$\pm$0.34 |
> |SimGRACE |67.07$\pm$0.82 |77.63$\pm$0.82 |87.44$\pm$0.18 |78.75$\pm$0.28 |43.46$\pm$0.23 |84.23$\pm$0.16 |80.29$\pm$0.30 |
> |GraphMAE2 |79.50$\pm$0.51 |67.07$\pm$0.91 |91.03$\pm$0.19 |76.24$\pm$0.11 |40.95$\pm$0.71 |80.16$\pm$0.10 |80.17$\pm$0.07 |
> |Unified I/O |**84.63$\pm$0.12** |**88.91$\pm$0.45** |**92.33$\pm$0.25** |**78.98$\pm$0.12** |**51.59$\pm$0.05** |**91.39$\pm$0.15** |**83.29$\pm$0.13** |

---

> ### Author Response · Authors · 2025-11-20
> **Response to Reviewer fnYM, Part 3/4**
>
> **W3-1**: Does increasing the number of training datasets benefit pre-training?
>
> **Response**:
>
> Why does increasing training datasets benefit?
> - `Comparison criterion requires a fix`. Based on your question, we analyzed the comparison criterion in Fig.4 (initial submission, revision Fig.S10(d)) and found that, except for the average number of nodes, the **dataset with the minimum dataset size** also potentially affects the performance:
>     - Does the minimum dataset size affect model performance? To explore this, we compared model pre-training with different combinations of three datasets.
>         - Fixing the minimum and increasing the average: [Fig.a in this link](https://anonymous.4open.science/r/Unified-IO-F951/scaling_datasets.md) shows that `simply adding larger datasets while keeping the smallest size unchanged is not an effective way` to expand the training set.
>         - Fixing the maximum and increasing the minimum: [Fig.b in this link](https://anonymous.4open.science/r/Unified-IO-F951/scaling_datasets.md) shows that when the dataset with the maximum dataset sizes remains, `raising the minimum dataset sizes yields improved performance`.
>         - Together, these findings suggest that mixing datasets with widely varying node scales is inefficient; `datasets should instead be chosen to maintain a similar node scale`.
>     - `Increasing training datasets with similar node scale benefits performance`:
>         - Setting: Based on the above observation, we propose to compare different dataset combinations with similar node scale. Training datasets are split into three groups with an average number of nodes around 1k, 10k, and 100k. Model performance is then normalized with the group average performance.
>         - Results: [Fig.c in this link](https://anonymous.4open.science/r/Unified-IO-F951/scaling_datasets.md) shows that either increasing the minimum data sizes or increasing training datasets with the same node scale can benefit model performance.
>     - [Revision]: The new results have been included in the revision Fig.6 and D.3.
>
> Is pre-training with our Unified I/O better than self-supervised learning?
> - Results in Tab.R2 and R3 show that pre-training with our Unified I/O can achieve `better performance` than the self-supervised baselines in both training-free inference and fine-tuning settings.
>
> **W3-2**: Exploring configurations with larger parameter volumes.
>
> **Response**: Thank you very much for your suggestions, which have helped us to provide more rigorous conclusions for the parameter scaling test.
> - More layers and hidden dimensions: Following your suggestions, we conducted additional pre-training experiments with hidden dimensions of {2048, 4096} and depths of {20, 32} layers. The results (see [Fig.b in this link](https://anonymous.4open.science/r/Unified-IO-F951/scaling_parameters.md)) show that:
>     - `Further scaling the number of layers causes over-parameterization` (connected dots). This can be alleviated by expanding training datasets, where results in [Fig.c in this link](https://anonymous.4open.science/r/Unified-IO-F951/scaling_parameters.md) indicate that more training datasets can better support parameter scaling.
>     - `Increasing the number of hidden dimensions to 4096 does not show over-parameterization`. This observation further supports the conclusion that scaling hidden dimensionality is a more stable and beneficial strategy for improving pre-trained GNN models.
>     - Due to GPU memory limitations, we are currently unable to investigate even larger parameter volumes, but we plan to explore more efficient parameter scaling techniques in future work.
> - Compared with the number of parameters: Thank you for your suggestion, which enables us to compare models in the same figure and avoids diminishing the overall trend caused by the average.
>     - As shown in [Fig.a in this link](https://anonymous.4open.science/r/Unified-IO-F951/scaling_parameters.md), the results still exhibit the clear influence driven by layer depth (connected dots) and hidden dimension. Notably, increasing parameter count by adding more layers does not consistently improve performance.
>     - These observations indicate that `model capacity cannot be assessed solely through parameter volume`. Instead, layer depth must be chosen appropriately for each architecture.
> - [Revision]: Due to the inability to show the impact of scaling layers on $N^2$, we keep the initial figure in the main context and note the existence of over-parameterization in Section 4.2 L415-419. New results have been included in the revision Appendix D.2.

---

> ### Author Response · Authors · 2025-11-20
> **Response to Reviewer fnYM, Part 4/4**
>
> **W4**: Does the output module and final loss design of Unified I/O cause unstable training?
>
> **Response**: No, the training of Unified I/O is stable.
> - Empirical justifications
>     - The training loss curves with different random seeds in [this anonymized repository](https://anonymous.4open.science/r/Unified-IO-F951/fig/loss.png) are `overall smooth and decrease stably across epochs`.
>     - We have reported the standard deviation under different random seeds in the initial submission. Results show that the model can consistently converge to a good solution.
> - Why is training stable:
>     - `Supervised objective provides informative early gradients`. Our supervised contrastive objective provides class-level supervision from the start, ensuring that early gradients are informative rather than random. This direct enforcement of intra-class coherence and inter-class separation avoids the cold-start issue common in self-supervised contrastive learning. Consequently, our training converges stably even from less optimal initialization.
>     - `Targets remain invariant during training`. Unified I/O makes predictions by adjusting the similarity between node representations and pseudo-label semantics. Because pseudo labels are uniformly selected in the semantics space and remain invariant throughout training, the model can be optimized against consistent and stable targets.
>
> [R1] TUDataset: A collection of benchmark datasets for learning with graphs. ICMLWorkshop'20
>
> [R2] Graph contrastive learning with augmentations. NeurIPS'20
>
> [R3] FUG: Feature-Universal Graph Contrastive Pre-training for Graphs with Diverse Node Features. NeurIPS'24
>
> [R4] SimGRACE: A Simple Framework for Graph Contrastive Learning without Data Augmentation. WWW'22
>
> [R5] What's Behind the Mask: Understanding Masked Graph Modeling for Graph Autoencoders. KDD'23
>
> [R6] GraphMAE2: A Decoding-Enhanced Masked Self-Supervised Graph Learner. KDD'23

---

> > ### Comment · Reviewer_fnYM · 2025-11-24
> >
> > I sincerely appreciate the authors’ thoughtful response, and I have carefully read all the reviewers’ comments. Regarding W1, based on the additional experimental results provided by the authors, Unified I/O does not demonstrate a significant performance advantage. Regarding W3, the newly added conclusion remains ambiguous — in my view, the discussion on the minimum dataset size is limited, and it is still unclear how the scaling law of a pre-trained model is reflected. In recognition of the authors’ continued efforts, I am willing to raise my score slightly; however, I believe the current version has not yet reached my threshold for acceptance.

---

> ### Author Response · Authors · 2025-11-24
> **Follow-up Response to Reviewer fnYM**
>
> Thank you very much for raising your score! We hope the following response can address your remaining concerns. If there are any additional points that the reviewer would like us to explore, we are standing by to address them.
>
> > The anonymous GitHub website is currently affected by server instability. If you encounter difficulty accessing it, we kindly refer you to the corresponding materials in our revised manuscript. We apologize for the inconvenience.
>
> **W1**: Significance of the performance gains.
>
> **Response**: To demonstrate this, we conducted statistical significance tests on the performance gains in Tab.R1. Our method is compared against baselines that also rely on graph data for pre-training (GraphCL for graph-level tasks; SVD, Lap, and Rand for both task types):
> - Graph-level tasks: The largest p-value across all comparisons is 0.003906, far below the 0.05 threshold, `confirming statistical significance`.
> - Node-level tasks: The largest p-value across all comparisons is 0.000183, also well below 0.05, again `indicating statistical significance`.
>
> We note that our intention is `not to outperform baselines pre-trained with node data on node-level tasks`. The node-data-based results in Tab.R1 serve to illustrate that cross-task transfer (graph->node or node->graph) introduces a clear performance gap in both directions. Importantly, Unified I/O consistently surpasses pre-trained baselines in the matched settings of "node pre-train, node test" and "graph pre-train, graph test".
>
> **W3**: Discussion on the minimum dataset size and the corresponding scaling law.
>
> **Response**: The newly added experiments clarify the scaling behavior of Unified I/O by giving two concrete conclusions:
> - `Scaling should be measured regarding the minimum dataset size`.
>     - The qualitative metrics for dataset size primarily include the minimum size, maximum size, and the average number of nodes.
>         - The new results examine their respective influences and show that the minimum dataset size, rather than the average number of nodes, governs the scaling behavior of the pre-trained model. This finding corrects the earlier criterion used in the initial submission. (Appendix D.3 or [Fig.a and b in this link](https://anonymous.4open.science/r/Unified-IO-F951/scaling_datasets.md))
>     - [New results]: We further provide the results by fixing the average and increasing the minimum. Results in Fig.S10(c) or [Fig.d in this link](https://anonymous.4open.science/r/Unified-IO-F951/scaling_datasets.md) further demonstrate that `raising the minimum dataset sizes yields improved performance`.
> - `Model performance exhibits a clear scaling trend`. As the minimum dataset size increases, the pre-trained model’s performance improves in a manner that is approximately linear with respect to the logarithm of the minimum data size. This provides a concrete characterization of the scaling law for Unified I/O. (Section 4.2 or [Fig.c in this link](https://anonymous.4open.science/r/Unified-IO-F951/scaling_datasets.md))

---

### Official Review · Reviewer_RULu · 2025-10-28

**Soundness:** 3
**Presentation:** 2
**Contribution:** 2
**Rating:** 6
**Confidence:** 4

**Summary:**

This paper proposes a unified I/O module that effectively aligns the dimensions and semantics of features and labels across diverse datasets. This design enables GNNs to be pre-trained on diverse datasets and directly adapted to unseen downstream datasets. The authors conduct extensive experiments by integrating the proposed module with a variety of GNN architectures. The experimental results demonstrate that the module can effectively unify feature and label spaces and integrate well with different GNN architectures.

**Strengths:**

S1. The proposed output module is novel, as it enables the pre-trained models to directly adapt to diverse target spaces.

S2. The proposed method demonstrates strong performance in node classification under various settings.

S3. Figure 4 reveals an interesting and counter-intuitive phenomenon that increasing the node scale of training datasets improves the model’s adaptability to test datasets, while increasing the number of training datasets may not lead to better performance.

**Weaknesses:**

W1. The proposed input module is similar to the feature unification mechanism proposed by FUG [1]. The authors should provide a more detailed discussion on the specific distinctions

W2. The explanation of why the proposed shared relation function can unify feature and label semantics mainly relies on intuition, lacking a theoretical justification or semantic alignment analysis.

W3. The downstream task only includes node classification. Although Appendix A claims that the method can be applied to general graph learning tasks, there is no experimental validation to support this claim.

W4. The compared methods are limited. How does the proposed method compare with SOTA GFMs, such as FUG [1], SAMGPT [2], RiemannGFM [3]?

[1] FUG: Feature-Universal Graph Contrastive Pre-training for Graphs with Diverse Node Features. NeurIPS' 24.

[2] SAMGPT: Text-free Graph Foundation Model for Multi-domain Pre-training and Cross-domain Adaptation. WWW' 25.

[3] RiemannGFM: Learning a Graph Foundation Model from Riemannian Geometry. WWW' 25.

**Questions:**

Apart from the weaknesses,

Q1. In some settings, the proposed method performs worse than GraphAny on homophilic graphs, but better on heterophilic graphs. As far as I know, GraphAny has designs for heterophily, while this method does not. Why does this method outperform GraphAny on heterophilic graphs?

---

> ### Author Response · Authors · 2025-11-20
> **Response to Reviewer RULu, Part 1/3**
>
> Thank you very much for your support and valuable suggestions that further broaden the potential application of our method to other tasks. We hope our response can adequately address your concerns. The discussion and new results have been added to the revision.
>
> **W1&W4**: Comparison with FUG.
>
> **Response**: Thank you for bringing this interesting work to our attention. The differences can be summarized as follows
> - Method design comparison. Different from FUG, our Unified I/O achieves
>     - Adaptive input mapping.
>         - `Unified I/O unifies semantics, then map` by employing channel relations as input to construct input mapping. The measured relations change adaptively based on the specific input while maintaining semantical consistency (see response to W2), ensuring adaptability to various datasets.
>         - `FUG directly maps` sampled node features into the input mapping. Identical feature values across different source spaces may correspond to entirely different semantics, making it nontrivial to map these values into a common space uniformly.
>     - Unified output.
>         - `Unified I/O supporting zero-shot predictions` by involving pseudo labels during pre-training.
>         - `FUG requires observed labels to make predictions` by only outputting node representations.
> - Quantitative comparison
>     - FUG: Due to the reliance of FUG on observed labels, we follow the training-free inference setting in Tab.1. Results in Tab.R1 show that our proposed Unified I/O can `consistently outperform FUG` across different test datasets, indicating its better effectiveness for pre-training.
>     - SAMGPT requires fine-tuning on the main model fusion block, which `falls out of the setting` with training-free inference in Tab.1 and zero-shot prediction in Tab.2. Thank you for bringing this interesting work to our attention. We have included it in the related work.
>     - RiemannGFM `has already been included in Tab.2`, which achieves inferior performance compared to our proposed method.
> - [Revision]: The new results and discussion have been included in the revision Tab.1, Section 2, and Appendix B.
>
> |**Tab.R1** Comparison with FUG (\%) |AmazonComputers |CORA |CoauthorPhysics |arxiv-year |twitch-gamer |tolokers |CHAMELEON |ACTOR |
> |:-- |:--: |:--: |:--: |:--: |:--: |:--: |:--: |:--: |
> |||||original split |||||
> |FUG |88.22$\pm$0.09 |30.70$\pm$0.96 |91.09$\pm$0.72 |42.54$\pm$0.30 |58.16$\pm$0.27 |75.95$\pm$0.34 |22.59$\pm$0.06 |25.53$\pm$0.11 |
> |Unified I/O |**89.85$\pm$0.18** |**82.32$\pm$0.97** |**92.85$\pm$0.48** |**42.58$\pm$0.17** |**59.99$\pm$0.05** |**76.44$\pm$0.44** |**62.13$\pm$0.43** |**35.35$\pm$0.26** |
> |||||1-shot |||||
> |FUG |27.26$\pm$0.29 |41.83$\pm$1.03 |67.70$\pm$1.35 |27.58$\pm$0.48 |49.93$\pm$0.48 |56.86$\pm$0.93 |23.39$\pm$0.36 |22.83$\pm$0.17 |
> |Unified I/O |**59.89$\pm$0.80** |**43.94$\pm$0.50** |**85.13$\pm$0.68** |**33.47$\pm$0.14** |**57.90$\pm$0.15** |**68.51$\pm$0.13** |**32.00$\pm$0.11** |**25.69$\pm$0.12** |
> |||||3-shot |||||
> |FUG |50.59$\pm$1.43 |47.77$\pm$1.21 |66.52$\pm$2.47 |24.02$\pm$0.34 |49.83$\pm$0.18 |57.93$\pm$1.17 |25.47$\pm$0.56 |20.35$\pm$0.33 |
> |Unified I/O |**68.33$\pm$0.28** |**49.21$\pm$0.91** |**84.68$\pm$0.67** |**35.32$\pm$0.29** |**57.64$\pm$0.06** |**71.77$\pm$0.25** |**33.76$\pm$0.11** |**25.20$\pm$0.11** |
>
> **W2**: Theoretical justification or empirical analysis on the unification of the feature/label semantics.
>
> **Response**: Thank you for your suggestion. We admit that further incorporating theoretical analysis can provide a more comprehensive analysis for the proposed method. Here, we provide a primary empirical analysis and will keep working on the theoretical part.
>
> Pseudo-label semantics for any dataset are uniformly selected from the space, directly empowering unification.
>
> Feature semantics for channels:
> - Setting: Textual datasets are employed for their availability of the original feature semantics. We use t-SNE to visualize feature semantics. Each dot represents a different feature channel, with dot colors representing different datasets.
> - Result: [click this link to the anonymized repository](https://anonymous.4open.science/r/Unified-IO-F951/semantics_alignment.md)
>     - Our Unified I/O: The feature semantics from different datasets form two major clusters, indicating that the learned representations are `unified into two common semantic subspaces`. Within each cluster, channels from different datasets do not fully mix but show partial segregation, suggesting that the semantic representations `still preserve dataset-specific characteristics`.
>     - Original feature semantics from different datasets form clearly separated clusters rather than merging into a shared global structure, indicating that the semantics are `not well aligned within a common semantic space`.
> - [Revision]: The new results have been included in the revision Appendix D.7.

---

> ### Author Response · Authors · 2025-11-20
> **Response to Reviewer RULu, Part 2/3**
>
> **W3**: Evaluation on more tasks.
>
> **Response**: Thank you for your suggestion, which helps us to evaluate our Unified I/O more comprehensively.
> - Settings: GNNs are pre-trained on graph-level datasets (COLLAB, IMDB-BINARY, MUTAG, and D&D) [R1], and evaluated on graph-level tasks (PROTEINS, REDDIT-BINARY, ENZYMES) [R1] and node-level tasks (AmazonComputers, Cora, arxiv-year, twitch-gamer). The experimental setup is kept the same as in Tab.1. Baselines include supervised method GCN, self-supervised learning method GraphCL [R2], parameter-free feature alignment methods (SVD, Laplacian projection, and random projection) combined with our unified output module. We also adopt FUG and GraphAny as baselines for node-level tasks. Neither methods support graph tasks and are thus pre-trained on (amazon-ratings, ogbn-arxiv, Facebook, and roman-empire).
> - Results in Tab.R2 show that
>     - Compared with GraphCL, SVD, Lap, and rand, pre-training with Unified I/O `achieves better performance on both graph and node tasks`.
>     - Compared with GCN, pre-training with Unified I/O `achieves comparable or even better performance on graph tasks`.
>     - `Transferring between graph and node tasks introduces a noticeable performance gap in both directions`. In particular, for node tasks, models pre-trained on graph-level datasets perform competitively on the original split but lag behind under the 1/3-shot settings when compared with models pre-trained directly on node-level datasets.
> - [Revision]: The new results have been included in the revision Appendix D.1.
>
> |**Tab.R2** Pre-training with graph tasks (\%) |ENZYMES |REDDIT-BINARY |PROTEINS |AmazonComputers |CORA |arxiv-year |twitch-gamer |
> |:-- |:--: |:--: |:--: |:--: |:--: |:--: |:--: |
> |||||original split ||||
> |GCN (sup) |37.41$\pm$0.32 |**78.40$\pm$0.41** |71.70$\pm$0.39 |91.09$\pm$0.13 |81.8$\pm$0.285 |48.03$\pm$0.41 |59.44$\pm$0.18 |
> |GraphCL |19.77$\pm$0.41 |71.26$\pm$0.32 |67.23$\pm$1.01 |88.67$\pm$0.48 |61.93$\pm$1.47 |OOM |OOM |
> |Lap |52.38$\pm$0.37 |72.04$\pm$0.71 |70.07$\pm$0.52 |78.69$\pm$0.76 |76.80$\pm$0.46 |40.86$\pm$0.64 |56.25$\pm$0.50 |
> |Rand |18.33$\pm$0.37 |52.20$\pm$0.43 |59.67$\pm$0.73 |34.73$\pm$0.68 |15.50$\pm$0.71 |37.00$\pm$0.76 |52.29$\pm$0.44 |
> |SVD |53.65$\pm$0.54 |71.08$\pm$0.54 |70.89$\pm$0.69 |75.67$\pm$0.49 |73.50$\pm$0.66 |40.57$\pm$0.55 |56.82$\pm$0.52 |
> |FUG |- |- |- |88.22$\pm$0.09 |30.70$\pm$0.96 |42.54$\pm$0.30 |58.16$\pm$0.27 |
> |GraphAny |- |- |- |82.94$\pm$0.82 |79.41$\pm$0.35 |38.36$\pm$0.53 |59.96$\pm$0.02 |
> |Unified I/O (node) |52.83$\pm$1.06 |69.65$\pm$0.36 |70.73$\pm$0.26 |89.85$\pm$0.18 |82.32$\pm$0.97 |42.58$\pm$0.17 |59.99$\pm$0.05 |
> |Unified I/O (graph) |**54.88$\pm$0.40** |73.09$\pm$0.43 |**72.45$\pm$0.44** |80.29$\pm$0.55 |80.30$\pm$0.70 |41.35$\pm$0.43 |57.73$\pm$0.66 |
> |||||1-shot ||||
> |GraphCL |16.71$\pm$0.69 |52.85$\pm$1.53 |51.89$\pm$3.68 |41.39$\pm$2.64 |27.83$\pm$1.20 |OOM |OOM |
> |Lap |17.33$\pm$2.46 |56.88$\pm$1.66 |55.62$\pm$1.91 |45.71$\pm$2.18 |22.80$\pm$2.00 |28.57$\pm$1.90 |53.45$\pm$1.88 |
> |Rand |17.68$\pm$2.06 |55.36$\pm$1.86 |49.56$\pm$1.56 |15.67$\pm$1.78 |16.20$\pm$2.26 |26.61$\pm$1.93 |52.56$\pm$2.16 |
> |SVD |17.87$\pm$1.37 |56.80$\pm$1.35 |55.47$\pm$1.60 |43.46$\pm$1.27 |18.90$\pm$2.44 |24.78$\pm$1.37 |53.45$\pm$1.17 |
> |FUG |- |- |- |27.26$\pm$0.36 |41.83$\pm$0.26 |27.58$\pm$0.11 |49.93$\pm$0.01 |
> |GraphAny |- |- |- |62.87$\pm$0.29 |53.63$\pm$1.03 |25.03$\pm$0.48 |49.65$\pm$0.48 |
> |Unified I/O (node) |17.82$\pm$1.27 |48.21$\pm$1.34 |52.27$\pm$1.75 |59.89$\pm$0.80 |43.94$\pm$0.50 |33.47$\pm$0.14 |57.90$\pm$0.15 |
> |Unified I/O (graph) |**18.08$\pm$1.82** |**58.89$\pm$2.40** |**56.07$\pm$0.81** |60.33$\pm$2.16 |25.90$\pm$0.92 |29.96$\pm$0.95 |54.60$\pm$1.47 |
> |||||3-shot ||||
> |GraphCL |18.16$\pm$0.88 |57.13$\pm$1.46 |53.73$\pm$0.82 |55.41$\pm$1.41 |34.97$\pm$0.64 |OOM |OOM |
> |Lap |20.42$\pm$1.29 |58.23$\pm$1.23 |58.60$\pm$1.26 |45.35$\pm$1.10 |34.80$\pm$1.39 |24.62$\pm$1.35 |54.51$\pm$1.26 |
> |Rand |18.62$\pm$1.28 |56.30$\pm$1.16 |51.38$\pm$1.16 |19.02$\pm$1.31 |30.70$\pm$1.37 |25.64$\pm$1.09 |53.04$\pm$1.43 |
> |SVD |20.67$\pm$1.10 |58.28$\pm$1.16 |55.95$\pm$1.35 |50.76$\pm$1.36 |35.60$\pm$1.26 |21.62$\pm$1.23 |53.26$\pm$1.13 |
> |FUG |- |- |- |50.59$\pm$0.29 |47.77$\pm$0.29 |24.02$\pm$0.19 |49.83$\pm$0.09 |
> |GraphAny |- |- |- |70.04$\pm$1.43 |66.32$\pm$1.21 |24.74$\pm$0.34 |54.71$\pm$0.18 |
> |Unified I/O (node) |19.00$\pm$1.93 |49.07$\pm$1.68 |54.46$\pm$1.36 |68.33$\pm$0.28 |49.21$\pm$0.91 |35.32$\pm$0.29 |57.64$\pm$0.06 |
> |Unified I/O (graph) |**22.05$\pm$2.00** |**59.32$\pm$0.93** |**60.46$\pm$1.57** |65.78$\pm$2.27 |38.90$\pm$1.17 |23.97$\pm$1.48 |55.56$\pm$1.43 |

---

> ### Author Response · Authors · 2025-11-20
> **Response to Reviewer RULu, Part 3/3**
>
> **Q**: Why does this method outperform GraphAny on heterophilic graphs?
>
> **Response**: The performance improvement is pronounced by the differences in the I/O design. Our proposed Unified I/O provides:
> - `Learnable input`: Fig.S6 (revision S11) indicates that preserving and capturing patterns in node features is important to the heterophilic task.
>     - GraphAny computes each input–to-label mapping using linear analytical solutions, and subsequently learns to fuse these mappings. Because each mapping is parameter-free and derived solely from the limited observed labels, it may introduce bias and lead to information loss at the initial input stage.
>     - Unified I/O uses a learnable parametric function to compute the input mapping, allowing the module to be optimized during pre-training directly at the input stage.
> - `Unified input for any backbone`
>     - `GraphAny is constrained with the less effective backbones` by employing linear GNNs as backbones to compute the input-to-label mapping.
>     - `Our Unified I/O empowers the usage of any GNN as a model backbone` by unifying the number and semantics of the feature channels.
>
> [R1] TUDataset: A collection of benchmark datasets for learning with graphs. ICMLWorkshop'20
>
> [R2] Graph contrastive learning with augmentations. NeurIPS'20

---

> > ### Comment · Reviewer_RULu · 2025-11-21
> >
> > Thank you very much for your responses. These responses have deeply touched me. I think some of the ideas presented in this work are very innovative and valuable. I believe that the fields of universal graph modeling and graph foundation models are still in a very preliminary exploration stage. Therefore, although this paper has some flaws, as an exploration effort, it is still very valuable and insightful.

---

> > > ### Author Response · Authors · 2025-11-22
> > >
> > > Thank you very much for your encouraging feedback. We truly appreciate your recognition of the work as a valuable and insightful exploration effort.
> > >
> > > Your constructive comments have been immensely helpful for improving the paper. Following your guidance, we will keep refining the manuscript, ensuring that all the issues you highlighted are properly addressed in the revision.
> > >
> > > Please feel free to let us know if there are any remaining issues. We are standing by to further clarify or elaborate as needed.
> > >
> > > Thank you again for your warm support!

---

### Official Review · Reviewer_Btqw · 2025-11-01

**Soundness:** 2
**Presentation:** 3
**Contribution:** 2
**Rating:** 2
**Confidence:** 4

**Summary:**

This paper proposed a new pre-training framework for graph data. Considering the varying input feature dimension and output label classes, the authors mainly devised I/O modules for input and output, which avoid direct connection with input data and actual downstrem tasks.

**Strengths:**

1. The motivation of finding graph foundation model is meaningful.
2. The organization is good to follow.

**Weaknesses:**

1. Techniques. \
At input side, the authors input feature similarity matrix, something like X^TX, rather than raw feature, followed by a all-one vector, so the model always receives the same length of input. But this operation is oversimplified, and lose a lot of information. Consider two cases, 1) one-hot raw features (e.g., user / item feature in recommendation), so X^TX is 0, 2) some datasets may have thousands of dimension features, and some datasets (e.g., molecule QM9) only have very limited feature dimensions, so X^TX could reflect totally different feature relations. How can we deal them uniformly? Further, features may appear very complex patterns, and only X^TX cannot fully capture them.\
At output side, different datasets have different number of classes, from single digits to hundreds / thousands. How to set the number of pseudo lables? Also, different classes sometimes are not totally independent, while pseudo labels are selected evenly in the sphere space.

2. Experiments. 1) In table 1, traditional semi-supervised GNNs are also need to compare, to show the necessity of pre-training. 2) In table 2, the proposed method has obvious improvement only on COMPUTERS if consider std.

**Questions:**

In Fig. 3 (b), why did the GNNs not occur over-smoothing, when stacking many layers?

---

> ### Author Response · Authors · 2025-11-20
> **Response to Reviewer Btqw, Part 1/2**
>
> Thank you very much for your valuable suggestions that help us improve our paper. We hope our response can adequately address your concerns. The discussion and new results have been added to the revision.
>
> **W1**: Effectiveness of the input module.
>
> **Response**: We would like to respectively clarify that `our input module takes raw features as input`. The similarity-like matrix is actually employed to construct the input mapping matrix $W$, decoupling the number of parameters from the dimensionality of the input features.
> - (Can this mapping capture complex patterns) `Yes`, we have provided an information loss study in Tab.S7 in the initial submission (revision S10). Results show that our method `does not cause severe information loss compared to traditional I/O` (i.e., MLP) in the supervised setting, demonstrating its effectiveness in learning effective I/O mappings to capture complex patterns.
> - (Handling one-hot raw features) The elements in the main diagonal $X^\top X$ are `nonzero`, representing the number of node samples activated in $X$ by each feature dimension. This ensures the input features are not multiplied by zero.
> - (Handling various datasets) Our method addresses this by ensuring adaptability to each specific dataset while maintaining the mapping target of the input module.
>     - Mapping as relations: Our submission proves that any mapping can be decomposed as the relation between two matrices. For the input module, the left matrix $S_{\mathtt{src}}$ (written as L for brevity) corresponds to the input data, and the right matrix $S_{\mathtt{tgt}}$ (written as R) corresponds to the hidden space.
>     - `Traditional fails with fixed L and R`. Traditional methods directly optimize L and R on each dataset, making the element values in L tailored to particular datasets and their quantity scales to the number of features in those datasets.
>     - `Ours can with adaptive L for specific inputs and fixed R for uniform mapping`.
>         - `Parametric function, adaptive computing`: A parametric function is employed to compute L based on specific inputs, ensuring adaptability to various datasets.
>         - `Shared function, same semantics, uniform mapping`: The parametric function is constructed as a shared relation function, targeting the same relation patterns across different datasets. This maintains the semantics of L, empowering the usage of the same R for a uniform mapping target.
>             - Why the same semantics? Measuring the relation between L and R requires alignment. For example, heights and weights can not be compared. Therefore, to ensure a fixed R as a uniform mapping target, it is important to maintain the semantics of L.
>             - Why relation function maintain semantics? Take the scaled product $X^\top X$ as an example. No matter what the original semantics of each feature dimension are, $X^\top X$ always measures their similarity. Even though the specific relation values vary, the semantical meaning remains.
>             - Empirical Proof. Results in [this link to the anonymized repository](https://anonymous.4open.science/r/Unified-IO-F951/semantics_alignment.md) show that `our method can align feature semantics in the common space`.
> - [Revision]: We have revised Section 3 to explicitly specify the relation between the similarity-like matrix and the mapping matrix.
>
> **W2**: Construction of the output module.
>
> **Response**:
> - (How to set the number of pseudo labels) The number of pseudo labels does not affect parameter volumes, and thus can be directly set as the number of real labels.
> - (Whether incorporating class relation in the pre-training) `Disregarding label knowledge for pre-training demonstrates better performance`.
>     - Why not: Although incorporating label knowledge (reflecting their relations) provides more information, it hinders the adaptability of the pre-trained models to downstream tasks when the label semantics are distinct from pre-training data.  As shown in the manuscript Tab.2, baselines relying on the label knowledge during pre-training `achieve inferior performance compared to our Unified I/O`.
>     - Disregarding is better: In contrast, our Unified I/O disregards the label knowledge during training and employs pseudo labels to do inference. Therefore, the predictions are made with pseudo labels regardless of the real label semantics, ensuring adaptability to various downstream datasets.

---

> ### Author Response · Authors · 2025-11-20
> **Response to Reviewer Btqw, Part 2/2**
>
> **W3-1**: Comparison with supervised GNNs in Tab.1.
>
> **Response**: Thank you for your suggestion. The new results have been included in the revision. As shown in Tab.R1,
> - Original split: Pre-training with Unified I/O `achieves comparable performance` with traditional supervised GNNs. On Cora, Chameleon, and Actor, pre-training models even surpass supervised baselines.
>     - We have also provided fine-tuning results in manuscript Tab.3, where `pre-trained models surpass supervised baselines with minimal hyperparameter tuning`.
> - 1/3-shot: Pre-training with Unified I/O `achieves better performance` compared to the supervised baselines.
>
> These results demonstrate that `model pre-training is necessary` for graph learning, particularly in `data-scarce settings`.
>
> [Revision]: The new results have been included in the revision Tab.1.
>
> |**Tab.R1** Comparison with supervised GNNs |AmazonComputers |Cora |CoauthorPhysics |arxiv-year |twitch-gamer |tolokers |CHAMELEON |ACTOR |
> |--: |:--: |:--: |:--: |:--: |:--: |:--: |:--: |:--: |
> |||||original split |||||
> |GraphSAGE |90.36$\pm$0.16 |80.10$\pm$0.25 |95.17$\pm$0.42 |45.86$\pm$0.20 |**61.09$\pm$0.32** |**80.72$\pm$0.26** |55.86$\pm$0.34 |31.34$\pm$0.20 |
> |GCN |**91.09$\pm$0.13** |81.80$\pm$0.28 |**95.52$\pm$0.20** |**48.03$\pm$0.41** |59.44$\pm$0.18 |77.56$\pm$0.40 |61.74$\pm$0.30 |22.67$\pm$0.12 |
> |GAT |87.67$\pm$0.23 |79.50$\pm$0.29 |91.30$\pm$0.25 |47.85$\pm$0.36 |59.12$\pm$0.25 |76.48$\pm$0.18 |48.09$\pm$0.21 |22.72$\pm$0.44 |
> |Unified I/O |89.85$\pm$0.18 |**82.32$\pm$0.97** |92.85$\pm$0.48 |42.58$\pm$0.17 |59.99$\pm$0.05 |76.44$\pm$0.44 |**62.13$\pm$0.43** |**35.35$\pm$0.26** |
> |||||1-shot |||||
> |GraphSAGE |28.18$\pm$0.31 |15.10$\pm$0.54 |22.30$\pm$0.27 |22.63$\pm$0.36 |50.96$\pm$0.36 |61.03$\pm$0.54 |30.18$\pm$0.74 |20.19$\pm$0.66 |
> |GCN |36.80$\pm$0.53 |32.00$\pm$0.98 |53.44$\pm$0.99 |26.87$\pm$0.75 |54.83$\pm$0.68 |66.17$\pm$0.73 |28.27$\pm$0.76 |23.44$\pm$0.59 |
> |GAT |29.93$\pm$0.23 |20.30$\pm$0.52 |19.17$\pm$0.31 |20.79$\pm$0.47 |53.27$\pm$0.46 |41.46$\pm$0.54 |18.60$\pm$0.68 |19.21$\pm$0.39 |
> |Unified I/O |**59.89$\pm$0.80** |**43.94$\pm$0.50** |**85.13$\pm$0.68** |**33.47$\pm$0.14** |**57.90$\pm$0.15** |**68.51$\pm$0.13** |**32.00$\pm$0.11** |**25.69$\pm$0.12** |
> |||||3-shot |||||
> |GraphSAGE |52.98$\pm$0.59 |27.70$\pm$0.48 |52.32$\pm$0.58 |27.80$\pm$0.51 |54.02$\pm$0.62 |62.02$\pm$0.45 |32.46$\pm$0.71 |19.39$\pm$0.49 |
> |GCN |65.64$\pm$0.30 |37.10$\pm$0.55 |76.44$\pm$0.48 |25.17$\pm$0.35 |53.40$\pm$0.58 |61.25$\pm$0.64 |33.57$\pm$0.41 |19.84$\pm$0.27 |
> |GAT |37.49$\pm$0.80 |18.50$\pm$0.99 |50.06$\pm$0.79 |24.07$\pm$0.76 |53.14$\pm$0.73 |68.89$\pm$0.81 |26.25$\pm$0.93 |20.88$\pm$0.43 |
> |Unified I/O |**68.33$\pm$0.28** |**49.21$\pm$0.91** |**84.68$\pm$0.67** |**35.32$\pm$0.29** |**57.64$\pm$0.06** |**71.77$\pm$0.25** |**33.76$\pm$0.11** |**25.20$\pm$0.11** |
>
> **W3-2**: Significance of the performance improvement in Tab.2.
>
> **Response**: To show the significance of our performance improvement, we apply statistical significance analysis to the results. Under the original split, 1-shot, and 3-shot evaluation settings, the test `reports a maximum p-value of 0.015 across all comparisons`, which is `below the 0.05 significance level`, indicating `statistically significant improvement`.
>
> Specifically, we use the Wilcoxon signed-rank test for the analysis. Due to the large standard deviation of the baseline models, the paired t-test becomes unstable. In contrast, the Wilcoxon test only uses the rank of performance differences instead of their absolute magnitude, and thus is insensitive to variance imbalance.
>
> **Q**: Why did the GNNs not experience over-smoothing when stacking many layers in Fig.3 (revision Fig.5)?
>
> **Response**: This can be attributed to the residual connection strategy during model training.

---

> ### Author Response · Authors · 2025-11-24
>
> Dear reviewer Btqw,
>
> The anonymous GitHub website is currently affected by server instability. If you encounter difficulty accessing it, we kindly refer you to Fig.S13 in our revised manuscript for the feature alignment results. We apologize for the inconvenience.
>
> Best Regards,
>
> Authors of Submission 787

---

### Author Response · Authors · 2025-11-29

Dear ACs, SACs, and PCs,

We would like to express our sincere gratitude for your efforts throughout this review process, especially given the unexpected circumstances. Thank you for your coordination and guidance. We are also grateful for the thoughtful comments and time the reviewers invested in evaluating our work. Their feedback has been invaluable in helping us further improve the manuscript.

Before the major OpenReview data leak (Nov 27), our paper had received scores of `6, 4, 4, 2`:
- Reviewer RULu (6) was satisfied with our responses, raised no further questions, and praised the work as “very valuable and insightful” as an exploration effort.
- Reviewer fnYM (4) acknowledged that his/her concerns had been partially addressed, primarily raised the score to 4, and asked a set of follow-up questions.
- Reviewer nQLH (4) found our responses satisfactory and raised several questions for further discussion.

**Summary of strengths**. We sincerely appreciate that the reviewers find our work:
- Well-organized and easy to follow. (reviewer Btqw, fnYM)
- Built upon a meaningful/clear motivation. (reviewer Btqw, fnYM)
- Exploring a direction with strong practical significance for graph foundation models. (reviewer fnYM, nQLH)
- Proposing a novel/clean method. (reviewer RULu, nQLH)
- Providing formal problem formulation and theoretical derivations to make the proposed method convincing and conceptually coherent. (reviewer fnYM)
- Demonstrating strong performance under various settings. (reviewer RULu, nQLH)
- Providing well-done empirical evaluation/interesting insights for GNN pre-training. (reviewer RULu, nQLH)

Below, we summarize our **point-to-point responses during the rebuttal phase**. We use [$\surd$] to denote that the response has been recognized by at least one reviewer, [D] to denote that the point is still under discussion, [O] to denote that the response has not received feedback.
- Summarizing the key differences/advantages to
    - [$\surd$] FUG (reviewer RULu: Part1/3-W1)
    - [$\surd$] GraphAny (reviewer RULu: Part3/3-Q; reviewer fnYM: Part2/4-W2)
    - [$\surd$] ZeroG (reviewer fnYM: Part2/4-W2)
    - [$\surd$] Deep Sets (Reviewer nQLH: Part1/2-W1)
    - [O] Traditional I/O modules (reviewer Btqw: Part1/2-W1, W2)
- Clarity
    - [$\surd$] Fig.5 (revision Fig.7) (reviewer nQLH: Part1/2-W2)
    - [$\surd$] State the conclusion for Fig.S6 (revision Fig.S11) in the main paper (reviewer nQLH: Part2/2-W4)
    - [$\surd$] Main figure (reviewer nQLH: Part2/2-Minor W)
    - [D] Definition for $S_\mathtt{src}$/$S_\mathtt{tgt}$/$\mathtt{f_{in}}$ (reviewer nQLH: Part1/2-W1, Follow-up-W1)
        - Reviewer nQLH recognized the clarity of our rebuttal.
        - Reviewer Btqw and fnYM recognized the writing of the submission.
- Providing results for
    - [$\surd$] Unification of the feature/label semantics (reviewer RULu: Part1/3-W2)
    - [$\surd$] Superior performance on more tasks (reviewer RULu: Part2/3-W3; reviewer fnYM: Part1/4-W1)
    - [$\surd$] More pre-training conditions
        - [$\surd$] Scaling pre-training data with the minimum dataset size (reviewer fnYM: Part3/4-W3-1, Follow-up-W3; reviewer nQLH: Part2/2-W3-2)
        - [$\surd$] Scaling with the parameter volumes (reviewer fnYM: Part3/4-W3-2; reviewer nQLH: Part2/2-W3-1)
    - [$\surd$] Proof of stable training (reviewer fnYM: Part4/4-W4)
    - Superior performance compared with
        - [$\surd$] FUG/SAMGPT/RiemannGFM (reviewer RULu: Part1/3-W4)
        - [$\surd$] Self-supervised methods (reviewer fnYM: Part2/4-W2)
        - [O] Supervised GNNs (reviewer Btqw: Part2/2-W3-1)
    - [O] The statistical significance of the performance gains (reviewer Btqw: Part2/2-W3-2; reviewer fnYM: Follow-up-W1)

We hope this summarization will assist in the continued discussion and evaluation of our work.

Best regards,

Authors of submission 787

---

### Meta-Review · Area_Chair_Qdzi · 2025-12-31

**Summary:**

In this paper, the authors study whether pure GNNs, without using language models, can be pre-trained on multiple graph datasets with different feature and label spaces. The authors propose unified input and output modules so that the main GNN parameters can be shared across datasets. The problem is interesting and timely, especially for graph foundation models. The research problem is timely and relevant to emerging work on graph foundation models, and the paper is generally well written and clearly motivated. However, reviewers had mixed opinions with many concerns. Some of them are addressed during the rebuttal, while several important concerns remain after the rebuttal.

**Reviewer Concerns:**

Concerns addressed by the authors:

1. The authors clarify raw features are still used and provide an information-loss style study and additional discussion of how the similarity-like construction is used for mapping rather than replacing features.

2. The authors added comparisons vs standard supervised GNNs and argue pretraining is particularly beneficial in few-shot settings.

3. The authors added experiments extending beyond node classification to include graph-level datasets/tasks and discuss cross-task transfer gaps.

4. The authors added discussion and additional comparisons (at least for some methods/settings).

Outstanding concerns:

1. Despite added empirical evidence, the core claim of :semantic unification” remains only partially supported. Also, it lacks a compelling theoretical justification.

2. Though, new baselines/analyses were added, performance is still mixed across datasets/settings (some reviewers questioned whether the chosen baseline set fully reflects SOTA in each regime)

**Reviewer Scores:**

RULu: stays positive (6).

nQLH: likely stays around 4 (generally satisfied, minor remaining clarifications).

Btqw:  given added supervised comparisons + significance testing, the reviewer may move from 2 to 4


fnYM: the reviewer could move from 2 to 4 due to added broader-task evaluation and additional comparisons/analysis, though concerns about mixed competitiveness and justification remain.

---

### Decision · Program_Chairs · 2026-01-26

Reject